# Advancing Machine-Generated Text Detection from an Easy to Hard Supervision Perspective

**Chenwang Wu**[1]     **Yiu-ming Cheung**[1*]     **Bo Han**[1]     **Defu Lian**[2]

[1]Department of Computer Science, Hong Kong Baptist University, Hong Kong, China
[2]School of Computer Science, University of Science and Technology of China, Hefei, China
{cscwwu, ymc, bhanml}@comp.hkbu.edu.hk, liandefu@ustc.edu.cn

## Abstract

Existing machine-generated text (MGT) detection methods implicitly assume labels as the "golden standard". However, we reveal boundary ambiguity in MGT detection, implying that traditional training paradigms are inexact. Moreover, limitations of human cognition and the superintelligence of detectors make inexact learning widespread and inevitable. To this end, we propose an easy-to-hard enhancement framework to provide reliable supervision under such inexact conditions. Distinct from knowledge distillation, our framework employs an easy supervisor targeting relatively simple longer-text detection tasks (despite weaker capabilities), to enhance the more challenging target detector. Firstly, longer texts targeted by supervisors theoretically alleviate the impact of inexact labels, laying the foundation for reliable supervision. Secondly, by structurally incorporating the detector into the supervisor, we theoretically model the supervisor as a lower performance bound for the detector. Thus, optimizing the supervisor indirectly optimizes the detector, ultimately approximating the underlying "golden" labels. Extensive experiments across diverse practical scenarios, including cross-LLM, cross-domain, mixed text, and paraphrase attacks, demonstrate the framework's significant detection effectiveness. The code is available at: `https://github.com/tmlr-group/Easy2Hard`.

## 1   Introduction

High-quality machine-generated text (MGT) is increasingly prominent due to its potential in areas like content creation [1], intelligent education [2], and customer service [3]. However, its misuse presents significant challenges, including misinformation [4], phishing attacks [5], and malicious impersonation [6]. Compounding this is research [7] indicating humans struggle to distinguish MGT from human-generated text (HGT), performing little better than random chance. This tension between MGT's risks and limited human detection highlights the urgent need for effective detection methods.

Existing detection methods can be primarily divided into: (1) metric-based methods, which detect differences by capturing the intrinsic statistical properties between MGTs and HGTs, using statistics such as Likelihood [8] and Entropy [9, 10]. In addition, a series of works represented by DetectGPT [11], Fast-DetectGPT [12], and DALD [13] utilize the probability curvature of text under LLMs as key detection features. (2) Model-based methods do not rely on explicit feature engineering. They input the full text into deep learning models, which automatically learn and extract discriminative implicit features end-to-end. This category includes energy-based models [14], GNN-based model [15], LLM [16], and other methods such as SeqXGPT [17], AI-Catcher [18], and RADAR [19]. With the powerful text representation learning capabilities of deep learning models, model-based methods typically demonstrate higher effectiveness in terms of detection performance and robustness.

---

*Corresponding author: Yiu-ming Cheung (ymc@comp.hkbu.edu.hk).

39th Conference on Neural Information Processing Systems (NeurIPS 2025).

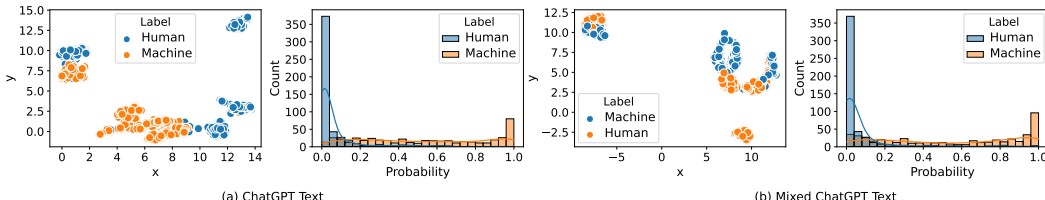

(a) ChatGPT Text          (b) Mixed ChatGPT Text

Figure 1: Boundary fuzziness evaluation between (mixed) MGT and HGT, which illustrates the latent space distribution and prediction confidence distribution under pure (Sub-Fig. 1 & 2) and mixed (Sub-Fig. 3 & 4) texts. The mixed text is obtained by replacing 1/4 of MGTs with HGTs.

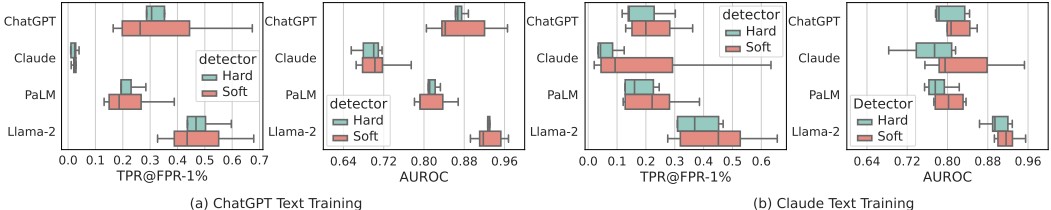

(a) ChatGPT Text Training          (b) Claude Text Training

Figure 2: Performance comparison with and without using soft labels in mixed text (1/4 of MGT was replaced with HGT). The detector is ChatGPT-D [20].

The aforementioned methods often implicitly rely on the assumption of perfect data labels; however, this may not hold in MGT detection. Specifically, considering the prevalence of human-machine collaboration scenarios [21] and the powerful generative capabilities of LLMs, MGTs may explicitly or implicitly resemble HGTs, leading to blurred boundaries. For instance, MGTs and HGTs overlap significantly in the latent space, especially in mixed texts, as shown in Fig. 1-1 and 1-3. Besides, Fig. 1-2 and 1-4 show that HGT prediction probabilities concentrate near 0 (0 indicates HGT, while 1 indicates MGT) while MGTs with human-like features exhibit a broad distribution across $[0, 1]$, reinforcing their boundary ambiguity. To further validate the inexactness of hard labels, we trained a detector on mixed texts (1/4 MGT replaced with HGT) using label smoothing [22] (with a small smoothing factor of 0.05). Notably, even though we cannot access the underlying "golden" labels, the soft label with a small smoothing factor is closer to the golden labels than hard labels in significant mixed texts. As shown in Fig. 2, the results clearly indicate that a higher upper bound for label smoothing means it has the potential for enhancement. These results collectively indicate that the existing learning paradigm is inexact. See Appendix D.5 for more details and results.

Despite this clear insight, resolving the issue is challenging. MGT detection task surpasses human cognition, rendering inexact label annotation widespread. Furthermore, existing MGT detectors are often more capable than humans, making it difficult to reliably assess the detection quality, i.e., the supervisor may be weaker than the detector, rendering inexact supervision inevitable. In summary, unlike the limited accidental errors in noisy label learning, the fundamental difficulty in annotation and evaluation makes inexact supervision widespread and inevitable. This leads us to ask:

*Is it possible to effectively learn from the supervisor when the provided label is inexact and the detection quality is difficult to assess?*

Our work is dedicated to exploring this question. To achieve this, some key issues need to be solved:

- **RQ1**. How can the supervisor provide more reliable supervision signals under inexact labels and unclear detection quality?

- **RQ2**. How can the detector be improved based on the feedback signals provided by supervisors?

To address this, we propose an easy-to-hard supervision framework to enhance detection. Its core idea is to utilize a carefully designed supervisor, focused on the relatively easier task of longer-text detection, to provide reliable feedback signals for guiding a more challenging target detector. To ensure the supervisor's reliable supervision signals, we meticulously consider its data quality improvement and structure design (**RQ1**). Firstly, we construct longer texts as supervisor data, which can theoretically alleviate the impact of inexact labels, laying the foundation for reliable supervision. Secondly, we deviate from traditional approaches that rely on soft labels to provide supervision

signals, because it is unclear whether the soft label is more informative under unclear detection quality. Instead, we structurally integrate the detector into the supervisor's design, establishing a connection between the supervisor and the detector, where the supervisor's performance serves as a lower bound for the detector's capabilities. This coupled structure allows the detector to be indirectly optimized via the supervisor (**RQ2**), finally encouraging convergence to the underlying "golden" labels. Our contributions can be summarized as follows:

- We analyze the inexact supervision inherent in existing MGT detection methods and highlight this widespread and inevitable limitation as a critical direction for future research.
- We propose an easy-to-hard supervision framework for enhanced detection, theoretically proving its optimization properties that facilitate convergence towards the underlying "golden" labels.
- We demonstrate the effectiveness of the proposed framework with negligible latency in various practical scenarios, including cross-LLM, cross-domain, mixed text, and paraphrasing attacks.

## 2    MGT Detection from an Inexact Supervision Perspective

**Traditional Learning Paradigm of MGT Detection**. Let $\mathcal{S}$ denote the set of all possible text sequences, and $\mathcal{Y} = \{0, 1\}$ denote the hard-label space, where 0 represents HGT and 1 represents MGT. Each sequence $s \in \mathcal{S}$ can be understood as multiple dependent sentences. Then, the dataset can be represented as $\mathcal{D} = \{(x_i, y_i)\}_{i=1}^{N}$, where $x_i \in \mathcal{S}^{n_i}$ is a text composed of $n_i$ sequences from $\mathcal{S}$, and $y_i \in \mathcal{Y}$ is its corresponding hard label. This definition understands texts as a collection of multiple sequences, similar to the definition in the existing work [23] [2]. For the detector $f$ parameterized by $\theta_f$, the learning paradigm uses the cross-entropy loss, which is usually as follows,

$$\mathcal{L}_{\text{train}} = -\frac{1}{N} \sum_{i=1}^{N} \left( y_i \log f\left(x_i, \theta\right) + (1 - y_i) \log(1 - f\left(x_i, \theta\right)) \right).$$

**Inexact Supervision Learning of MGT Detection**. In the above learning paradigm, it is implicitly assumed that the label $y_i$ is "golden standard"; however, this may not hold in MGT detection.

First, the boundary between MGT and HGT is often unclear, leading to potentially inexact hard labels. This blurred distinction stems from (1) human-machine collaboration [24, 25], where texts involve both human and machine contributions (e.g., LLM drafting followed by human editing); (2) the powerful generative capabilities of LLMs, where they trained on extensive human data are capable of producing highly human-like texts, especially for specific text types like short texts [26]. Therefore, MGTs may (explicitly or implicitly) contain human-like features, rendering hard labels inexact. This is also confirmed by the empirical results (Fig. 1 and Fig. 2) shown in the Introduction above, and more results can be found in Appendix D.5.

An intuitive solution like knowledge distillation [27] faces a key challenge: its effectiveness mainly relies on the target model being weaker than the strong teacher model. This is difficult for MGT detection because human cognition in distinguishing MGT is limited (e.g., [7] reports that human accuracy for GPT-3 texts is 49.9%). Besides, current MGT detectors show super intelligence and are even smarter than humans (e.g., RADAR achieves 87.3% of accuracy on Essay), making it even difficult to distinguish the quality of two predictions. For example, it is unclear whether 90% or 95% confidence is better for an MGT. Therefore, in MGT detection, the supervisor may be weak, and even picking a strong teacher model is challenging.

Noisy label learning (NLL) [28, 29] is another consideration, but its classic assumption is that there exist true, clearly discrete labels with only a limited random error occurring during the annotation process. Instead, in MGT detection, as emphasized above, limitations of human cognition in distinguishing MGT lead to widespread and complex inexactnesses in labeling (with correct categories), rather than limited random errors. Consequently, existing NLL techniques may require significant redefinition to accommodate this structure of inexact labels. See Appendix A.4 for more discussion.

In summary, due to human cognition's limitations and detectors' superintelligence, inexact supervision learning in MGT detection becomes widespread and inevitable. Therefore, this paper focuses on designing effective supervised learning algorithms under conditions where labels are generally inexact and detectors are difficult to evaluate reliably.

---

[2]This is a natural assumption where text often has multiple topics, and sentences for each topic are dependent.

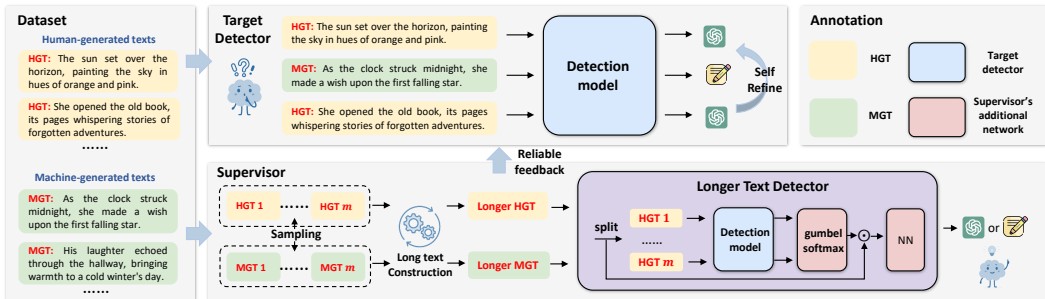

Figure 3: The easy-to-hard supervision framework, which uses a carefully designed supervisor, focused on relatively simple task of longer-text detection, to guide a more challenging target detector.

# 3 Easy to Hard Supervision Enhancement Framework

To effectively learn from the inexact supervision lens, a supervisor capable of providing reliable supervision signals and a detector capable of continuous learning from this feedback are essential. To this end, we propose an easy-to-hard supervision enhancement framework, as shown in Fig. 3. It includes two key components: an "easy" supervisor, carefully designed for the simpler task of longer-text detection, and a "hard" target detector for the more challenging MGT detection. Briefly, the supervisor is carefully designed from both data and architecture aspects to provide more reliable feedback signals to the detector, while the detector continuously improves by integrating this feedback with its original learning paradigm. We will detail how they are designed to achieve this subsequently.

## 3.1 Supervisor: Providing Reliable Supervision Signals

In MGT detection, given the inexact labels and difficulty in reliably evaluating detection quality, the supervisor must provide as reliable supervision signals as possible. To achieve this crucial goal, our approach focuses on two key aspects: improving the supervisor's training data quality and designing its model architecture.

**Data Quality Improvement**. A supervisor's ability to provide reliable supervision is closely linked to its own performance; thus, effective supervisor learning is a fundamental prerequisite for achieving reliable supervision. Recognizing the inherent imprecision in training data, improving its training data quality is a natural and feasible method. Drawing upon existing research [23], which indicates a positive correlation between text length and MGT detection and implies that longer texts are easier to detect accurately, we focus the supervisor on detecting longer texts by constructing new longer text as its input data.

Specifically, let the longer text-label pair be denoted as $(x_{long}, y_{long})$, then its generation rule is: first sample $y_{long} \sim \mathcal{Y}$, and then the corresponding longer text $x_{long}$ is constructed as follows:

$$x_{long} = \oplus_{j=1}^{k} x^{(j)}, \text{ where each } x^{(j)} \sim \{(x, y) \in \mathcal{D} | y = y_{long}\}. \tag{1}$$

Here, $\oplus$ denotes the text splicing operator. The rationale behind this splicing operation is that joining MGTs or HGTs does not alter their fundamental nature as machine-generated or human-written text, respectively. The following theorem will prove this design's rationality by revealing the relationship between the distribution difference and the length of the text.

**Theorem 3.1** (**Distribution Difference for Longer Text**). *Let $h(s)$ and $m(s)$ be the distributions for human-generated and machine-generated sequences on $s \in \mathcal{S}$, respectively, with the total variation distance $TV(m, h) = \delta > 0$. For the text contains $n$ sequences, let $\alpha \geq 0$ denote the ratio of human-like component incorporated in MGT. For longer text formed by concatenating $k$ independent length-$n$ texts, the total variation distance between their distributions, $TV_{long}$ can be bounded by:*

$$1 - 2exp(-\frac{nk(1-\alpha)^2\delta^2}{2}) \leq \mathrm{TV}_{long} \leq 1 - (1 - \delta)^{nk(1-\alpha)}.$$

The theorem indicates that increasing text number $k$ (original $k = 1$) for longer text will amplify the distribution difference $TV_{long}$ between HGT and MGT. This leads to clearer classification boundaries

and reduced hard label ambiguity. Training supervisors with this more discriminative data is crucial for achieving its better performance and providing reliable supervision. Furthermore, a larger mixed text ratio $\alpha$ tends to a smaller distribution distance, theoretically supporting the conjecture in Section 2 regarding unclear boundaries due to (explicit or implicit) mixed text presence.

**Reliable Feedback**. Under the reliable learning of the supervisor, the core issue lies in providing reliable supervision signals to the target detector. An intuitive approach is to mimic knowledge distillation by directly using the supervisor's soft labels as supervision. However, this faces challenges in MGT detection. First, the detector's superintelligence makes it unclear if the supervisor's soft labels offer superior information. Second, while the supervisor also performs detection tasks, it focuses on longer texts with different distribution characteristics; these domain differences make ensuring soft label quality challenging when used for more difficult target detection.

Therefore, instead of directly providing soft label supervision, we cleverly integrate the target detector into the supervisor's design. To achieve this, we reveal the theoretical upper bound of the supervisor's performance, and link factors influencing it to the detector's performance. This link allows the supervisor's performance to be viewed as a theoretical lower bound for the detector's capability. Thus, maximizing supervisor performance indirectly enhances detector performance. Specifically,

**Theorem 3.2** (**Detection Power for Longer Text**). *Under the assumption of Theorem 3.1, the supervisor's AUROC$_{supv.}$ satisfies*

$$\text{AUROC}_{supv.} \leq 1 - \frac{1}{2} \cdot (1 - \delta)^{2nk(1-\alpha)}.$$

The theorem reveals that the supervisor's performance is the lower bound of $1 - \frac{1}{2} \cdot (1 - \delta)^{2nk(1-\alpha)}$. If $k$ is an optimizable variable positively correlated with detector performance, then optimizing the supervisor favors larger $k$, thereby indirectly enhancing detector performance.

In the framework, $k$ represents the number of original texts $x^{(j)}$ concatenated to form the longer text for the supervisor, while the detector is aimed at identifying each original text $x^{(j)}$. To establish a positive correlation between $k$ and the detector's performance (i.e., better detector performance corresponds to a larger $k$), we revisit the detector as a gating mechanism acting on each original text $x^{(j)}$ within the supervisor's input: If the detector misclassfies text $x^{(j)}$, that it is filtered out, reducing the concatenation length $k$. In this way, a larger value of $k$ corresponds to higher accuracy. When $k$ reaches its maximum value (no filtering), the detector achieves an accuracy of 100%. Formally, we change supervisor's input from $x_{long}$ in Eq. 1 to

$$x'_{long} = \begin{cases} \oplus_{j=1}^{k} \left( x^{(j)} \odot \text{argmax}(f(x^{(j)}, \theta_f)) \right) & \text{if} \quad y_{long} = 1, \\ \oplus_{j=1}^{k} \left( x^{(j)} \odot (1 - \text{argmax}(f(x^{(j)}, \theta_f))) \right) & \text{if} \quad y_{long} = 0. \end{cases} \tag{2}$$

Here, the element-wise multiplication $\odot$ is performed at the input embedding level. To ensure the supervisor's feedback could be back-propagated to the detector, we use Gumbel Softmax [30] to replace discrete argmax. Further, we simplify the $y_i = 0$ branch and simplify Eq. 2 to

$$x'_{long} = \oplus_{j=1}^{k} \left( x^{(j)} \odot \text{Gumbel}(f(x^{(j)}, \theta_f)) \right), \tag{3}$$

i.e., let HGT distribution $h(s)$ collapse to Dirac $\delta_0$ distribution [31], which has following benefits.

**Theorem 3.3** (**Distribution Difference after HGT Distribution Collapse**). *Under the assumption of Theorem 3.1 and assuming that $m(0) \to 0$ [3], then if $h(s)$ collapses to a Dirac $\delta_0$ distribution, we have $\lim_{m(0) \to 0} TV(h, m) = 1$.*

This theorem indicates that this simplification maximizes the distribution distance between MGT and HGT. Similar to data quality improvement, this reduces the learning difficulty for the supervisor.

In summary, the supervisor $g$ parameterized by $\theta_g$ makes predictions for the long text $x_{long}$ as $g(x'_{long}, \theta_g)$, where $x'_{long}$ is calculated by Eq. 3. Assuming that the longer text dataset is $\mathcal{D}_{long}$, the supervisor's training loss $\mathcal{L}_{supv.}$ using cross entropy is as follows,

$$\mathcal{L}_{supv.} = -\frac{1}{|\mathcal{D}_{long}|} \sum_{(x_{long}, y_{long})} \left( y_{long} \log g \left( x'_{long}, \theta_g \right) + (1 - y_{long}) \log \left( 1 - g \left( x'_{long}, \theta_g \right) \right) \right). \tag{4}$$

---

[3]This is a mild assumption, where LLM usually does not correspond to zero vectors to ensure that the text has sufficient information and is non-trivial.

**Theorem 3.4** (**The Effectiveness of the Proposed Framework**). *Under the assumption of Theorem 3.1, and assuming that the MGT's golden label is approximately the proportion of pure machine-generated content distinct from HGT, if the supervisor reaches the best possible one, the detector converges to the underlying golden labels.*

See Appendix A.5 for the discussion of the rationality of this golden label approximation. Unlike traditional methods that merely fit binary hard labels, this theorem reveals how the supervisor guides the detector towards convergence with more accurate underlying "golden" labels.

### 3.2 Detector: Learning from Reliable Signals

The detector is the target optimization model, which will improve itself by learning category information from hard labels and feedback from the supervisor. First, as an enhancement framework, we do not alter the original detector's structure and training paradigm, thus ensuring flexible extensibility and convenient application to various detectors. Therefore, the first loss is the original detector loss $\mathcal{L}_{ori}$, and the specific form depends on the implementation of the detector. Secondly, the design of the supervisor is closely linked to the detector, allowing it to indirectly optimize the detector, thus the second loss is supervisor loss $\mathcal{L}_{supv.}$ shown in Eq. 4. In summary, we jointly train the supervisor and the detector, and the overall optimization objective with a coefficient $\lambda$ is as follows:

$$\theta_f, \theta_g = \arg\max_{\theta_f, \theta_g} \mathcal{L}_{ori.} + \lambda \cdot \mathcal{L}_{supv.}. \tag{5}$$

The target detector $f$ is trained only on the original, natural text, which allows it to learn the true data distribution. Instead, the concatenated text is used only to train the supervisor $g$. This intentional separation ensures the distribution shift (natural text vs. longer text) does not compromise the target detector's performance.

### 3.3 Overall Framework

Alg. 1 outlines the proposed framework's process. For each training batch, the supervisor's training data, a longer text set $\mathcal{D}'_B$, is first constructed based on the current batch data $\mathcal{D}_B$ (Line 4). Then, the original detector loss $\mathcal{L}_{ori.}$ (Line 5) and the supervisor's loss $\mathcal{L}_{supv.}$ (Line 6) are computed based on $\mathcal{D}_B$ and $\mathcal{D}'_B$, respectively. Finally, joint training is performed for the detector and supervisor (Line 7).

Notably, supervisor data is constructed from the current batch $\mathcal{D}_B$ rather than the dataset $\mathcal{D}$. This design enables the reuse of $f(x^{(j)}, \theta_f)$, rendering the computational delay for $x''$ in Eq. 4 negligible. The main training delay stems from the

---

**Algorithm 1** Easy to Hard Supervision Framework

1: **Input:** Train data $\mathcal{D} = \{(x_i, y_i)\}_{i=1}^N$, the detector $f(x, \theta_f)$, the supervisor $g(x'_{long}, \theta_g)$, training epochs $T$, learning rate $\eta$.
2: **for** $t = 0$ **to** $T - 1$ **do**
3:     **for** each batch of samples $\mathcal{D}_B \sim \mathcal{D}$ **do**
4:         Based on the texts of $\mathcal{D}_B$, Construct $N'$ longer texts by Eq. 1, denoted as $\mathcal{D}'_B$.
5:         Calculate detector's original loss $\mathcal{L}_{ori.}$.
6:         Calculate supervisor's loss $\mathcal{L}_{supv.}$ by Eq. 4.
7:         Jointly optimize $\theta_g$ and $\theta_g$ by Eq. 5.
8:     **end for**
9: **end for**
10: **Return** the trained detector $f(x, \theta_f)$.

---

supervisor's forward and backward passes. Fortunately, by comprehensively considering performance and efficiency (see Appendix D.4), the supervisor uses a simple network (e.g., the three-layer fully connected network in our work) to achieve good supervision, resulting in computational costs trivial compared to the detector's typically complex pre-trained model. Therefore, the proposed framework introduces only minimal training latency, and the detector's inference time remains unchanged due to its unmodified nature. See Appendix A for more discussion of the proposed framework.

## 4 Experiments

**Datasets and LLMs**. We conduct experiments on two public datasets, Essay [16] and DetectRL [32], to validate our effectiveness. The Essay dataset comprises MGTs generated by GPT4All, ChatGPT,

Table 1: Performance concerning TPR@FPR-1% (%) on DetectRL. The detection model is trained on text generated by PaLM.

| Method | Sentence-level | | | | | Paragraph-level | | | | |
|---|---|---|---|---|---|---|---|---|---|---|
| | PaLM | ChatGPT | Claude | Llama-2 | Avg. | PaLM | ChatGPT | Claude | Llama-2 | Avg. |
| Likelihood | $4.83_{\pm0.39}$ | $1.58_{\pm0.23}$ | $0.72_{\pm0.13}$ | $5.54_{\pm0.40}$ | 3.17 | $25.66_{\pm2.41}$ | $10.21_{\pm1.40}$ | $1.78_{\pm0.38}$ | $38.39_{\pm0.92}$ | 19.01 |
| Log-Rank | $4.84_{\pm0.41}$ | $1.23_{\pm0.25}$ | $0.72_{\pm0.13}$ | $5.25_{\pm0.87}$ | 3.01 | $27.49_{\pm1.13}$ | $11.55_{\pm1.93}$ | $2.08_{\pm0.65}$ | $41.93_{\pm0.47}$ | 20.76 |
| Entropy | $0.47_{\pm0.16}$ | $0.36_{\pm0.26}$ | $0.52_{\pm0.22}$ | $0.49_{\pm0.33}$ | 0.46 | $6.95_{\pm0.78}$ | $0.25_{\pm0.16}$ | $1.51_{\pm0.34}$ | $2.03_{\pm0.73}$ | 2.68 |
| NPR | $2.24_{\pm0.24}$ | $1.72_{\pm0.20}$ | $1.03_{\pm0.06}$ | $3.95_{\pm0.69}$ | 2.23 | $6.80_{\pm1.59}$ | $3.71_{\pm2.48}$ | $3.29_{\pm4.24}$ | $16.93_{\pm6.50}$ | 7.68 |
| DetectGPT | $0.72_{\pm0.17}$ | $0.38_{\pm0.09}$ | $0.25_{\pm0.13}$ | $0.84_{\pm0.17}$ | 0.54 | $5.51_{\pm1.21}$ | $7.00_{\pm1.41}$ | $11.10_{\pm2.45}$ | $5.27_{\pm0.45}$ | 7.22 |
| FastGPT | $1.33_{\pm0.34}$ | $0.27_{\pm0.10}$ | $0.09_{\pm0.06}$ | $1.65_{\pm0.63}$ | 0.84 | $18.15_{\pm1.44}$ | $11.87_{\pm1.21}$ | $0.44_{\pm0.17}$ | $29.49_{\pm0.70}$ | 14.99 |
| ChatGPT-D | $9.71_{\pm1.11}$ | $9.21_{\pm1.70}$ | $3.13_{\pm0.46}$ | $13.91_{\pm1.82}$ | 8.99 | $25.12_{\pm6.74}$ | $17.21_{\pm5.86}$ | $4.62_{\pm1.11}$ | $44.28_{\pm8.56}$ | 22.81 |
| **ChatGPT-E** | $11.52_{\pm0.78}$ | $11.24_{\pm2.23}$ | $3.43_{\pm0.39}$ | $15.72_{\pm1.18}$ | 10.48 | $27.81_{\pm9.32}$ | $22.52_{\pm9.29}$ | $5.86_{\pm1.76}$ | $47.14_{\pm10.86}$ | 25.83 |
| MPU | $27.38_{\pm1.63}$ | $26.45_{\pm3.30}$ | $7.31_{\pm0.75}$ | $30.42_{\pm2.67}$ | 22.89 | $70.43_{\pm1.63}$ | $79.16_{\pm1.71}$ | $18.22_{\pm1.94}$ | $87.89_{\pm1.23}$ | 63.92 |
| **MPU-E** | $30.40_{\pm2.40}$ | $30.04_{\pm4.29}$ | $7.40_{\pm0.47}$ | $\mathbf{31.55_{\pm3.37}}$ | 24.85 | $75.75_{\pm3.16}$ | $\mathbf{84.23_{\pm4.85}}$ | $19.78_{\pm2.21}$ | $\mathbf{90.31_{\pm1.94}}$ | 67.52 |
| RADAR | $34.38_{\pm1.19}$ | $39.89_{\pm5.10}$ | $10.66_{\pm1.59}$ | $26.98_{\pm1.78}$ | 27.98 | $80.67_{\pm3.10}$ | $83.54_{\pm2.45}$ | $39.18_{\pm5.16}$ | $86.01_{\pm2.61}$ | 72.35 |
| **RADAR-E** | $\mathbf{38.60_{\pm2.51}}$ | $\mathbf{45.32_{\pm5.55}}$ | $\mathbf{11.71_{\pm1.60}}$ | $31.00_{\pm3.13}$ | 31.65 | $\mathbf{82.15_{\pm5.80}}$ | $84.10_{\pm3.44}$ | $\mathbf{44.25_{\pm9.65}}$ | $86.58_{\pm3.14}$ | **74.27** |

Table 2: Performance concerning TPR@FPR-1% (%) on Essay. The detection model is trained on text generated by GPT4All.

| Setting | Method | GPT4All | ChatGPT | ChatGPT-turbo | ChatGLM | Dolly | Claude | Avg. |
|---|---|---|---|---|---|---|---|---|
| Sent. level | Likelihood | $9.18_{\pm1.56}$ | $14.05_{\pm0.57}$ | $5.46_{\pm0.46}$ | $26.04_{\pm4.35}$ | $6.86_{\pm1.13}$ | $2.82_{\pm0.17}$ | 10.73 |
| | Log-Rank | $8.98_{\pm1.11}$ | $13.35_{\pm0.58}$ | $4.97_{\pm0.50}$ | $29.04_{\pm3.44}$ | $6.78_{\pm1.66}$ | $2.32_{\pm0.05}$ | 10.91 |
| | Entropy | $1.36_{\pm0.26}$ | $2.40_{\pm0.55}$ | $1.65_{\pm0.44}$ | $1.10_{\pm0.31}$ | $1.70_{\pm0.43}$ | $1.44_{\pm0.18}$ | 1.61 |
| | NPR | $6.13_{\pm0.66}$ | $7.13_{\pm0.54}$ | $4.04_{\pm0.74}$ | $14.14_{\pm1.85}$ | $4.71_{\pm0.51}$ | $3.26_{\pm0.43}$ | 6.57 |
| | DetectGPT | $4.11_{\pm0.57}$ | $3.57_{\pm0.37}$ | $3.48_{\pm0.33}$ | $5.77_{\pm1.47}$ | $3.70_{\pm0.75}$ | $3.44_{\pm0.17}$ | 4.01 |
| | FastGPT | $12.38_{\pm1.60}$ | $10.30_{\pm0.26}$ | $4.58_{\pm0.68}$ | $28.86_{\pm2.71}$ | $9.01_{\pm0.66}$ | $2.72_{\pm0.43}$ | 11.31 |
| | ChatGPT-D | $15.64_{\pm0.36}$ | $11.50_{\pm0.71}$ | $9.94_{\pm1.11}$ | $23.96_{\pm1.59}$ | $4.66_{\pm0.40}$ | $2.30_{\pm0.47}$ | 11.33 |
| | **ChatGPT-E** | $16.13_{\pm1.12}$ | $12.20_{\pm0.90}$ | $10.49_{\pm0.79}$ | $25.46_{\pm1.70}$ | $4.96_{\pm0.11}$ | $2.40_{\pm0.38}$ | 11.94 |
| | MPU | $18.99_{\pm1.09}$ | $14.38_{\pm1.70}$ | $12.88_{\pm2.02}$ | $30.91_{\pm3.76}$ | $6.00_{\pm1.05}$ | $2.95_{\pm0.39}$ | 14.35 |
| | **MPU-E** | $22.90_{\pm2.44}$ | $17.06_{\pm1.80}$ | $15.30_{\pm2.43}$ | $33.53_{\pm3.07}$ | $7.12_{\pm0.86}$ | $3.58_{\pm0.24}$ | 16.58 |
| | RADAR | $28.85_{\pm2.25}$ | $30.88_{\pm2.17}$ | $32.56_{\pm1.59}$ | $39.99_{\pm2.43}$ | $13.79_{\pm1.04}$ | $\mathbf{11.06_{\pm1.22}}$ | 26.19 |
| | **RADAR-E** | $\mathbf{32.80_{\pm3.24}}$ | $\mathbf{34.33_{\pm2.59}}$ | $\mathbf{36.69_{\pm1.36}}$ | $\mathbf{44.90_{\pm3.37}}$ | $\mathbf{14.78_{\pm2.43}}$ | $10.39_{\pm0.74}$ | **28.98** |
| Para. level | Likelihood | $46.33_{\pm16.49}$ | $68.62_{\pm13.32}$ | $73.60_{\pm14.63}$ | $92.86_{\pm4.84}$ | $20.67_{\pm10.79}$ | $12.36_{\pm6.32}$ | 52.41 |
| | Log-Rank | $63.74_{\pm12.98}$ | $79.47_{\pm7.88}$ | $81.29_{\pm11.25}$ | $96.61_{\pm2.49}$ | $25.92_{\pm9.00}$ | $19.47_{\pm8.24}$ | 61.08 |
| | Entropy | $3.78_{\pm0.91}$ | $11.07_{\pm2.00}$ | $16.31_{\pm1.79}$ | $8.35_{\pm3.44}$ | $3.91_{\pm1.40}$ | $6.22_{\pm1.85}$ | 8.27 |
| | NPR | $78.50_{\pm3.99}$ | $85.91_{\pm1.56}$ | $9.02_{\pm7.27}$ | $95.13_{\pm1.71}$ | $58.52_{\pm7.27}$ | $8.40_{\pm1.17}$ | 55.91 |
| | DetectGPT | $31.53_{\pm6.96}$ | $39.47_{\pm8.30}$ | $7.24_{\pm1.24}$ | $30.31_{\pm7.82}$ | $20.48_{\pm3.80}$ | $5.64_{\pm0.74}$ | 22.45 |
| | FastGPT | $0.18_{\pm0.17}$ | $0.40_{\pm0.26}$ | $68.27_{\pm4.92}$ | $1.70_{\pm0.73}$ | $0.00_{\pm0.00}$ | $7.69_{\pm1.45}$ | 13.04 |
| | ChatGPT-D | $58.13_{\pm3.64}$ | $49.91_{\pm7.73}$ | $34.80_{\pm5.79}$ | $86.74_{\pm3.63}$ | $16.52_{\pm2.30}$ | $2.31_{\pm1.08}$ | 41.40 |
| | **ChatGPT-E** | $59.82_{\pm4.32}$ | $51.24_{\pm7.11}$ | $35.56_{\pm5.50}$ | $85.67_{\pm6.46}$ | $17.09_{\pm2.85}$ | $2.98_{\pm1.10}$ | 42.06 |
| | MPU | $71.07_{\pm7.13}$ | $71.64_{\pm7.01}$ | $48.98_{\pm7.98}$ | $94.78_{\pm2.39}$ | $28.69_{\pm4.95}$ | $7.64_{\pm2.46}$ | 53.80 |
| | **MPU-E** | $78.31_{\pm6.35}$ | $74.09_{\pm7.14}$ | $52.09_{\pm8.18}$ | $96.88_{\pm0.56}$ | $32.08_{\pm5.53}$ | $9.20_{\pm3.09}$ | 57.11 |
| | RADAR | $91.03_{\pm2.80}$ | $84.27_{\pm3.00}$ | $54.22_{\pm7.28}$ | $97.10_{\pm1.73}$ | $48.02_{\pm8.85}$ | $42.40_{\pm4.35}$ | 69.51 |
| | **RADAR-E** | $\mathbf{93.76_{\pm1.88}}$ | $\mathbf{88.53_{\pm2.88}}$ | $\mathbf{59.24_{\pm7.84}}$ | $\mathbf{97.86_{\pm1.48}}$ | $\mathbf{53.41_{\pm8.30}}$ | $\mathbf{55.87_{\pm5.06}}$ | **74.78** |

ChatGPT-turbo, ChatGLM, Dolly, and Claude. The DetectRL dataset includes MGTs from PaLM, ChatGPT, Claude, and Llama-2. Furthermore, the DetectRL dataset contains mixed text, paraphrase attack text, and cross-domain text to simulate and evaluate the detection performance in various complex real-world scenarios. Please refer to Appendix D.1 for detailed descriptions of these datasets and Appendix D.4 for the implementation details of the experiments.

**Baselines**. We selected the following representative baselines, including: (1) metric-based methods: Likelihood [8], Log-Rank [11], Entropy [9], NPR [33], DetectGPT [11], and Fast-DetectGPT (FastGPT) [34]; (2) model-based methods: ChatGPT-D [20], RADAR [19], and MPU [35]. Notably, the proposed enhancement framework can be easily applied to model-based methods, and we denote the enhanced versions using the proposed framework as ChatGPT-E, RADAR-E, and MPU-E. Please refer to Appendix D.2 for detailed information on these baselines.

**Evaluation metric**. Following [36, 37, 38], considering the negative impact on users when HGT is misclassified as MGT, and thus, we typically expect a very low false positive rate (FPR). Therefore, we focus on the true positive rate (TPR) at low TPR. Specifically, we evaluate TPR when FPR is set at 1%, and denote it as TPR@FPR-1%. Besides, we use the area under the receiver operating characteristic curve (AUROC), which is commonly used in MGT detection.

## 4.1 Performance Evaluation

We conduct extensive evaluations in various practical scenarios [39], sentence-level, paragraph-level, mixed, paraphrasing, and cross-domain texts. Their detailed descriptions are in Appendix D.3.

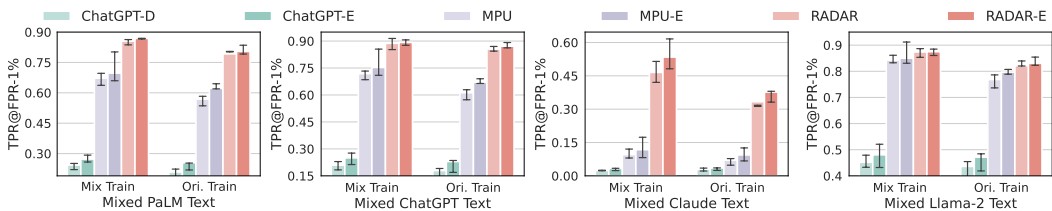

Figure 4: Test performance (TPR@FPR-1%) under various LLM mixed texts. Detectors are trained on text generated by PaLM. For each sub-figure, the left group: detectors are trained on mixed text, and the right group: detectors are trained on original text.

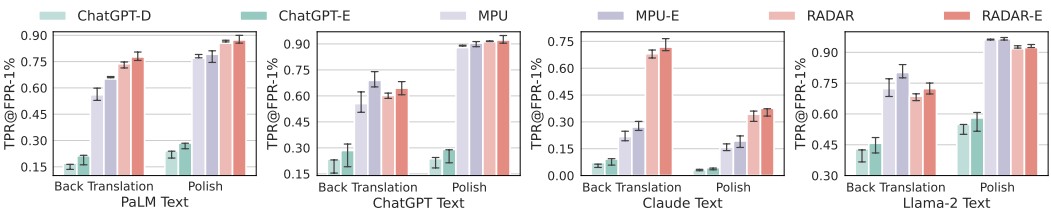

Figure 5: Robustness (TPR@FPR-1%) against paraphrasing attacks (Back Translation and Polish). Detectors are trained on the PaLM texts and tested on the paraphrasing texts of various LLMs.

**Sentence-level Detection**. Tables 1 (left) and 2 (top) present sentence-level detection performance w.r.t. TPR@FPR-1%, for models trained on PaLM and GPT4All texts, respectively. See Tables 6-23 in the Appendix for AUROC and results trained on other LLMs. Overall, the proposed enhancement strategy substantially improves the original models' performance. For example, on the DetectRL dataset, the average TPR@FPR-1% for RADAR increases from 27.98% to 31.65% with the enhanced RADAR-E, while performance rises from 26.19% to 28.98% on the Essay dataset. Furthermore, the results consistently show that model-based methods generally surpass metric-based methods. For instance, FastGPT, the best-performing metric-based method in our experiment, is only comparable to ChatGPT-D on the Essay dataset and notably inferior to model-based methods on the DetectRL dataset. This highlights the significance of our enhancement targeted at model-based approaches. Interestingly, we also observe that cross-LLM detection performance is not always inferior to within-LLM detection. For example, Table 1 shows RADAR performing worse on texts generated by PaLM compared to ChatGPT. We reasonably suspect that this may be related to the quality of the generated text, and the text generated by ChatGPT is more discriminative.

**Paragraph-level Detection**. Tables 1 (right) and 2 (bottom) show the paragraph-level detection performance in terms of TPR@FPR-1%, where detectors are trained on PaLM and GPT4All, respectively. For performance on AUROC and additional results trained on other LLMs, please refer to Tables 6-12 and 24-34 in the Appendix. We observe enhancing effects similar to those at the sentence level, highlighting the proposed method's benefit at this granularity. Moreover, by comparing detection performance at the paragraph level and the sentence level, it is evident that sentence-level detection is significantly more challenging, similar to the finding in [17]. This underscores the significance of sentence-level detection and encourages it to be a primary focus for future studies.

**Mixed Text Detection**. Fig. 4 illustrates the test performance (TPR@FPR-1%) on explicit mixed text, and the corresponding AUROC performance can be found in Fig. 12 in the Appendix. Our experiments were conducted under two settings: training on mixed text and training on original text. Compared to the performance improvements shown in Table 1 (right), the enhancements in detecting mixed text are more pronounced. This may be attributed to the inherently less precise hard labels in mixed text, the core issue our proposed strategy directly addresses, thereby demonstrating greater potential for improvement and highlighting the rationale behind our design. Besides, comparing the detectors trained on mixed text (left group) and original text (right group), it can be found that the former has better robustness in detecting mixed texts, which is akin to adversarial training [40].

**Paraphrasing Text Detection**. Recent studies have shown that MGT detectors are vulnerable to paraphrasing attacks [41, 42], where paraphrased text can bypass detection. To assess the robustness improvement in adversarial scenarios, we explored the detection performance on texts subjected to

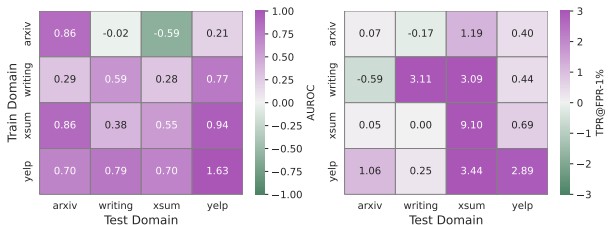

Figure 6: The performance improvement of the enhanced RADAR-E compared to RADAR. The figure shows their differences for AUROC (left) and TPR@FPR-1% (right). Positive values indicate performance improvement.

Table 3: Running time (seconds).

| Paragraph-level | Training Time | | Inference Time | |
|---|---|---|---|---|
| | Essay | DetectRL | Essay | DetectRL |
| Likelihood | 9.3 | 12.1 | 41.5 | 54.2 |
| Log-Rank | 11.2 | 13.8 | 50.3 | 62.0 |
| Entropy | 7.6 | 10.6 | 34.8 | 47.8 |
| NPR | 308.0 | 527.0 | 1384.7 | 2369.2 |
| DetectGPT | 299.2 | 577.3 | 1345.0 | 2595.5 |
| FastGPT | 32.1 | 37.1 | 144.2 | 166.6 |
| ChatGPT-D | 18.2 | 31.7 | 4.7 | 8.4 |
| **ChatGPT-E** | 20.9 | 32.8 | 4.7 | 8.4 |
| MPU | 17.9 | 31.9 | 4.7 | 8.3 |
| **MPU-E** | 19.2 | 33.1 | 4.7 | 8.3 |
| RADAR | 29.8 | 46.1 | 7.9 | 13.3 |
| **RADAR-E** | 32.0 | 48.4 | 7.9 | 13.4 |

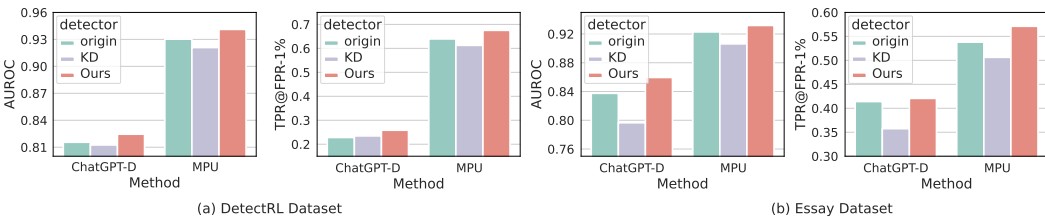

Figure 7: Performance comparison with Knowledge Distillation.

two common paraphrasing attacks: Polish and Back Translation [32], as illustrated in Fig. 5 and Fig. 13 in the Appendix. Notably, detectors employing the proposed enhancement framework exhibited greater robustness in these attack scenarios. Compared to non-attack performance (Table 1, right), performance generally declined under attack. This not only reaffirms the vulnerability of existing MGT detection methods but also underscores the critical need to enhance detector robustness.

**Cross Domain Detection**. Cross-domain performance was evaluated across four high-risk domains: arXiv Archive, Writing Prompts, XSum, and Yelp Reviews. Results are presented in Fig. 6, with additional results in Fig. 14 in the Appendix. In most settings, models employing the proposed strategy demonstrate significantly enhanced generalization capabilities. This is likely attributable to our framework reducing the model's reliance on inexact hard labels, thereby mitigating overfitting.

**Running Time**. Table 3 presents the runtime comparison in paragraph-level settings (sentence-level results are in Appendix Table 36). As discussed in Section 3.3, since the target detector is not modified, the proposed framework introduces no additional inference delay. Regarding training overhead, the longer text for the supervisor is constructed within the batch, and the intermediate results $f(x^{(j)}, \theta_f)$ from the detector can be reused. This ensures negligible overhead during the data preparation phase. The primary training delay is the supervisor's forward and backward passes. However, as the supervisor model (i.e., three-layer fully connected network in the experiments) is significantly simpler than detectors, the overall training delay is negligible.

## 4.2 Comparison with Knowledge Distillation

Fig. 7 presents the performance comparison between the proposed framework and Knowledge Distillation (KD), which is based on the "strong teacher enhances weak student" concept. Here, we selected RADAR, which has the best detection performance in the experiment, as the teacher model to guide ChatGPT-D and MPU. The results indicate that knowledge distillation even led to a decrease in the performance of the student models in most settings. This is primarily because the effectiveness of knowledge distillation largely depends on the quality of the teacher model (i.e., high-quality soft labels). However, obtaining such a teacher model is challenging in complex MGT detection scenarios.

## 4.3 Supervision Quality Assessment

To further verify whether the supervisor guides the detector to learn more accurate "golden" labels for enhancement, we introduce a knowledge distillation-based analysis method for verification. Specifically, we use the detectors before and after enhancement as teacher models, distill a student

Table 4: The impact of teacher model (RADAR) enhancement on Knowledge Distillation.

| Method | DetectRL | | | | Essay | | | | | |
|---|---|---|---|---|---|---|---|---|---|---|
| | PaLM | ChatGPT | Claude | Llama-2 | GPT4All | ChatGPT | ChatGPT-turbo | ChatGLM | Dolly | Claude |
| Origin-KD | 66.01 | 73.28 | 15.77 | 83.66 | 69.34 | 41.29 | **93.97** | 24.92 | 64.00 | 5.47 |
| Ours-KD | **67.49** | **76.00** | **16.81** | **84.77** | **70.11** | **42.53** | 93.97 | **25.68** | **65.47** | **6.00** |

Table 5: Case study in RADAR detector.

| | MGTs that can be detected by both RADAR and RADAR-D | MGTs that can only be detected by RADAR-D |
|---|---|---|
| **Case 1** | When I thought of the comment, I wrote it as Milio's, and I thought'meh. This is just another pizza shop. But after eating there, I realized that Milio was not just another Pizza shop. The food is delicious, the service is very good, and the atmosphere is warm and seductive. ... | The atmosphere is excellent with lots of big screen TV's to watch the game. The service was decent as well. The waitress was very friendly and watchful, but the kitchen seemed backed up because our food took over an hour to come out. ... |
| **Case 2** | The two-dimensional kagome lattice has been identified as a promising platform to realize quantum spin liquid states due to its unique geometry. In this paper, we report the synthesis and characterization of a series of two-dimensional kagome antiferromagnets, ZnxCu4-x(OD)6Cl2, ... | Sussex further improved their hopes of reaching the One-Day Taza knockout stages by beating Surrey for a third consecutive regrouped win. ... |
| **Case 3** | Scientists, intrigued by my resilience, devised a daring plan to send me into the eeigmatic depths of a black hole. Embracing my inner badass, I agreed, embarking on a journey that would test the ilmits of my extraordinary abilities. ... | This place should have scas of glowing reviews! My wife and I had a great tife. It's hard to believe that such a grea restaurant could be in a strip mall. ... |

model respectively. By comparing these student models' performance, we can verify the quality of the supervisory signal. If the student distilled from the enhanced detector outperforms the student distilled from the original detector, it indicates that our approach has indeed enabled the detector to learn better detection scores from the supervisor, thereby improving detection performance. Table 4 presents the distillation performance, where Origin-KD and Ours-KD denote the original RADAR detector and our enhanced RADAR (i.e., RADAR-E) guided MPU detector, respectively. By comparison, we observe that Ours-KD achieves superior performance, proving that the proposed method learns better detection scores from the supervisor to enhance detection.

## 4.4 Case Study

We prepare some examples for quantitative analysis, as shown in Table 5. Specifically, the table shows sequentially selected three MGTs from the DetectRL dataset. The text in the left column is accurately identified as "MGT" by both the baseline version of RADAR and our improved version RADAR-E. The text in the right column represents "win" cases for our approach, that is, they are misclassified by RADAR but correctly detected by RADAR-D. We can find that the easy-to-detect texts (left column) typically exhibit more complex idioms, formal vocabulary, and convoluted expressions, which are typical of machine-generated text. Instead, the difficult-to-detect texts (right column), which our method successfully detects, are characterized by short, plain, straightforward, and somewhat colloquial sentences, which are much closer to a typical human-like distribution. The characteristics of difficult-to-detect text and the effectiveness of the enhancement strategy intuitively demonstrate our advantages in processing ambiguous machine-generated text.

## 5 Conclusion

Existing detection methods are limited by inaccurate labels leading to inexact training, while the limitations of human cognition and the superintelligence of detectors make inexact learning widespread and inevitable. To address this, the paper has proposed an easy-to-hard supervision enhancement framework. This framework utilizes easier supervisors (weaker models trained on simpler long-text detection tasks) to enhance the more complex target detection model. By structurally integrating the detector into the supervisor's design, we model the supervisor as a lower performance bound for the detector. Optimizing the supervisor thus indirectly optimizes the detector, leading to improved alignment with the underlying golden labels. Extensive experiments across diverse practical scenarios demonstrate that this framework significantly improves MGT detection capabilities.

## Acknowledgment

This work was supported by the RGC Senior Research Fellow Scheme under the grant: SRFS2324-2S02, RGC Young Collaborative Research Grant No. C2005-24Y, RGC General Research Fund No. 12200725, and NSFC General Program No. 62376235.

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

# Appendix

# A Further Discussion

## A.1 The solution to Inexact Supervision

In the introduction, we emphasized that this paper aims to learn effectively from the supervisor when (1) the provided label is inexact, and (2) the detection quality is difficult to assess. Therefore, we proposed an easy-to-hard enhancement framework. Here, we systematically discuss how the proposed framework solves them.

To address the first challenge, we design supervisors to focus on the relatively easier task of longer text detection. Theoretically, longer texts have a greater class distribution distance (Theorem 3.1), which helps mitigate the negative impact of inexact labels. Additionally, we collapse the HGT distribution as a Dirac $\delta_0$ distribution, which theoretically increases the distribution distance (Theorem 3.3), further aiding the supervisor's learning.

To address the second challenge, we do not employ the method of supervisors directly providing soft labels to detectors, as is common in knowledge distillation. This is due to the evaluation difficulty and the fact that the supervisor may be weak and cannot guarantee the quality of the provided soft label. Instead, we structurally integrate the detector into the supervisor's design, establishing a connection between the supervisor and the detector, where the supervisor's performance serves as a lower bound for the detector's capabilities. This coupled structure allows the detector to be indirectly optimized via the supervisor, avoiding the larger error caused by directly providing label-based signals. Accordingly, the enhancement roadmap is: maximize supervisor performance $\rightarrow$ maximize $k$ $\rightarrow$ maximize detector performance.

## A.2 Scalability of the Proposed Approach

We examine the scalability of our approach to large-scale scenarios from the perspectives of computational compatibility and framework generality.

- **Computational compatibility**. A key aspect of our framework is its computational efficiency. Incorporating an auxiliary supervisor throughout the training process adds negligible overhead, as shown in Table 3. This supervisor can be designed as a lightweight component that avoids bottlenecks, ensuring seamless integration into standard large-scale training pipelines with large models and massive datasets. In essence, the bias is in the training objective (Eq. 5), not in a restrictive architectural change that would hinder scaling.

- **Framework generality**. The proposed framework applies to model-based approaches that rely on data and computing power rather than metric-based methods centered on specific features. This reflects the philosophy of "The Bitter Lesson"—instead of forcing explanations of the detection mechanism through specific features, we learn to construct the most effective representations for detection using data and computation. Moreover, the enhancement framework is not built on narrowly designed knowledge; rather, it represents a general learning architecture under inexact supervision by using an easier supervisor to refine labels for the target model. Its applicability may extend beyond text detection to any inexact supervision problem characterized by inherently ambiguous labels.

Essentially, our framework employs an efficient computational method (training the supervisor on longer texts) to provide better learning signals for another method (the detector), rather than imposing a rigid, artificially designed bias that limits learning and scalable.

## A.3 Different from Knowledge Distillation

The proposed framework exhibits significant differences from Knowledge Distillation (KD) [27] in several key aspects:

- **Model**. KD is essentially a "strong teacher-weak student" enhancement paradigm, where a typically more powerful and complex teacher model guides a student model to improve performance or achieve model compression. In contrast, the proposed framework follows an easy-to-hard enhancement paradigm, enhancing the target detector through a supervisor that focuses on the relatively easier task of long text detection. Thanks to the relative simplicity of the task, the

supervisor model can often be designed in a more lightweight manner. For example, in the paper, it is modeled as a simple three-layer fully connected network, which significantly reduces complexity compared to detectors usually built upon complex pre-trained models.

- **Task**. In traditional KD, both the teacher and student models are typically trained and evaluated on the same task, with the teacher model providing guidance based on its superior performance in that task. However, in the proposed framework, although the overarching goal for both the supervisor and the detector is MGT detection, the tasks they focus on differ significantly in terms of difficulty and data characteristics. The supervisor deals with longer texts that are relatively easier to detect, while the detector handles the more challenging original texts.

- **Supervision Signal**. The core of KD lies in using the soft labels output by the teacher model as a supervision signal to guide the student model, with the typical optimization goal being the minimization of the divergence between the output distributions of the student and teacher models. This is to enable the student model to learn the decision boundaries and generalization capabilities of the teacher model. Its effectiveness depends on the teacher model providing high-quality soft labels. However, in MGT detection, the detector's superintelligence makes it unclear if the supervisor's soft labels offer superior information. Therefore, the proposed framework does not directly rely on the supervisor's soft labels. Instead, by structurally incorporating the detector into the supervisor, we consider the supervisor's performance as a theoretical lower bound for the detector's capabilities. Thus, optimizing the supervisor indirectly enhances the detector.

### A.4 Challenge of Noisy Label Learning

Noisy label learning (NLL) [29, 28] is primarily designed to address incidental and limited labeling errors, whereas in MGT detection, the issue to be addressed is the widespread inexactness in labels (the categories are correct) due to limitations of human cognition. There are significant differences in both the problem and the techniques involved:

- **Nature of the Problem**. NLL assumes the existence of clear, discrete ground truth, viewing erroneous labels in the training set as occasional random errors, with its core focus being the correction of these errors, emphasizing label accuracy. In contrast, the imprecision in MGT detection stems from the ambiguity of the issue itself— the "machine-generated" attribute of text is not always a clear binary judgment. The focus in MGT detection is on hard labels that do not sufficiently represent the sample, rather than labeling errors.

- **Technical Means**. Classic strategies in NLL, such as robust loss functions [43, 29] and sample selection [44], attempt to identify and prioritize training on "clean" samples while reducing or ignoring the influence of "erroneous" ones. This mechanism typically implies the assumption that noise is limited and identifiable, and the default fact is that when the noise ratio > 50%, it is impossible to learn without additional assumptions [44]. However, in MGT detection, due to the limitations of human cognition in recognizing inherent data ambiguity, these inexact samples are both prevalent and complex. Even when clearly distinguishable samples exist, it is challenging for human cognition to reliably filter them out. This makes it difficult for NLL techniques to adapt to MGT detection.

To further verification, this paper also experimentally evaluates the feasibility of noisy label learning, as shown in Appendix D.8.

### A.5 Reasonableness of the Golden Label Assumption

In MGT detection, traditional binary labels ("human-generated" or "machine-generated") struggle to accurately reflect the degree of "machineness" in text. Considering the limitations of human cognition, perfect, clear golden labels might be impossible to define. To this end, recognizing that the degree of "machineness" largely depends on the explicit mixed text in human-machine collaboration scenarios, as well as the implicit mixed texts when LLMs generate human-like text, we naturally approximate MGT golden labels as the proportion of purely machine-generated content distinguishable from HGT (even if unknown). Compared to simple binary labels, this continuous proportional labeling can more finely and quantitatively capture the true characteristics of texts and their degree of "machineness". For example, a piece of text might be 30% machine-edited, so using this proportion would approximate its golden label to 0.3, which indicates the degree of machine-likeness in this piece of text, information that cannot be represented by a binary label.

Undeniably, the text generation in practical scenarios can be highly complex and unstructured (e.g., clause-level, word-level modification), making the simple use of proportion as an approximation of the golden label imperfect. However, in the current absence of more precise metrics, this proportion-based label provides a relatively unbiased ground truth approximation that better reflects machine contribution. It aids models in understanding the distribution characteristics of data within a continuous label space, similar to Mixup [45]. Future research could further explore more precise quantification methods for defining the degree of "machineness" in texts for MGT detection.

### A.6 Differences from Existing Theoretical Results

Our theoretical results build upon Chakraborty's foundational theory [23] but differ substantially in several aspects:

- **Lower bounds vs. upper bounds**. While existing work [23] establishes critical theoretical lower bounds for detection, we extend to the theoretical upper bounds (Theorem 3.2 and the right-hand side of Theorem 3.1) to further explore the potential of detection. Although they are both related to text length, they measure different bounds.
- **Guidance for the proposed framework**. In addition to exploring the detection potential, it is more important to guide the design of the proposed framework: by coupling the influencing factors of the supervisor's upper bound with the detector, maximizing the supervisor's performance indirectly optimizes the detector, i.e., maximizing the supervisor's performance -> optimizing the influencing factors of the upper bound -> optimizing the detector. Instead, such a guided optimization mechanism cannot be achieved with the theoretical lower bounds.

### A.7 Limitation

The proposed framework in the paper is currently applicable to enhancing model-based detection methods with relatively good performance. One future research direction could explore ways to effectively enhance metric-based MGT detection methods under inexact supervision learning conditions. Moreover, Theorem 3.4 reveals that the proposed framework tends to lead the detector to converge to predefined "golden" labels: the proportion of purely machine-generated content distinguishable from HGT. As discussed in Section A.5, in the absence of more precise metrics, although this approximate gold label provides a relatively less biased ground truth compared to hard labels, it is undeniably imperfect. Future research may focus on the quantification of more precise golden labels and the design of detection algorithms that approximate them.

### A.8 Broader Impact

This paper presents work that aims to advance the field of trustworthy machine learning and large language models, specifically by improving machine-generated text detection. We do not involve human subjects, potentially harmful insights, potential conflicts of interest and sponsorship, discrimination and bias concerns, privacy and security issues, legal compliance, or other ethical issues.

## B Related Work

Existing MGT detection methods can be broadly categorized into three categories: proactive watermarking-based methods, post-hoc metric-based methods and model-based methods.

### B.1 Watermark-based Method

Watermarks are embedded as subtle yet consistent information at the inception of machine-generated text while maintaining expected grammar and semantics, which facilitate traceability by enabling the detection of machine-generated text [46]. Kirchenbauer et al. [47] randomly partition tokens into two categories, biasing the probability distribution to favor one category, resulting in a higher frequency of these tokens in watermarked text. This allows for watermark detection using statistical hypothesis testing. Furthermore, simplifying this approach to a fixed word list has demonstrated greater robustness against paraphrase attacks [48]. Chen et al. [49] evaluated the impact of watermarks on different capabilities of large language models from a cognitive science perspective, finding

that knowledge recall and logical reasoning are more adversely affected than language generation. To address this, they introduced Watermarking with Mutual Exclusion, dynamically optimizing token use during decoding by applying exclusion rules to recognized lexical redundancies. In addition to manually designing watermarks, leveraging the capabilities of language models to directly learn to generate watermarked text is promising. This includes training student models [50] and semantically invariant watermark models [51]. While watermark-based methods exhibit strong detection capabilities, they are limited by the requirement for full access to the generation model, which hinders their practicality in detection tasks, as there is often no prior knowledge of the generation model, let alone the ability to modify its generation rules. Besides, they are also vulnerable to attacks [52, 53], which exacerbates the challenge of detection.

## B.2    Metric-based Methods

Metric-based methods detect based on the statistical differences between generated and natural text. Mitchell et al. [11] found that slight perturbations in generated text can result in rewritten text having a lower log probability than the original text, leading to the design of DetectGPT. Similarly, Solaiman et al. [8] performed detection based on the higher log probability of generated text compared to natural text. Additionally, using relative likelihood instead of absolute likelihood has proven to be more competitive than previous zero-shot methods [54]. Considering the intensive computational cost of DetectGPT, Fast-DetectGPT [12] introduced conditional probability curvature to measure differences in token selection and used sampling to improve efficiency. DNA-GPT [55] divided the text into two parts and used them to generate the other part, performing detection by measuring the difference between the generated and the original. Since LLMs are often black boxes in practice [39], these methods use a proxy model to extract features, leading to performance degradation due to distribution discrepancies between the proxy and target models. To address this, Zeng et al. [56] proposed a distribution-aligned detection framework, ensuring enhanced detection capability and resilience against rapid model iterations with minimal training investment. Nguyen-Son et al. [57] observed that the similarity between original and generated texts is significantly higher than between generated and subsequently regenerated texts. Accordingly, they proposed SimLLM, which detects by estimating the similarity between input sentences and their generated counterparts. Yu et al. [58] captured intrinsic features of text by identifying layers with the greatest distribution differences when projecting onto lexical space, and using intrinsic rather than semantic features for detection has proven to yield better performance. Given that existing methods struggle with out-of-distribution data, token cohesiveness [59] has been shown to be a good indicator, with LLM-generated text often exhibiting higher token cohesiveness than human-written text. Tulchinskii et al. [60] discovered that the intrinsic dimension of text is a good metric, as generated text typically has an average intrinsic dimension about 1.5 lower than natural text. Clearly, metrics-based methods are manually designed features based on limited data, casting doubt on their broader effectiveness.

## B.3    Model-based Methods

Model-based methods do not involve explicit feature extraction; instead, they leverage the powerful representation capabilities of deep learning models to implicitly learn distinguishable features by taking the entire text as input. Energy-based models [61], which perform well in continuous space, have been applied to text sequence detection, demonstrating better adaptability to changes in LLM architectures [14]. SeqXGPT [17] focuses on sentence-level detection, employing a detection framework that uses the probability list from a white-box LLM as detection features. AI-Catcher [18] integrates multilayer perceptrons and convolutional neural networks to learn statistical features and high-level semantic representations for detection. Addressing the challenge of limited information in short text detection, Zhang et al. [62] utilized contextual information to simultaneously predict whether multiple sentences were written by machines or humans. Zhong et al. [15] represented the factual structure of a given document as an entity graph, where adjacent nodes denote sentences with consistent relationships, and employed graph neural networks to learn sentence representations, combining them into the document representation for prediction. Hu et al. [19] proposed RADAR, a robust detector resistant to paraphrasing attacks, by employing adversarial learning between a paraphraser and a detector. Lee et al. [63] observed that LLM-generated text has higher estimated preference alignment compared to human-written text, making it easily detectable by utilizing LLMs trained to mimic human preference distributions. Guo et al. [64] posited that the key to detection lies in distinguishing different authors' writing styles, proposing a detection framework based on

multitask auxiliary and hierarchical contrastive learning. Considering the similarity between short texts and human texts, MPU [35] treated short texts as partially unlabeled and adopted a multiscale positive-unlabeled strategy for training. These model-based methods, when supervised on specific datasets, demonstrate higher effectiveness and robustness [32].

## C Theoretical Analysis

### C.1 Proof of Theorem 3.1

*Theorem.* 3.1 (**Distribution Difference for Longer Text**). *Let $h(s)$ and $m(s)$ be the distributions for human-generated and machine-generated sequences on $s \in \mathcal{S}$, respectively, with the total variation distance $TV(m, h) = \delta > 0$. For the text contains $n$ sequences, let $\alpha \geq 0$ denote the ratio of human-like component incorporated in MGT. For longer text formed by concatenating $k$ independent length-$n$ texts, the total variation distance between their distributions, $TV_{long}$ can be bounded by:*

$$1 - 2exp(-\frac{nk(1-\alpha)^2\delta^2}{2}) \leq \mathrm{TV}_{long} \leq 1 - (1-\delta)^{nk(1-\alpha)}.$$

*Proof.* We will prove the upper and lower bounds of the total variance distance of longer texts $TV_{long}$, respectively.

**Upper bound**. First, we introduce a necessary lemma to help our proof.

**Lemma C.1** (**Coupling Lemma** [65]). *Suppose that $P$ and $Q$ are given. For every coupling $(X, Y)$ of $P$ and $Q$,*

$$\mathrm{TV}(P, Q) = \inf\{Pr(X \neq Y) \mid (X, Y) \text{ is a coupling of } (P, Q)\}.$$

A direct corollary of this lemma is that for any coupling $(X, Y)$ of $(P, Q)$, there is:

$$\mathrm{TV}(P, Q) \leq Pr(X \neq Y).$$

For the longer text containing $k$ texts, each of which contains $n$ sequences, therefore, it is natural that the longer MGT is i.i.d. sampled from the $m^{\times nk}$, where $m^{\times nk} := m \times m \times \ldots \times m$ ($nk$ times) denotes the product distribution. Similarly, longer HGT is i.i.d. sampled from the $h^{\times nk}$. In our setting, HGT with $\alpha$ ratio is mixed into MGT. Therefore, the longer MGT is revisited as $m^{\times(1-\alpha)nk}h^{\times\alpha nk}$.

Based on the Coupling Lemma, to obtain the upper bound, we need to construct a random pair $(X, Y) = ((X_1, \ldots, X_{kn}), (Y_1, \ldots, Y_{kn}))$, so that $X \sim m^{\times(1-\alpha)nk}h^{\times\alpha nk}$ and $Y \sim h^{\times nk}$, and then calculate $Pr(X \neq Y)$. we can construct this coupling by coupling each component $(X_i, Y_i)$ independently.

Specifically, by the Coupling Lemma, there exists a coupling $(U, V)$ of $m$ and $h$ such that $Pr(U \neq V) = \mathrm{TV}(m, h) = \delta$. We can choose such a coupling $(U, V)$. Then, we construct a coupling for $(X, Y)$ as follows:

- For the first $(1-\alpha)nk$ components $(i = 1, \ldots, (1-\alpha)nk)$: Let $(X_i, Y_i)$ be drawn independently and identically from $(U, V)$. Thus, $X_i \sim m$ and $Y_i \sim h$, and $Pr(X_i \neq Y_i) = \delta$.

- For the last $\alpha nk$ components $(i = (1-\alpha)nk + 1, \ldots, nk)$: we need $X_i \sim h$ and $Y_i \sim h$. The simplest coupling is to let $X_i = Y_i$ and draw this common value independently from the distribution $h$. Then, $X_i = Y_i \sim h$ and $Pr(X_i \neq Y_i) = 0$.

Now we calculate the probability of $X \neq Y$ under this coupling. $X \neq Y$ iff there is at least one index $i \in \{1, \ldots, nk\}$ such that $X_i \neq Y_i$.

$$Pr(X \neq Y) = Pr\left(\cup_{i=1}^{nk}\{X_i \neq Y_i\}\right).$$

Using the probability of complementary events:

$$Pr(X \neq Y) = 1 - Pr(X = Y) = 1 - Pr\left(\cap_{i=1}^{nk}\{X_i = Y_i\}\right).$$

Since $(X_i, Y_i)$ pairs are constructed independently, the event $\{X_i = Y_i\}$ is independent for different $i$. therefore:

$$Pr\left(\cap_{i=1}^{nk} \{X_i = Y_i\}\right) = \prod_{i=1}^{nk} Pr\left(X_i = Y_i\right).$$

According to our coupling construction:

- For $i = 1, \ldots, (1-\alpha)nk$ : $Pr\left(X_i = Y_i\right) = 1 - Pr\left(X_i \neq Y_i\right) = 1 - \delta$.

- For $i = (1-\alpha)nk + 1, \ldots, nk$ : $Pr\left(X_i = Y_i\right) = 1 - Pr\left(X_i \neq Y_i\right) = 1 - 0 = 1$.

Therefore,

$$Pr(X = Y) = \left(\prod_{i=1}^{(1-\alpha)nk} (1 - \delta)\right) \cdot \left(\prod_{i=(1-\alpha)nk+1}^{nk} 1\right) = (1 - \delta)^{(1-\alpha)nk}.$$

Therefore, $Pr(X \neq Y) = 1 - (1 - \delta)^{(1-\alpha)nk}$, and accordingly, $\text{TV}_{long} \leq Pr(X \neq Y) = 1 - (1 - \delta)^{(1-\alpha)nk}$. The upper bound is proved.

**Lower bound**. From the definition of total variance distance, we know that there exists a specific measurable subset $A \in \mathcal{S}$ such that

$$Pr\left(s \sim m \in A\right) - Pr\left(s \sim h \in A\right) = \delta.$$

If the probability of a single sample drawn from $h$ falling into set $A$ is $Pr\left(s \sim h \in A\right) = p$, the probability of a single sample drawn from $m$ falling into $A$ is $Pr\left(s \sim m \in A\right) = p + \delta$, where $\delta > 0$.

According to the proof of the upper bound above, the longer MGT can be considered as sampling a set of $(1 - \alpha)nk$ i.i.d. sequences $\{s_i\}_{i=1}^{(1-\alpha)nk}$ from distribution $m$, and sampling a set of $\alpha nk$ i.i.d. sequences $\{s_i\}_{i=(1-\alpha)nk+1}^{nk}$ from distribution $h$. Then, the expected number of sequences belonging to $A$ is $(q + (1 - \alpha)\delta)nk$. Similarly, the longer HGT can be considered as sampling a set of $nk$ i.i.d. sequences $s_{i=1}^{nk}$ from distribution $h$, the expected number of sequences in $A$ is $qnk$. Then we can apply the Chernoff bound to have

$$Pr\left(\text{at least } \left(q + \frac{(1-\alpha)\delta}{2}\right) kn \text{ sequences of MGT are in} A\right) \leq \exp^{-\frac{(1-\alpha)^2\delta^2 kn}{2}},$$

and

$$Pr\left(\text{at most } \left(q + \frac{(1-\alpha)\delta}{2}\right) nk \text{ sequences of HGT are in } A\right) \leq \exp^{-\frac{(1-\alpha))^2\delta^2 nk}{2}}.$$

Now consider the even $E$ where $nk$ sequences containing at least $\left(q + \frac{(1-\alpha)\delta}{2}\right)$ sequences of $A$, then we have

$$\text{TV}_{long} \geq Pr\left(E|\text{Long MGT}\right) - Pr\left(E|\text{Long HGT}\right)$$
$$\geq \left(1 - \exp^{-\frac{(1-\alpha)^2\delta^2 nk}{2}}\right) - \exp^{-\frac{(1-\alpha)^2\delta^2 nk}{2}}$$
$$= 1 - 2\exp^{-\frac{(1-\alpha)^2\delta^2 nk}{2}}.$$

Therefore, the lower bound is proved.

$\square$

## C.2 Proof of Theorem 3.2

*Theorem.* 3.2 (**Detection Power for Longer Text**). *Under the assumption of Theorem 3.1, the supervisor's $AUROC_{supv.}$ satisfies:*

$$\text{AUROC}_{supv.} \leq 1 - \frac{1}{2} \cdot (1 - \delta)^{2nk(1-\alpha)}.$$

*Proof.* Invoking Proposition 1 in existing work [23], we have

$$\text{AUROC}_{supv.} \leq \frac{1}{2} + \text{TV}_{long} - \frac{\text{TV}_{long}^2}{2}. \tag{6}$$

Since the right-hand part is the monotonically increasing function of $\text{TV}_{long}$, combing the upper bound in Theorem 3.1, we can bound

$$\text{AUROC}_{supv.} \leq \frac{1}{2} + \text{TV}_{long} - \frac{\text{TV}_{long}^2}{2}$$

$$\leq \frac{1}{2} + 1 - (1 - \delta)^{(1-\alpha)nk} - \frac{(1 - (1 - \delta)^{(1-\alpha)nk})^2}{2}$$

$$= 1 - \frac{1}{2} \cdot (1 - \delta)^{2(1-\alpha)nk}.$$

The theorem is proved. □

## C.3 Proof the Theorem 3.3

*Theorem.* 3.3(**Distribution Difference after HGT Distribution Collapse**). *Under the assumption of Theorem 3.1 and assuming that $m(0) \to 0$, then if $h(s)$ collapses to a Dirac delta distribution, we have $\lim_{m(0) \to 0} TV(h, m) = 1$.*

*Proof.* The assumption that $h$ converges in distribution to Dirac measure $\delta_0$ implies that for the set $A = \{0\}$, $Pr(s \sim h \in \{0\}) \to \delta_0(\{0\}) = 1$. We are given that $Pr(s \sim m \in \{0\}) \to 0$. Consider the measurable set $A = \{0\}$. The difference in probabilities for this set tends to:

$$|Pr(s \sim h \in \{0\}) - Pr(s \sim m \in \{0\})| \to |1 - 0| = 1.$$

By the definition of the total variation distance as a supremum, $\text{TV}(h, m) \geq | Pr(s \sim h \in \{0\}) - Pr(s \sim m \in \{0\}) |$. Taking the limit, we get $\lim_{m(0) \to 0} \text{TV}(h, m) \geq 1$. Since the total variation distance between any two probability distributions is at most 1 (i.e., $\text{TV}(h, m) \leq 1$), we have $\lim_{m(0) \to 0} \text{TV}(h, m) = 1$. The theorem is proved. □

## C.4 Proof of Theorem 3.4

*Theorem.* 3.4(**The Effectiveness of the Proposed Framework**). Under the assumption of Theorem 3.1, and assuming that the MGT golden label is approximately the proportion of pure machine-generated content distinct from HGT. If the supervisor reaches the best possible one, the detector converges to the underlying golden labels.

*Proof.* According to our approximation of the golden label (i.e., the proportion of pure machine-generated content distinguished from HGT in MGT), the golden label of MGT is $1 - \alpha$. In the proposed framework, we use Gumbel Softmax, whose mathematical expectation maintains the original prediction distribution of the detector. Therefore, under the expected behavior, for longer MGT, suppose that the predicted soft label of the detector is $1 - \alpha'$, then texts with $\alpha'$ proportion are filtered. Here we consider two cases:

- $\alpha' \geq \alpha$. To maximize the supervisor's performance, it is necessary to retain the MGT part as much as possible, that is, the HGT of $\alpha$ proportion and the MGT with $\alpha' - \alpha$ proportion are filtered. After filtered, the longer MGT can be considered as sampling a set of $(1 - \alpha')nk$ i.i.d. sequences

$\{s_i\}_{i=1}^{(1-\alpha')nk}$ from distribution $m$. For longer HGT, since it collapses to Direc $\delta_0$ distribution, it can be considered as any length, which is also set as $(1-\alpha')nk$ i.i.d. sequences from distribution $h$. Similar to the proof of Theorem 3.1, we have the total variance distance between longer MGT and HGT is $\text{TV}_{long} = \text{TV}(m^{(1-\alpha')nk}, h^{(1-\alpha')nk}) \leq 1 - (1-\delta)^{(1-\alpha')nk}$. Furthermore, similar to the proof of Theorem 3.2, the supervisor's performance satisfies:

$$\text{AUROC}_{supv.} \leq 1 - \frac{1}{2} \cdot (1-\delta)^{2nk(1-\alpha')}.$$

When the supervisor achieves the best possible one, $\alpha'$ should be minimum, i.e., $\alpha' = \alpha$.

- $\alpha' \leq \alpha$. Similarly, to maximize supervisor's performance, the original HGT of $\alpha'$ proportion is filtered, and the MGT retains. Therefore, the total variance distance between longer MGT and HGT is $\text{TV}_{long} = \text{TV}(m^{\times(1-\alpha)nk}h^{\times(\alpha-\alpha')nk}, h^{\times(1-\alpha)nk}h^{\times(\alpha-\alpha')nk}) \leq \text{TV}(m^{\times(1-\alpha)nk}, h^{(1-\alpha)nk})$, and the best supervisor also is $\text{AUROC}_{supv.} = 1 - \frac{1}{2} \cdot (1-\delta)^{2nk(1-\alpha)}$ when $\alpha' = \alpha$.

In summary, when the supervisor is optimal, the detector's prediction probability for MGT is $1 - \alpha$. The theorem is proved. □

## D Additional Experiments

### D.1 Datasets

A detailed description of the datasets used in the paper is as follows:

- **Essay** [16]. The essay dataset comprises 1,000 text samples. The HGT samples are the original essays from IvyPanda, which spanned numerous subjects and educational levels (high school to university). To create the MGT samples, it first employed ChatGPT-turbo to generate a specific prompt designed to align with the content of each original essay. This prompt was then fed into various LLMs, including GPT4All, ChatGPT, ChatGPT-turbo, ChatGLM, Dolly, and Claude, which produced machine-generated essays in response. This generation strategy allowed for the generation of a diverse corpus of LLM-generated essays linked to the topics of the initial source documents.

- **DetectRL** [32]. It comprises human-written samples from four sources: arXiv academic abstracts (2002-2017), XSum news, Writing Prompts stories, and Yelp Reviews. These domains were specifically chosen because they are considered particularly susceptible to generating deceptive text when LLMs are misused. For each source dataset, it chooses 2,800 human-written samples as HGTs. For MGTs, it selects four LLMs that widely used in the real world, including GPT-3.5-turbo (ChatGPT), PaLM-2-bison (PaLM), Claude-instant (Claude), and Llama-2-70b (Llama-2), to generate machine texts. In addition, the dataset includes various practical attack scenarios. The first is the paraphrase attack, which uses the Dipper paraphraser [41] and Google Translate's Back-translation to rewrite the generated MGT. The second is the mixed text, which randomly replaces a quarter of the original machine-generated text with human-written text, but its label remains in the "machine-generated" category.

### D.2 Baselines

To verify the effectiveness of the proposed strategy, we compare it with metric-based methods such as Likelihood, Log-Rank, Entropy, NPR, DetectGPT, and Fast-DetectGPT, as well as model-based methods such as ChatGPT-D, MPU, and RADAR.

- **Likelihood** [8]. It employs LLM to quantify the log probability at the token level. To derive a detection score for a given text, these individual token log probabilities are then averaged. Notably, a higher score suggests that the text is more likely to have been machine-generated.

- **Log-Rank** [11]. It assigns a score to a text by averaging values derived from the log rank of each token. Specifically, for each token, based on the context that precedes it, a language model provides a rank for that word among its possible predictions. The logarithm of this predicted rank is then taken. The text's score is the mean of these log-rank values across all words. It should be noted that a lower score indicates a higher probability of the text being machine-generated.

- **Entropy** [9]. Similar to the Log-Rank score, the Entropy score for a text is the average of the conditional entropy for each token, given its prior context. It is worth noting that machine-generated texts are likely to have a lower Entropy score.

- **DetectGPT** [11]. It assesses a text by quantifying how minor perturbations affect its log probability under the LLM. The core intuition posits that text generated by an LLM typically resides near local optima within the model's log probability landscape. Consequently, applying small alterations to model-generated text tends to result in a reduced log probability according to the model, compared to the original text. Conversely, applying similar minor perturbations to human-written text does not consistently lead to a decrease and may result in the log probability being either higher or lower than that of the original text.

- **NPR** [33]. Similar to DetectGPT, the Normalized Perturbed Log-Rank (NPR) method also applies perturbations to the original text. The rationale behind NPR is that both MGTs and HGTs are susceptible to minor disturbances, which is indicated by an increase in their Log-Rank score after such perturbations. However, this effect is notably more pronounced in MGTs.

- **Fast-DetectGPT (FastGPT)** [34]. It reveals the limitation of DetectGPT's intensive computational cost, uses the conditional probability curvature metric to identify the difference in token selection between LLMs and humans in a given environment, and proposes to use a more efficient sampling step instead of the perturbation step of DetectGPT.

- **ChatGPT-D** [20]. It is built upon a RoBERTa model that was fine-tuned using the HC3 dataset, and solely utilizes the pure answered text from the dataset.

- **MPU** [35]. The Multiscale Positive-Unlabeled (MPU) training framework attempts to solve the difficulty of short text detection without sacrificing long text. First, it considers the similarity between short machine text and human text, regards this part of short text as partially "unlabeled", and reformulates AI text detection as a partially positive unlabeled (PU) problem. Then, in this PU context, it uses a length-sensitive multi-scale PU loss to train the detection model.

- **RADAR** [19]. It trains a robust MGT detector through an adversarial learning strategy. Specifically, it includes a paraphraser and a detector, where the paraphraser aims to generate real content to evade AI text detection, while the detector tries to detect this part of the content. Both sides improve the robustness of the model in this adversarial training environment.

The implementation of these baseline methods is mainly based on the MGT detection benchmark [66], while Fast-DetectGPT is based on the DetectRL benchmark [32]. In order to compare them fairly with our enhancement methods, we also fine-tune these model-based methods on the given dataset. The learning rate, training epochs and other parameter settings of the enhancement strategy are consistent with them, as shown in Appendix D.4.

### D.3 Evaluation Scenario

To comprehensively evaluate the effectiveness of the proposed framework, we conduct experiments in the following practical scenarios:

- **Cross-LLM**. We trained the detector on texts generated by one LLM and tested it on texts generated by various LLMs to evaluate its cross-LLM generalization performance. The main text presents our results of training on the DetectRL dataset based on PaLM and testing on other LLMs (see Table 1), as well as the results of training on the Essay dataset based on GPT4All (see Tables 2). Additionally, we provide the complete results of training and testing on texts generated by all LLMs in Tables 6-34 in the Appendix. These experiments cover both sentence-level and paragraph-level granularity. For sentence-level experiments, we selected sentences with a token number of at least 5 as valid samples. For paragraph-level experiments, we used the original text without any modifications.

- **Cross-Domain**. The DetectRL dataset comprises four different domains: arXiv academic abstracts, XSum news articles, Writing Prompts stories, and Yelp Reviews. We use this dataset to evaluate the model's cross-domain performance by training the detector on texts from a source domain and testing it on texts from the remaining target domains. In this cross-domain evaluation setup, all MGTs are generated using the default PaLM model. Fig. 6 and Fig. 14 present the heatmap that visually illustrates the relative performance improvements of the proposed framework when applied to various baseline detection models.

- **Paraphrase Attack**. It involves paraphrasing text to preserve its original meaning. We conduct experiments on the DetectRL dataset, which contains paraphrased data from Polish and Back Translation. As in the existing adversarial attack setting, we train on clean text and evaluate its robustness on paraphrased text. Specifically, the detector is trained on the clean PaLM texts and tested on the paraphrasing texts of various LLMs, with results shown in Fig. 5 and Fig. 13.

- **Mixed Text**. In the real world, text is often not purely generated by humans or machines independently, which makes mixed text very common in practice. The DetectRL dataset contains mixed text by replacing one quarter of the sentences in an LLM-generated text with human-written text at random. In this scenario, we perform two settings: (1) The detector was trained on original (non-mixed) text and tested on mixed text; and (2) The detector was trained and tested on mixed text. Fig. 4 and Fig. 12 show the performance under mixed settings, where for each sub-figure, the left group denotes detectors trained on mixed texts, and the right group denotes detectors trained on original texts.

### D.4 Implementation Details

**Supervisor Implementation**. As a supervisor providing reliable guidance to the detector, we model it as a three-layer fully connected neural network in this work. Although it appears simple, this choice is made after in-depth consideration of many aspects:

- **Performance**. Firstly, providing reliable supervision signals depends on the supervisor having satisfactory performance itself. Based on this consideration, there is a tendency to select models with higher expressive capacity as the supervisor. However, the core idea of the proposed enhancement framework is to improve the quality of input data by maximizing the supervisor's performance, thereby promoting the improvement of the detector. If the supervisor's model is over-parameterized, it may prioritize adjusting its parameters to directly fit imperfect data, rather than effectively transmitting signals to the detector to improve data. This may weaken the enhancement effect. For example, if the number of original text segments that constitute a longer text is 4 (i.e., $k = 4$), and the average accuracy of the current detector for these text segments is 50%, this means that the supervisor's long text has only 2 text segments on average, which is defective. In this case, an over-parameterized supervisor model may directly fit this defective data through parameter memory, rather than improving this defective data through optimization (i.e., approaching the length of 4). Based on this consideration, the supervisor tends to be a simple model, which is inconsistent with the above considerations. Thus, as a compromise, we adopt a three-layer fully connected neural network. On the one hand, benefiting from the proposed two data quality enhancements (Theorem 3.1 and Theorem 3.3), even a three-layer network can achieve satisfactory performance, as shown in Appendix D.9.1. On the other hand, the design of this simple model prevents the weakening of enhancement effects on the detector typically caused by over-parameterized models.

- **Efficiency**. As an enhancement framework, minimizing the introduced training delay is preferable. As discussed in Section 3.3, the longer text for the supervisor is constructed within the batch, and the intermediate results $f(x^{(j)}, \theta_f)$ from the detector can be reused. This ensures negligible overhead during the data preparation phase. The primary training delay is the supervisor model's forward and backward passes. This implies that simpler models can achieve reduced training delays. Therefore, we chose a three-layer fully connected neural network, whose training delay is almost negligible, as shown in Table 3 and Table 36.

In summary, using a three-layer fully connected neural network as the supervisor is a comprehensive consideration of performance and efficiency. In the specific implementation of the supervisor, the size of the three hidden layers are 256, 64, and 2, respectively. The input of the supervisor needs to convert the text into the embedding, which is obtained by the tokenizer of the detector and using the embedding layer of the detector. Let $e(x)$ represent the embedding of text $x$, then Eq. 3 can be rewritten as:

$$e(x'') = \oplus_{j=1}^{k} \left( e(x)^{(j)} \odot \mathrm{Gumbel}(f(e(x)^{(j)}, \theta_f)) \right).$$

Here, the text slicing operator $\oplus$ is a vector concatenation operation.

**Hyperparameter Setting**. We fixed five different random seeds (1-5) and conducted five independent repeat experiments. For the DetectRL and Essay datasets, we randomly selected 10% of the data as the training set, with the remaining 90% evenly divided into validation and test sets. To ensure a

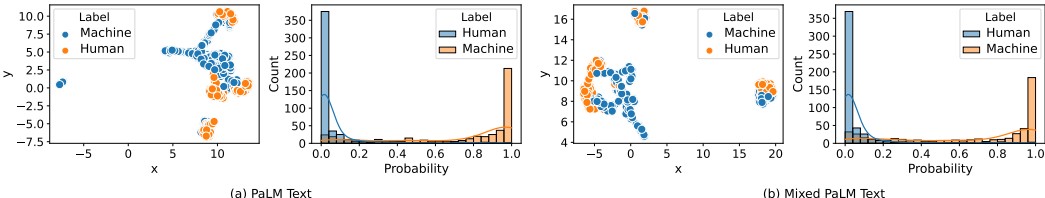

(a) PaLM Text        (b) Mixed PaLM Text

Figure 8: Boundary fuzziness evaluation between MGT (PaLM) and HGT, which illustrates the latent space distribution and prediction confidence distribution under pure (Sub-Fig. 1 & 2) and mixed texts (Sub-Fig. 3 & 4). The mixed text is to replace 1/4 of MGT with HGT.

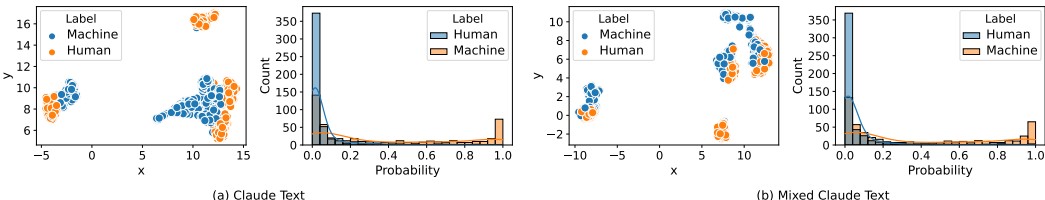

(a) Claude Text        (b) Mixed Claude Text

Figure 9: Boundary fuzziness evaluation between MGT (Claude) and HGT, which illustrates the latent space distribution and prediction confidence distribution of pure texts (Sub-Figure 1 & 2) and mixed texts (Sub-Figure 3 & 4). The mixed text is to replace 1/4 of MGT with HGT.

fair comparison with baseline models, the enhanced models used the same hyperparameters as their corresponding base models. Specifically, all models were fine-tuned for 5 epochs, with a batch size set to 32. Regarding learning rates, we set 5e-6 for relatively smaller models like ChatGPT-D and MPU. For the larger RADAR model, we found that a learning rate of 5e-6 led to unstable training, so a smaller learning rate of 1e-6 was chosen. For supervisor-related hyperparameters, the default settings are as follows: the number of texts in longer texts ($k = 3$), the number of longer texts per batch ($N' = 128$), and the weight ($\lambda = 10$). For performance analysis under more hyperparameter settings, please refer to Appendix D.9.

### D.5 More Results of Boundary Fuzziness

Continuing the discussion on the blurred boundaries between MGT and HGT mentioned in the introduction, Fig. 8, Fig. 9, and Fig. 10 further show the boundary fuzziness evaluation results on more LLM texts. To enhance the persuasiveness of the visualization results, the analyses characterizing the latent space distribution and prediction confidence distribution are based on the RADAR [19], which performed best in our experiments. The analysis results across various LLMs consistently indicate that there is a general blurriness in the boundary between MGT and HGT.

Continuing the discussion in the introduction about the inexactness of hard-label-based training, Fig. 11 further presents experimental results on mixed texts of additional LLMs. It can be observed that in most settings, even the simple application of label smoothing can improve detection performance. This result also indicates that traditional hard-label-based learning may be inexact.

### D.6 More Performance Comparisons

**Sentence-level Detection**. Continuing the sentence-level detection setting discussed in Section 4.1, Tables 6-12 (left) and Tables 13-23 provide a detailed comparison of the performance of various detectors trained on texts generated by different LLMs in the DetectRL and Essay datasets, respectively. From these tables, it can be observed that the experimental results are consistent with the main conclusions in the main text. Notably, in a total of 312 cross-LLM detection settings, the proposed enhancement strategy outperformed the corresponding baseline models in 87% of the cases. This widespread and consistent improvement highlights the general applicability and practical value of the proposed enhancement framework.

**Paragraph-level Detection**. Continuing the paragraph-level detection setting in Section 4.1, Table 6-12 (right) shows the performance comparison of detectors trained with various LLMs in the DetectRL

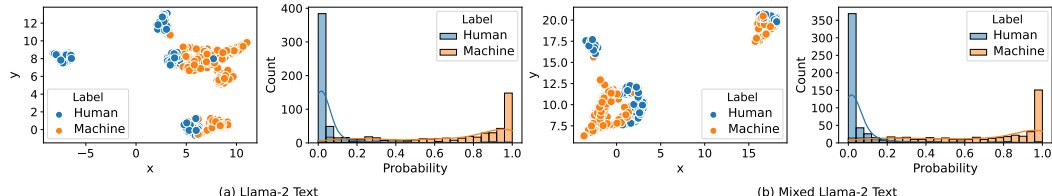

(a) Llama-2 Text         (b) Mixed Llama-2 Text

Figure 10: Boundary fuzziness evaluation between MGT (Llama-2) and HGT, which illustrates the latent space distribution and prediction confidence distribution of pure texts (Sub-Figure 1 & 2) and mixed texts (Sub-Figure 3 & 4). The mixed text is to replace 1/4 of MGT with HGT.

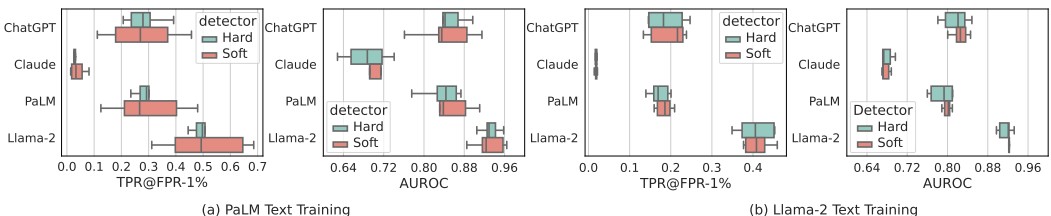

(a) PaLM Text Training         (b) Llama-2 Text Training

Figure 11: Performance comparison with and without using soft labels in mixed text (1/4 of MGT was replaced with HGT). The detector is ChatGPT-D [20].

dataset, while Table 24-34 shows the performance comparison in the Essay dataset. We can also get the same conclusion as the main text. In addition, similar to the sentence-level setting, under the 312 cross-LLM settings, the proposed enhancement strategy outperforms the basic model in 85% of the settings. This extensive enhancement effect is very valuable.

**Mixed Text Detection**. Continuing the mixed detection settings from Section 4.1, Fig. 12 presents a performance comparison of various methods on mixed texts based on the AUROC metric. Similarly, compared to the performance improvement on original texts (see the right part of Table 6), the proposed method demonstrates a more significant enhancement in detecting mixed texts, highlighting the rationality of our design.

**Paraphrasing Text Detection**. Section 4.1 has demonstrated robustness enhancement against paraphrase attacks in terms of TPR@FPR-1%. Here we supplement with Fig. 13, showing AUROC performance under the same attack settings. Similarly, the AUROC metric also indicates that the proposed strategy enhances the robustness of the original detection models.

**Cross Domain Detection**. In addition to the cross-domain performance on RADAR shown in Section 4.1, Fig. 14 illustrates the proposed strategy's enhancement of cross-domain performance for ChatGPT-D and MPU. Similar findings can be observed, where the enhanced versions exhibit better cross-domain generalization capabilities in most settings.

**Test on Newer LLMs**. Based on the Essay prompts, we employed the latest LLMs—GPT-4o, GPT-4.1, DeepSeek-R1, and Llama4 Maverick—to generate machine text. The detection results are shown in Table 35, where detectors are trained on GPT4All texts from the Essay dataset. As can be seen, our proposed enhancement strategy remains significantly effective on these newer LLMs. Moreover, compared to the LLMs used in Table 2, the detection performance on these more advanced LLMs has declined, underscoring the ongoing arms race between detection and large-model development: detectors design stronger strategies based on current generation models, and then newly emerging LLMs produce higher-quality text that is harder to distinguish, thus driving further progress on both sides.

**Running Time**. Table 36 presents a comparison of the runtime (including inference and training time) of various detection models under the sentence-level setting. Similar to the paragraph-level setting, the additional training delay introduced by the enhancement strategy at the sentence level is minimal. This is primarily due to the lightweight design of the supervisor model and the method of constructing long text data within batches during training.

Table 6: Performance concerning AUROC (%) on DetectRL. The detection model is trained on text generated by PaLM.

| Method | Sentence-level | | | | | Paragraph-level | | | | |
|---|---|---|---|---|---|---|---|---|---|---|
| | PaLM | ChatGPT | Claude | Llama-2 | Avg. | PaLM | ChatGPT | Claude | Llama-2 | Avg. |
| Likelihood | $59.72_{\pm 0.80}$ | $58.04_{\pm 0.58}$ | $45.02_{\pm 0.20}$ | $67.82_{\pm 0.55}$ | 57.65 | $71.42_{\pm 0.49}$ | $66.61_{\pm 0.99}$ | $42.47_{\pm 1.27}$ | $78.58_{\pm 0.41}$ | 64.77 |
| Log-Rank | $59.06_{\pm 0.88}$ | $55.92_{\pm 0.62}$ | $44.28_{\pm 0.21}$ | $67.71_{\pm 0.54}$ | 56.74 | $71.64_{\pm 0.49}$ | $65.99_{\pm 0.96}$ | $42.11_{\pm 1.31}$ | $79.96_{\pm 0.45}$ | 64.93 |
| Entropy | $49.43_{\pm 0.38}$ | $48.73_{\pm 5.78}$ | $49.37_{\pm 2.49}$ | $48.68_{\pm 6.15}$ | 49.05 | $60.73_{\pm 0.91}$ | $63.08_{\pm 1.05}$ | $51.73_{\pm 0.96}$ | $66.21_{\pm 0.88}$ | 60.44 |
| NPR | $53.41_{\pm 0.69}$ | $53.59_{\pm 0.20}$ | $47.60_{\pm 0.41}$ | $57.86_{\pm 0.35}$ | 53.11 | $51.30_{\pm 1.56}$ | $52.81_{\pm 3.38}$ | $42.88_{\pm 9.10}$ | $56.86_{\pm 9.88}$ | 50.96 |
| DetectGPT | $53.06_{\pm 0.64}$ | $54.31_{\pm 0.29}$ | $47.35_{\pm 0.37}$ | $56.93_{\pm 0.36}$ | 52.91 | $57.92_{\pm 0.72}$ | $50.04_{\pm 0.98}$ | $65.37_{\pm 0.59}$ | $49.21_{\pm 0.55}$ | 55.63 |
| FastGPT | $62.43_{\pm 0.48}$ | $55.88_{\pm 0.58}$ | $40.81_{\pm 0.31}$ | $67.12_{\pm 0.37}$ | 56.56 | $59.94_{\pm 0.93}$ | $61.26_{\pm 0.93}$ | $24.14_{\pm 1.06}$ | $70.39_{\pm 0.44}$ | 53.93 |
| ChatGPT-D | $73.20_{\pm 2.02}$ | $74.14_{\pm 2.61}$ | $66.72_{\pm 1.73}$ | $78.18_{\pm 1.26}$ | 73.06 | $82.61_{\pm 2.30}$ | $82.23_{\pm 2.49}$ | $68.93_{\pm 1.89}$ | $92.43_{\pm 1.61}$ | 81.55 |
| **ChatGPT-E** | $75.33_{\pm 1.41}$ | $77.12_{\pm 2.21}$ | $68.33_{\pm 0.83}$ | $79.56_{\pm 1.22}$ | 75.09 | $83.98_{\pm 3.07}$ | $83.50_{\pm 3.88}$ | $69.81_{\pm 2.81}$ | $92.50_{\pm 2.36}$ | 82.45 |
| MPU | $87.61_{\pm 0.59}$ | $90.66_{\pm 0.62}$ | $75.96_{\pm 1.12}$ | $88.33_{\pm 0.58}$ | 85.64 | $97.11_{\pm 0.22}$ | $98.07_{\pm 0.38}$ | $77.91_{\pm 0.58}$ | $98.98_{\pm 0.09}$ | 93.02 |
| **MPU-E** | $89.54_{\pm 0.55}$ | $92.18_{\pm 0.53}$ | $76.96_{\pm 0.88}$ | $\mathbf{89.43_{\pm 0.38}}$ | 87.03 | $97.80_{\pm 0.09}$ | $98.76_{\pm 0.34}$ | $80.62_{\pm 0.66}$ | $\mathbf{99.27_{\pm 0.14}}$ | 94.11 |
| RADAR | $89.69_{\pm 0.26}$ | $92.78_{\pm 0.40}$ | $76.29_{\pm 0.82}$ | $87.14_{\pm 0.48}$ | 86.48 | $98.14_{\pm 0.40}$ | $98.87_{\pm 0.29}$ | $90.37_{\pm 0.72}$ | $98.99_{\pm 0.19}$ | 96.59 |
| **RADAR-E** | $\mathbf{90.74_{\pm 0.57}}$ | $\mathbf{93.74_{\pm 0.65}}$ | $\mathbf{77.65_{\pm 1.07}}$ | $88.14_{\pm 0.92}$ | **87.57** | $\mathbf{98.29_{\pm 0.50}}$ | $\mathbf{99.04_{\pm 0.27}}$ | $\mathbf{91.68_{\pm 1.15}}$ | $99.13_{\pm 0.17}$ | **97.03** |

Table 7: Performance concerning AUROC (%) on DetectRL. The detection model is trained on text generated by ChatGPT.

| Method | Sentence-level | | | | | Paragraph-level | | | | |
|---|---|---|---|---|---|---|---|---|---|---|
| | PaLM | ChatGPT | Claude | Llama-2 | Avg. | PaLM | ChatGPT | Claude | Llama-2 | Avg. |
| Likelihood | $59.72_{\pm 0.80}$ | $58.04_{\pm 0.58}$ | $45.02_{\pm 0.20}$ | $67.82_{\pm 0.55}$ | 57.65 | $71.42_{\pm 0.49}$ | $66.61_{\pm 0.99}$ | $42.47_{\pm 1.27}$ | $78.58_{\pm 0.41}$ | 64.77 |
| Log-Rank | $59.06_{\pm 0.88}$ | $55.92_{\pm 0.62}$ | $44.28_{\pm 0.21}$ | $67.71_{\pm 0.54}$ | 56.74 | $71.64_{\pm 0.49}$ | $65.99_{\pm 0.96}$ | $42.11_{\pm 1.31}$ | $79.96_{\pm 0.45}$ | 64.93 |
| Entropy | $50.36_{\pm 0.58}$ | $55.89_{\pm 0.50}$ | $52.56_{\pm 0.15}$ | $56.28_{\pm 0.44}$ | 53.77 | $60.73_{\pm 0.91}$ | $63.08_{\pm 1.05}$ | $51.73_{\pm 0.96}$ | $66.21_{\pm 0.88}$ | 60.44 |
| NPR | $53.41_{\pm 0.69}$ | $53.59_{\pm 0.20}$ | $47.60_{\pm 0.41}$ | $57.86_{\pm 0.35}$ | 53.11 | $51.69_{\pm 1.13}$ | $54.31_{\pm 0.87}$ | $38.48_{\pm 0.87}$ | $62.01_{\pm 0.73}$ | 51.62 |
| DetectGPT | $53.06_{\pm 0.64}$ | $54.31_{\pm 0.29}$ | $47.35_{\pm 0.37}$ | $56.93_{\pm 0.36}$ | 52.91 | $48.23_{\pm 7.75}$ | $49.58_{\pm 0.89}$ | $47.16_{\pm 15.11}$ | $49.76_{\pm 0.93}$ | 48.68 |
| FastGPT | $62.43_{\pm 0.48}$ | $55.88_{\pm 0.58}$ | $40.81_{\pm 0.31}$ | $67.12_{\pm 0.37}$ | 56.56 | $59.94_{\pm 0.93}$ | $61.26_{\pm 0.93}$ | $24.14_{\pm 1.06}$ | $70.39_{\pm 0.44}$ | 53.93 |
| ChatGPT-D | $72.38_{\pm 1.74}$ | $74.98_{\pm 2.67}$ | $67.09_{\pm 1.63}$ | $77.83_{\pm 1.94}$ | 73.07 | $79.36_{\pm 2.04}$ | $84.04_{\pm 4.39}$ | $69.50_{\pm 3.29}$ | $91.62_{\pm 2.16}$ | 81.13 |
| **ChatGPT-E** | $73.64_{\pm 0.95}$ | $76.91_{\pm 1.82}$ | $68.17_{\pm 1.45}$ | $78.91_{\pm 0.93}$ | 74.41 | $80.03_{\pm 1.64}$ | $88.09_{\pm 3.32}$ | $70.52_{\pm 2.49}$ | $93.13_{\pm 1.84}$ | 82.94 |
| MPU | $84.21_{\pm 0.46}$ | $94.25_{\pm 0.10}$ | $78.88_{\pm 0.90}$ | $89.11_{\pm 0.46}$ | 86.61 | $94.22_{\pm 0.29}$ | $99.14_{\pm 0.19}$ | $79.14_{\pm 1.02}$ | $99.15_{\pm 0.13}$ | 92.91 |
| **MPU-E** | $85.24_{\pm 0.63}$ | $95.15_{\pm 0.17}$ | $\mathbf{79.54_{\pm 0.83}}$ | $\mathbf{89.94_{\pm 0.35}}$ | **87.47** | $95.78_{\pm 0.35}$ | $99.58_{\pm 0.14}$ | $82.66_{\pm 1.06}$ | $\mathbf{99.42_{\pm 0.08}}$ | 94.36 |
| RADAR | $85.00_{\pm 0.43}$ | $95.05_{\pm 0.16}$ | $77.61_{\pm 0.83}$ | $87.48_{\pm 0.54}$ | 86.28 | $97.22_{\pm 0.41}$ | $99.56_{\pm 0.11}$ | $86.43_{\pm 0.79}$ | $99.29_{\pm 0.07}$ | 95.62 |
| **RADAR-E** | $\mathbf{85.91_{\pm 0.47}}$ | $\mathbf{95.63_{\pm 0.09}}$ | $78.28_{\pm 0.95}$ | $88.28_{\pm 0.44}$ | 87.03 | $\mathbf{97.34_{\pm 0.34}}$ | $\mathbf{99.65_{\pm 0.14}}$ | $\mathbf{86.87_{\pm 0.36}}$ | $99.36_{\pm 0.13}$ | **95.80** |

## D.7 Performance under Noisy Labels

Our framework assumes that hard labels are correct by default. To this end, this section verifies the detection performance in the presence of noisy labels. We randomly flipped 10% of the labels and then evaluated detectors trained on these noisy data, as shown in Fig. 15. First, we found that the proposed enhanced framework remained effective. Second, compared to the noiseless results in Table 1, our method is generally less affected by noise, verifying our mitigation efforts.

## D.8 Comparison with Noisy Label Learning

In addition to discussing the challenges of noisy label learning (NLL) techniques in Appendix A.4, we also evaluated them experimentally. We selected two typical NLL methods: Co-teaching [44] and SAM [29], applying them to the ChatGPT-D baseline model for evaluation, denoted as ChatGPT-Co and ChatGPT-SAM, respectively. The performance comparison is shown in Fig. 16. It can be observed that the direct application of these NLL techniques not only failed to improve performance but, in most cases, even led to a decline in detector performance. This is mainly because the core objective of NLL techniques is to identify and correct limited erroneous labels in training data. However, in the context of MGT detection, the labels themselves are correct; the issue lies in widespread labeling inexactness. Therefore, NLL strategies designed based on the assumption of erroneous labels do not align with the nature of imperfect labels (ambiguity) in MGT detection, making them difficult to effectively apply to tasks aimed at enhancing MGT detection.

## D.9 Sensitivity Analysis

### D.9.1 About the Supervisor

**Sensitivity w.r.t. Original Text Number $k$ for Longer Text**. A core idea in the design of the supervisor is to use longer texts to enhance data quality. Theorem 3.1 theoretically demonstrates that

Table 8: Performance concerning TPR@FPR-1% (%) on DetectRL. The detection model is trained on text generated by ChatGPT.

| Method | Sentence-level | | | | | Paragraph-level | | | | |
|---|---|---|---|---|---|---|---|---|---|---|
| | PaLM | ChatGPT | Claude | Llama-2 | Avg. | PaLM | ChatGPT | Claude | Llama-2 | Avg. |
| Likelihood | $4.83_{\pm0.39}$ | $1.58_{\pm0.23}$ | $0.72_{\pm0.13}$ | $5.54_{\pm0.40}$ | 3.17 | $25.66_{\pm2.41}$ | $10.21_{\pm1.40}$ | $1.78_{\pm0.38}$ | $38.39_{\pm0.92}$ | 19.01 |
| Log-Rank | $4.84_{\pm0.41}$ | $1.23_{\pm0.25}$ | $0.72_{\pm0.13}$ | $5.25_{\pm0.87}$ | 3.01 | $27.49_{\pm1.13}$ | $11.55_{\pm1.93}$ | $2.08_{\pm0.65}$ | $41.93_{\pm0.47}$ | 20.76 |
| Entropy | $0.62_{\pm0.15}$ | $0.58_{\pm0.11}$ | $0.67_{\pm0.12}$ | $0.77_{\pm0.15}$ | 0.66 | $6.95_{\pm0.78}$ | $0.25_{\pm0.16}$ | $1.51_{\pm0.34}$ | $2.03_{\pm0.73}$ | 2.68 |
| NPR | $2.24_{\pm0.24}$ | $1.72_{\pm0.20}$ | $1.03_{\pm0.06}$ | $3.95_{\pm0.69}$ | 2.23 | $6.33_{\pm1.38}$ | $2.87_{\pm1.13}$ | $1.19_{\pm0.30}$ | $20.00_{\pm1.61}$ | 7.60 |
| DetectGPT | $0.72_{\pm0.17}$ | $0.38_{\pm0.09}$ | $0.25_{\pm0.13}$ | $0.84_{\pm0.17}$ | 0.54 | $3.76_{\pm2.57}$ | $4.82_{\pm1.22}$ | $5.88_{\pm6.24}$ | $6.28_{\pm1.35}$ | 5.19 |
| FastGPT | $1.33_{\pm0.34}$ | $0.27_{\pm0.10}$ | $0.09_{\pm0.06}$ | $1.65_{\pm0.63}$ | 0.84 | $18.15_{\pm1.44}$ | $11.87_{\pm1.21}$ | $0.44_{\pm0.17}$ | $29.49_{\pm0.70}$ | 14.99 |
| ChatGPT-D | $9.56_{\pm1.25}$ | $9.73_{\pm1.63}$ | $3.26_{\pm0.49}$ | $13.56_{\pm1.48}$ | 9.03 | $18.15_{\pm3.77}$ | $19.28_{\pm8.65}$ | $4.23_{\pm1.88}$ | $41.98_{\pm10.40}$ | 20.91 |
| **ChatGPT-E** | $10.52_{\pm0.46}$ | $10.86_{\pm1.89}$ | $3.54_{\pm0.47}$ | $15.37_{\pm1.70}$ | 10.07 | $22.79_{\pm2.95}$ | $32.11_{\pm11.47}$ | $6.01_{\pm2.87}$ | $49.57_{\pm7.72}$ | 27.62 |
| MPU | $31.44_{\pm1.39}$ | $46.28_{\pm2.27}$ | $11.15_{\pm0.69}$ | $34.12_{\pm1.46}$ | 30.75 | $60.59_{\pm1.07}$ | $89.12_{\pm2.27}$ | $18.69_{\pm0.51}$ | $90.53_{\pm1.10}$ | 64.73 |
| **MPU-E** | $\mathbf{34.76_{\pm1.73}}$ | $52.61_{\pm2.26}$ | $12.82_{\pm1.59}$ | $\mathbf{36.15_{\pm1.47}}$ | 34.09 | $68.03_{\pm3.12}$ | $94.49_{\pm1.13}$ | $23.49_{\pm2.38}$ | $\mathbf{92.78_{\pm1.00}}$ | 69.70 |
| RADAR | $33.11_{\pm2.31}$ | $56.45_{\pm2.03}$ | $14.11_{\pm1.62}$ | $33.11_{\pm1.64}$ | 34.19 | $74.24_{\pm4.08}$ | $93.00_{\pm1.10}$ | $28.68_{\pm2.40}$ | $87.64_{\pm1.84}$ | 70.89 |
| **RADAR-E** | $34.70_{\pm1.61}$ | $\mathbf{62.24_{\pm1.22}}$ | $\mathbf{14.62_{\pm1.81}}$ | $34.41_{\pm1.71}$ | **36.49** | $\mathbf{74.26_{\pm4.67}}$ | $\mathbf{94.61_{\pm1.48}}$ | $28.68_{\pm2.45}$ | $89.37_{\pm2.25}$ | **71.73** |

Table 9: Performance concerning AUROC (%) on DetectRL. The detection model is trained on text generated by Claude.

| Method | Sentence-level | | | | | Paragraph-level | | | | |
|---|---|---|---|---|---|---|---|---|---|---|
| | PaLM | ChatGPT | Claude | Llama-2 | Avg. | PaLM | ChatGPT | Claude | Llama-2 | Avg. |
| Likelihood | $40.28_{\pm0.80}$ | $41.96_{\pm0.58}$ | $54.98_{\pm0.20}$ | $32.18_{\pm0.55}$ | 42.35 | $28.58_{\pm0.49}$ | $33.39_{\pm0.99}$ | $57.53_{\pm1.27}$ | $21.42_{\pm0.41}$ | 35.23 |
| Log-Rank | $40.94_{\pm0.88}$ | $44.08_{\pm0.62}$ | $55.72_{\pm0.21}$ | $32.29_{\pm0.54}$ | 43.26 | $28.36_{\pm0.49}$ | $34.01_{\pm0.96}$ | $57.89_{\pm1.31}$ | $20.04_{\pm0.45}$ | 35.07 |
| Entropy | $50.36_{\pm0.58}$ | $55.89_{\pm0.50}$ | $52.56_{\pm0.15}$ | $56.28_{\pm0.44}$ | 53.77 | $43.13_{\pm8.29}$ | $41.96_{\pm10.38}$ | $48.50_{\pm1.29}$ | $39.87_{\pm12.68}$ | 43.36 |
| NPR | $46.59_{\pm0.69}$ | $46.41_{\pm0.20}$ | $52.40_{\pm0.41}$ | $42.14_{\pm0.35}$ | 46.89 | $48.31_{\pm1.13}$ | $45.69_{\pm0.87}$ | $61.52_{\pm0.87}$ | $37.99_{\pm0.73}$ | 48.38 |
| DetectGPT | $46.94_{\pm0.64}$ | $45.69_{\pm0.20}$ | $52.65_{\pm0.37}$ | $43.07_{\pm0.36}$ | 47.09 | $57.92_{\pm0.72}$ | $50.04_{\pm0.98}$ | $65.37_{\pm0.59}$ | $49.21_{\pm0.55}$ | 55.63 |
| FastGPT | $62.43_{\pm0.48}$ | $55.88_{\pm0.58}$ | $40.81_{\pm0.31}$ | $67.12_{\pm0.37}$ | 56.56 | $59.94_{\pm0.93}$ | $61.26_{\pm0.93}$ | $24.14_{\pm1.06}$ | $70.39_{\pm0.44}$ | 53.93 |
| ChatGPT-D | $70.48_{\pm1.69}$ | $71.20_{\pm2.30}$ | $67.25_{\pm1.78}$ | $75.99_{\pm1.39}$ | 71.23 | $77.99_{\pm2.09}$ | $79.01_{\pm3.35}$ | $72.45_{\pm4.20}$ | $90.02_{\pm1.58}$ | 79.87 |
| **ChatGPT-E** | $70.45_{\pm1.43}$ | $71.15_{\pm1.92}$ | $66.91_{\pm1.46}$ | $76.01_{\pm1.14}$ | 71.13 | $77.09_{\pm2.20}$ | $80.08_{\pm4.86}$ | $74.11_{\pm6.18}$ | $90.52_{\pm2.22}$ | 80.45 |
| MPU | $81.53_{\pm0.66}$ | $89.80_{\pm0.39}$ | $84.89_{\pm0.71}$ | $86.18_{\pm0.65}$ | 85.60 | $93.73_{\pm0.41}$ | $97.93_{\pm0.22}$ | $97.00_{\pm0.83}$ | $98.80_{\pm0.19}$ | 96.87 |
| **MPU-E** | $82.12_{\pm0.52}$ | $91.19_{\pm0.53}$ | $\mathbf{87.33_{\pm0.59}}$ | $\mathbf{86.65_{\pm0.42}}$ | 86.82 | $94.29_{\pm0.68}$ | $\mathbf{98.30_{\pm0.27}}$ | $98.76_{\pm0.17}$ | $\mathbf{98.86_{\pm0.18}}$ | 97.55 |
| RADAR | $84.92_{\pm0.45}$ | $92.62_{\pm0.49}$ | $84.58_{\pm0.72}$ | $84.97_{\pm0.46}$ | 86.77 | $96.75_{\pm0.25}$ | $97.80_{\pm0.21}$ | $99.26_{\pm0.22}$ | $98.28_{\pm0.20}$ | 98.02 |
| **RADAR-E** | $\mathbf{85.47_{\pm0.63}}$ | $\mathbf{93.14_{\pm0.49}}$ | $86.43_{\pm0.43}$ | $85.57_{\pm0.21}$ | **87.65** | $\mathbf{97.12_{\pm0.34}}$ | $97.98_{\pm0.35}$ | $\mathbf{99.38_{\pm0.23}}$ | $98.42_{\pm0.29}$ | **98.23** |

this strategy can lead to better feature discrimination, thereby mitigating the negative effects of hard labels and fostering effective learning. To empirically verify these theoretical findings, we analyze the impact of the number of original texts constituting longer texts (i.e., the value of $k$) on supervisor performance, as shown in Fig. 17. Experimental results align with theoretical predictions: increasing the number of original text segments $k$ that constitute longer texts helps improve the supervisor's learning performance.

**Sensitivity w.r.t. the Number of Longer Texts $N'$ Per Batch**. Increasing the number of longer texts processed by the supervisor (reflected in the number of longer texts per batch, $N'$) should enhance its detection capability. To investigate this, we experimented with different settings of $N'$ to evaluate the supervisor's performance, as shown in Fig. 18. The experimental results meet expectations: as the value of $N'$ increases, the detection performance of the supervisor indeed improves.

**Sensitivity w.r.t. Supervisor Loss Coefficient $\lambda$**. In the joint training process shown in Eq. 5, the weight $\lambda$ of the supervisor loss term directly determines the emphasis on supervisor training within the overall optimization objective. To explore its impact on the supervisor's performance, we experimented with different $\lambda$ settings to evaluate the supervisor's performance, as illustrated in Fig. 19. it can be observed that in the relatively weaker ChatGPT-D, the supervisor requires a larger $\lambda$. In contrast, for the relatively stronger MPU and RADAR, the supervisor's performance initially improves with an increase in $\lambda$ value but reaches saturation and does not show significant further improvement. This might be related to the implementation of the supervisor, which, as explained in Appendix D.4, uses the input embeddings from the detector as its embeddings. In the stronger MPU and RADAR, the higher quality of embeddings provides a better initialization for the supervisor's learning, alleviating the overly large focus on the supervisor.

Table 10: Performance concerning TPR@FPR-1% (%) on DetectRL. The detection model is trained on text generated by Claude.

| Method | Sentence-level | | | | | Paragraph-level | | | | |
|---|---|---|---|---|---|---|---|---|---|---|
| | PaLM | ChatGPT | Claude | Llama-2 | Avg. | PaLM | ChatGPT | Claude | Llama-2 | Avg. |
| Likelihood | $0.50_{\pm0.23}$ | $0.35_{\pm0.11}$ | $1.30_{\pm0.32}$ | $0.25_{\pm0.04}$ | 0.60 | $0.72_{\pm0.21}$ | $0.30_{\pm0.10}$ | $3.73_{\pm0.65}$ | $0.10_{\pm0.09}$ | 1.21 |
| Log-Rank | $0.44_{\pm0.19}$ | $0.35_{\pm0.07}$ | $1.14_{\pm0.23}$ | $0.23_{\pm0.06}$ | 0.54 | $0.54_{\pm0.20}$ | $0.22_{\pm0.09}$ | $3.31_{\pm0.61}$ | $0.02_{\pm0.05}$ | 1.03 |
| Entropy | $0.62_{\pm0.15}$ | $0.58_{\pm0.11}$ | $0.67_{\pm0.12}$ | $0.77_{\pm0.15}$ | 0.66 | $1.80_{\pm1.83}$ | $0.12_{\pm0.11}$ | $1.24_{\pm0.17}$ | $0.57_{\pm1.08}$ | 0.93 |
| NPR | $2.06_{\pm0.35}$ | $1.92_{\pm0.37}$ | $2.67_{\pm0.10}$ | $1.84_{\pm0.33}$ | 2.12 | $7.32_{\pm0.91}$ | $7.27_{\pm1.75}$ | $10.48_{\pm2.22}$ | $5.39_{\pm0.59}$ | 7.61 |
| DetectGPT | $0.89_{\pm0.13}$ | $1.12_{\pm0.20}$ | $1.62_{\pm0.45}$ | $0.55_{\pm0.07}$ | 1.05 | $5.51_{\pm1.21}$ | $7.00_{\pm1.41}$ | $11.10_{\pm2.45}$ | $5.27_{\pm0.45}$ | 7.22 |
| FastGPT | $1.33_{\pm0.34}$ | $0.27_{\pm0.10}$ | $0.09_{\pm0.06}$ | $1.65_{\pm0.63}$ | 0.84 | $18.15_{\pm1.44}$ | $11.87_{\pm1.21}$ | $0.44_{\pm0.17}$ | $29.49_{\pm0.70}$ | 14.99 |
| ChatGPT-D | $8.24_{\pm1.40}$ | $7.63_{\pm0.93}$ | $3.54_{\pm0.80}$ | $12.34_{\pm0.59}$ | 7.94 | $15.75_{\pm4.99}$ | $13.47_{\pm5.83}$ | $4.89_{\pm2.38}$ | $35.65_{\pm6.84}$ | 17.44 |
| **ChatGPT-E** | $8.23_{\pm1.06}$ | $7.89_{\pm1.06}$ | $3.29_{\pm0.53}$ | $12.46_{\pm0.99}$ | 7.97 | $19.11_{\pm5.70}$ | $20.00_{\pm10.15}$ | $6.58_{\pm4.18}$ | $38.47_{\pm8.49}$ | 21.04 |
| MPU | $20.01_{\pm0.94}$ | $24.98_{\pm1.86}$ | $18.46_{\pm1.37}$ | $26.94_{\pm1.14}$ | 22.60 | $54.61_{\pm2.57}$ | $76.66_{\pm4.77}$ | $69.15_{\pm6.21}$ | $\mathbf{86.06}_{\pm1.37}$ | 71.62 |
| **MPU-E** | $21.16_{\pm1.63}$ | $31.80_{\pm2.94}$ | $\mathbf{22.34}_{\pm1.39}$ | $\mathbf{29.00}_{\pm1.21}$ | 26.08 | $56.91_{\pm3.79}$ | $\mathbf{78.12}_{\pm4.86}$ | $85.44_{\pm2.20}$ | $85.54_{\pm2.18}$ | 76.50 |
| RADAR | $\mathbf{23.93}_{\pm1.98}$ | $\mathbf{42.08}_{\pm1.12}$ | $19.68_{\pm0.47}$ | $23.40_{\pm0.86}$ | **27.27** | $\mathbf{68.13}_{\pm7.46}$ | $67.66_{\pm3.84}$ | $93.13_{\pm1.12}$ | $73.03_{\pm5.44}$ | 75.49 |
| **RADAR-E** | $23.28_{\pm2.77}$ | $41.12_{\pm2.53}$ | $21.57_{\pm1.79}$ | $21.78_{\pm1.79}$ | 26.94 | $67.81_{\pm8.69}$ | $68.73_{\pm7.87}$ | $\mathbf{94.12}_{\pm1.52}$ | $73.94_{\pm8.61}$ | 76.15 |

Table 11: Performance concerning AUROC (%) on DetectRL. The detection model is trained on text generated by Llama-2.

| Method | Sentence-level | | | | | Paragraph-level | | | | |
|---|---|---|---|---|---|---|---|---|---|---|
| | PaLM | ChatGPT | Claude | Llama-2 | Avg. | PaLM | ChatGPT | Claude | Llama-2 | Avg. |
| Likelihood | $59.72_{\pm0.80}$ | $58.04_{\pm0.58}$ | $45.02_{\pm0.20}$ | $67.82_{\pm0.55}$ | 57.65 | $71.42_{\pm0.49}$ | $66.61_{\pm0.99}$ | $42.47_{\pm1.27}$ | $78.58_{\pm0.41}$ | 64.77 |
| Log-Rank | $59.06_{\pm0.88}$ | $55.92_{\pm0.62}$ | $44.28_{\pm0.21}$ | $67.71_{\pm0.54}$ | 56.74 | $71.64_{\pm0.49}$ | $65.99_{\pm0.96}$ | $42.11_{\pm1.31}$ | $79.96_{\pm0.45}$ | 64.93 |
| Entropy | $50.36_{\pm0.58}$ | $55.89_{\pm0.50}$ | $52.56_{\pm0.15}$ | $56.28_{\pm0.44}$ | 53.77 | $60.73_{\pm0.91}$ | $63.08_{\pm1.05}$ | $51.73_{\pm0.96}$ | $66.21_{\pm0.88}$ | 60.44 |
| NPR | $53.41_{\pm0.69}$ | $53.59_{\pm0.20}$ | $47.60_{\pm0.41}$ | $57.86_{\pm0.35}$ | 53.11 | $51.69_{\pm1.13}$ | $54.31_{\pm0.87}$ | $38.48_{\pm0.87}$ | $62.01_{\pm0.73}$ | 51.62 |
| DetectGPT | $53.06_{\pm0.64}$ | $54.31_{\pm0.29}$ | $47.35_{\pm0.37}$ | $56.93_{\pm0.36}$ | 52.91 | $44.74_{\pm5.97}$ | $49.35_{\pm0.74}$ | $41.06_{\pm12.51}$ | $50.41_{\pm0.87}$ | 46.39 |
| FastGPT | $62.43_{\pm0.48}$ | $55.88_{\pm0.58}$ | $40.81_{\pm0.31}$ | $67.12_{\pm0.37}$ | 56.56 | $59.94_{\pm0.93}$ | $61.26_{\pm0.93}$ | $24.14_{\pm1.06}$ | $70.39_{\pm0.44}$ | 53.93 |
| ChatGPT-D | $71.95_{\pm1.72}$ | $73.82_{\pm2.54}$ | $65.92_{\pm1.95}$ | $78.71_{\pm1.80}$ | 72.60 | $80.90_{\pm2.22}$ | $84.86_{\pm3.60}$ | $70.41_{\pm2.05}$ | $93.66_{\pm1.27}$ | 82.46 |
| **ChatGPT-E** | $73.16_{\pm2.01}$ | $75.58_{\pm2.88}$ | $67.24_{\pm1.65}$ | $80.09_{\pm2.28}$ | 74.02 | $82.30_{\pm1.26}$ | $86.33_{\pm1.81}$ | $72.35_{\pm1.67}$ | $94.32_{\pm0.93}$ | 83.82 |
| MPU | $83.85_{\pm0.33}$ | $90.38_{\pm0.48}$ | $74.43_{\pm0.80}$ | $90.85_{\pm0.34}$ | 84.88 | $94.06_{\pm0.29}$ | $97.95_{\pm0.34}$ | $74.59_{\pm1.11}$ | $99.13_{\pm0.13}$ | 91.43 |
| **MPU-E** | $85.08_{\pm0.41}$ | $91.61_{\pm0.42}$ | $75.09_{\pm0.89}$ | $\mathbf{91.90}_{\pm0.31}$ | 85.92 | $95.43_{\pm0.25}$ | $98.79_{\pm0.27}$ | $77.82_{\pm0.75}$ | $99.42_{\pm0.14}$ | 92.86 |
| RADAR | $86.82_{\pm0.41}$ | $93.36_{\pm0.37}$ | $75.32_{\pm0.65}$ | $89.82_{\pm0.25}$ | 86.33 | $97.57_{\pm0.34}$ | $99.30_{\pm0.13}$ | $88.09_{\pm1.29}$ | $99.33_{\pm0.07}$ | 96.07 |
| **RADAR-E** | $\mathbf{87.88}_{\pm0.30}$ | $\mathbf{94.24}_{\pm0.28}$ | $\mathbf{76.47}_{\pm0.74}$ | $90.98_{\pm0.23}$ | **87.39** | $\mathbf{97.77}_{\pm0.47}$ | $\mathbf{99.44}_{\pm0.19}$ | $\mathbf{89.47}_{\pm1.68}$ | $\mathbf{99.43}_{\pm0.04}$ | **96.53** |

#### D.9.2 About the Detector

**Sensitivity w.r.t. Original Text Number $k$ for Longer Text**. According to our theoretical results (Theorem 3.1), longer text lengths help achieve greater distribution distance for text data, thereby simplifying the supervisor's learning difficulty and laying the foundation for providing reliable supervision to the detector (empirically proved in Fig. 17). To this end, we explore the impact of different original text numbers $k$ for longer text on detector performance, as shown in Fig. 20. The setup with $k = 0$ represents the original detector baseline without using the enhancement strategy. The experimental results align with the theoretical predictions: using longer texts for supervised learning enhances the supervisor's performance (Fig. 17), thereby providing more reliable supervisory signals and ultimately improving detector performance.

**Sensitivity w.r.t. the Number of Longer Texts $N'$ Per Batch**. The performance of the supervisor significantly impacts the target detector's performance, and the supervisor's own effectiveness largely depends on the amount of longer texts (proved in Fig. 18). Here we aim to explore the impact of varying quantities of longer text data on the performance of the detector, as shown in Fig. 21. $N' = 0$ represents the original model without enhancement. The results are as expected: as the amount of long text data used for training the supervisor increases, the detector's learning is better supported, leading to improved performance. This can also be seen from the consistent trend of changes in Fig. 18 and 21. Although increasing the data volume might introduce additional computational overhead, as indicated in our previous runtime analysis, even with relatively large data settings (e.g., N=128), the additional training delay remains minimal.

**Sensitivity w.r.t. Supervisor Loss Coefficient $\lambda$**. We also investigated the impact of the supervisor's loss term coefficient $\lambda$ on detector performance, as illustrated in Fig. 22. We can also find that the detector's performance change curve is consistent with the change of the supervisor (Fig. 19), which also indirectly emphasizes the guiding role of the supervisor on the detector.

Table 12: Performance concerning TPR@FPR-1% (%) on DetectRL. The detection model is trained on text generated by Llama-2.

| Method | Sentence-level | | | | | Paragraph-level | | | | |
|---|---|---|---|---|---|---|---|---|---|---|
| | PaLM | ChatGPT | Claude | Llama-2 | Avg. | PaLM | ChatGPT | Claude | Llama-2 | Avg. |
| Likelihood | $4.83_{\pm0.39}$ | $1.58_{\pm0.23}$ | $0.72_{\pm0.13}$ | $5.54_{\pm0.40}$ | 3.17 | $25.66_{\pm2.41}$ | $10.21_{\pm1.40}$ | $1.78_{\pm0.38}$ | $38.39_{\pm0.92}$ | 19.01 |
| Log-Rank | $4.84_{\pm0.41}$ | $1.23_{\pm0.25}$ | $0.72_{\pm0.13}$ | $5.25_{\pm0.87}$ | 3.01 | $27.49_{\pm1.13}$ | $11.55_{\pm1.93}$ | $2.08_{\pm0.65}$ | $41.93_{\pm0.47}$ | 20.76 |
| Entropy | $0.62_{\pm0.15}$ | $0.58_{\pm0.11}$ | $0.67_{\pm0.12}$ | $0.77_{\pm0.15}$ | 0.66 | $6.95_{\pm0.78}$ | $0.25_{\pm0.16}$ | $1.51_{\pm0.34}$ | $2.03_{\pm0.73}$ | 2.68 |
| NPR | $2.24_{\pm0.24}$ | $1.72_{\pm0.20}$ | $1.03_{\pm0.06}$ | $3.95_{\pm0.69}$ | 2.23 | $6.33_{\pm1.38}$ | $2.87_{\pm1.13}$ | $1.19_{\pm0.30}$ | $20.00_{\pm1.61}$ | 7.60 |
| DetectGPT | $0.72_{\pm0.17}$ | $0.38_{\pm0.09}$ | $0.25_{\pm0.13}$ | $0.84_{\pm0.17}$ | 0.54 | $3.09_{\pm2.15}$ | $4.75_{\pm1.19}$ | $3.76_{\pm5.61}$ | $6.33_{\pm1.30}$ | 4.48 |
| FastGPT | $1.33_{\pm0.34}$ | $0.27_{\pm0.10}$ | $0.09_{\pm0.06}$ | $1.65_{\pm0.63}$ | 0.84 | $18.15_{\pm1.44}$ | $11.87_{\pm1.21}$ | $0.44_{\pm0.17}$ | $29.49_{\pm0.70}$ | 14.99 |
| ChatGPT-D | $9.36_{\pm1.21}$ | $9.00_{\pm2.35}$ | $3.04_{\pm0.61}$ | $13.69_{\pm1.91}$ | 8.77 | $22.32_{\pm3.93}$ | $23.34_{\pm7.14}$ | $5.41_{\pm2.08}$ | $48.58_{\pm6.35}$ | 24.91 |
| **ChatGPT-E** | $10.30_{\pm1.79}$ | $10.11_{\pm1.75}$ | $3.15_{\pm0.42}$ | $15.46_{\pm2.96}$ | 9.75 | $24.00_{\pm3.90}$ | $24.55_{\pm6.55}$ | $5.69_{\pm2.64}$ | $52.41_{\pm7.39}$ | 26.66 |
| MPU | $23.94_{\pm1.21}$ | $27.25_{\pm1.98}$ | $6.78_{\pm0.58}$ | $33.77_{\pm0.39}$ | 22.93 | $55.97_{\pm4.02}$ | $74.81_{\pm1.66}$ | $15.50_{\pm1.62}$ | $89.25_{\pm0.74}$ | 58.88 |
| **MPU-E** | $24.82_{\pm1.25}$ | $29.96_{\pm2.26}$ | $7.13_{\pm0.97}$ | $34.58_{\pm1.50}$ | 24.12 | $62.30_{\pm4.30}$ | $81.58_{\pm1.59}$ | $17.16_{\pm2.00}$ | $92.86_{\pm0.90}$ | 63.47 |
| RADAR | $31.27_{\pm0.94}$ | $43.70_{\pm3.06}$ | $11.38_{\pm0.72}$ | $36.68_{\pm1.44}$ | 30.76 | $75.40_{\pm4.54}$ | $88.68_{\pm2.15}$ | $34.14_{\pm3.84}$ | $88.03_{\pm1.79}$ | 71.56 |
| **RADAR-E** | $34.58_{\pm1.92}$ | $49.46_{\pm3.36}$ | $12.39_{\pm1.04}$ | $39.59_{\pm1.23}$ | **34.01** | $76.12_{\pm6.60}$ | $91.52_{\pm1.25}$ | $38.64_{\pm6.29}$ | $90.14_{\pm2.24}$ | **74.10** |

Table 13: Performance concerning AUROC (%) on Essay under sentence-level settings. The detection model is trained on text generated by GPT4All.

| Method | GPT4All | ChatGPT | ChatGPT-turbo | ChatGLM | Dolly | Claude | Avg. |
|---|---|---|---|---|---|---|---|
| Likelihood | $81.83_{\pm0.40}$ | $82.98_{\pm0.28}$ | $75.88_{\pm0.46}$ | $89.59_{\pm0.41}$ | $75.42_{\pm0.33}$ | $62.62_{\pm0.35}$ | 78.05 |
| Log-Rank | $80.82_{\pm0.42}$ | $81.54_{\pm0.34}$ | $73.73_{\pm0.53}$ | $89.42_{\pm0.38}$ | $74.27_{\pm0.28}$ | $60.11_{\pm0.42}$ | 76.65 |
| Entropy | $59.77_{\pm0.44}$ | $64.67_{\pm0.21}$ | $65.86_{\pm0.26}$ | $60.94_{\pm0.33}$ | $58.29_{\pm0.43}$ | $59.10_{\pm0.29}$ | 61.44 |
| NPR | $72.16_{\pm0.47}$ | $71.46_{\pm0.36}$ | $68.92_{\pm0.74}$ | $77.78_{\pm0.38}$ | $67.21_{\pm0.59}$ | $64.14_{\pm0.59}$ | 70.28 |
| DetectGPT | $73.01_{\pm0.34}$ | $72.19_{\pm0.33}$ | $70.72_{\pm0.61}$ | $77.19_{\pm0.30}$ | $68.59_{\pm0.39}$ | $65.74_{\pm0.55}$ | 71.24 |
| FastGPT | $81.65_{\pm0.20}$ | $80.14_{\pm0.40}$ | $71.65_{\pm0.67}$ | $89.85_{\pm0.26}$ | $77.29_{\pm0.34}$ | $63.19_{\pm0.20}$ | 77.29 |
| ChatGPT-D | $80.64_{\pm0.66}$ | $78.63_{\pm0.76}$ | $74.95_{\pm0.72}$ | $86.73_{\pm1.04}$ | $66.33_{\pm1.21}$ | $60.58_{\pm1.10}$ | 74.64 |
| **ChatGPT-E** | $81.56_{\pm1.24}$ | $79.21_{\pm1.23}$ | $75.39_{\pm1.35}$ | $87.74_{\pm1.17}$ | $66.76_{\pm1.23}$ | $60.34_{\pm1.62}$ | 75.16 |
| MPU | $87.83_{\pm0.71}$ | $85.44_{\pm0.79}$ | $83.60_{\pm0.97}$ | $91.58_{\pm0.60}$ | $73.46_{\pm0.58}$ | $69.08_{\pm0.95}$ | 81.83 |
| **MPU-E** | $89.54_{\pm0.45}$ | $87.25_{\pm0.38}$ | $86.06_{\pm0.52}$ | $92.89_{\pm0.31}$ | $75.82_{\pm0.24}$ | $72.09_{\pm0.92}$ | 83.94 |
| RADAR | $91.55_{\pm0.38}$ | $91.62_{\pm0.34}$ | $91.86_{\pm0.45}$ | $94.02_{\pm0.36}$ | $83.48_{\pm0.44}$ | $80.43_{\pm0.73}$ | 88.83 |
| **RADAR-E** | $92.39_{\pm0.41}$ | $92.44_{\pm0.36}$ | $92.76_{\pm0.26}$ | $94.81_{\pm0.33}$ | $83.99_{\pm0.45}$ | $80.80_{\pm1.10}$ | **89.53** |

### D.10  Ablation Study of Supervisor

In our framework design, we aim to encourage the detector to correctly classify each sample within longer texts, rather than strictly requiring the predicted probability of the correct class to approach 1. This raises the question of whether directly using the loss function only focused on the class could achieve similar goals. Therefore, we conduct an ablation study using two class-only loss functions to highlight the role of the supervisor in enhancement.

First, we define the training loss of the baseline model after removing the supervisor module as follows:

$$\mathcal{L}_{supv.} = -\frac{1}{kN'} \sum_{i=1}^{N'} \sum_{j=1}^{k} \left( y_{long,i} \log \text{gumbel}(f(x_i^{(j)}, \theta_f)) + (1 - y_{long,i}) \log(1 - \text{gumbel}(f(x_i^{(j))}, \theta_f))) \right).$$

To ensure that $\text{gumbel}(f(x_i^{(j)}, \theta_f))$ is meaningful, the non-hard-label version of Gumbel-Softmax is used.

Second, we use Hinge loss to replace $\mathcal{L}_{supv.}$ as follows,

$$\mathcal{L}_{supv.} = \frac{1}{kN'} \sum_{i=1}^{N'} \sum_{j=1}^{k} \max(0, -(y_{long,i} * 2 - 1) * (f(x_i^{(j)}, \theta_f) * 2 - 1)).$$

These two variants are defined as Gumbel and Hinge. Aside from the form of the supervisory signal, all other experimental settings (such as $k$, $N'$, $\lambda$) remain consistent. The experimental results are shown in Fig. 23. It can be observed that using these alternative loss functions focused solely on class can enhance detection performance in some settings, yet there are instances of instability. For example, applying the Hinge loss to the RADAR model on the Essay dataset resulted in a performance decline. Furthermore, even though these alternatives provided performance improvements in certain

Table 14: Performance concerning AUROC (%) on Essay under sentence-level settings. The detection model is trained on text generated by ChatGPT.

| Method | GPT4All | ChatGPT | ChatGPT-turbo | ChatGLM | Dolly | Claude | Avg. |
|---|---|---|---|---|---|---|---|
| Likelihood | $81.83_{\pm0.40}$ | $82.98_{\pm0.28}$ | $75.88_{\pm0.46}$ | $89.59_{\pm0.41}$ | $75.42_{\pm0.33}$ | $62.62_{\pm0.35}$ | 78.05 |
| Log-Rank | $80.82_{\pm0.42}$ | $81.54_{\pm0.34}$ | $73.73_{\pm0.53}$ | $89.42_{\pm0.38}$ | $74.27_{\pm0.28}$ | $60.11_{\pm0.42}$ | 76.65 |
| Entropy | $59.77_{\pm0.44}$ | $64.67_{\pm0.21}$ | $65.86_{\pm0.26}$ | $60.94_{\pm0.33}$ | $58.29_{\pm0.43}$ | $59.10_{\pm0.29}$ | 61.44 |
| NPR | $72.16_{\pm0.47}$ | $71.46_{\pm0.36}$ | $68.92_{\pm0.74}$ | $77.78_{\pm0.38}$ | $67.21_{\pm0.59}$ | $64.14_{\pm0.59}$ | 70.28 |
| DetectGPT | $73.01_{\pm0.34}$ | $72.19_{\pm0.33}$ | $70.72_{\pm0.61}$ | $77.19_{\pm0.30}$ | $68.59_{\pm0.39}$ | $65.74_{\pm0.55}$ | 71.24 |
| FastGPT | $81.65_{\pm0.20}$ | $80.14_{\pm0.40}$ | $71.65_{\pm0.67}$ | $89.85_{\pm0.26}$ | $77.29_{\pm0.34}$ | $63.19_{\pm0.20}$ | 77.29 |
| ChatGPT-D | $78.41_{\pm2.07}$ | $77.16_{\pm1.52}$ | $72.56_{\pm1.48}$ | $85.11_{\pm1.95}$ | $64.39_{\pm1.49}$ | $58.52_{\pm1.12}$ | 72.69 |
| **ChatGPT-E** | $79.75_{\pm2.55}$ | $78.43_{\pm1.98}$ | $73.31_{\pm3.00}$ | $86.80_{\pm2.10}$ | $65.95_{\pm1.83}$ | $59.77_{\pm2.35}$ | 74.00 |
| MPU | $87.78_{\pm0.44}$ | $87.10_{\pm0.36}$ | $83.70_{\pm0.36}$ | $92.91_{\pm0.67}$ | $74.35_{\pm0.62}$ | $69.62_{\pm0.61}$ | 82.58 |
| **MPU-E** | $89.28_{\pm0.28}$ | $88.59_{\pm0.25}$ | $85.96_{\pm0.32}$ | $93.93_{\pm0.56}$ | $76.39_{\pm0.56}$ | $71.87_{\pm1.16}$ | 84.34 |
| RADAR | $90.82_{\pm0.66}$ | $92.47_{\pm0.52}$ | $91.89_{\pm0.54}$ | $94.58_{\pm0.46}$ | $83.77_{\pm0.78}$ | $80.75_{\pm1.63}$ | 89.05 |
| **RADAR-E** | $\mathbf{91.06}_{\pm0.30}$ | $\mathbf{92.82}_{\pm0.31}$ | $\mathbf{92.32}_{\pm0.20}$ | $\mathbf{94.85}_{\pm0.51}$ | $\mathbf{84.25}_{\pm0.41}$ | $\mathbf{81.31}_{\pm1.40}$ | **89.43** |

Table 15: Performance concerning TPR@FPR-1% (%) on Essay under sentence-level settings. The detection model is trained on text generated by ChatGPT.

| Method | GPT4All | ChatGPT | ChatGPT-turbo | ChatGLM | Dolly | Claude | Avg. |
|---|---|---|---|---|---|---|---|
| Likelihood | $9.18_{\pm1.56}$ | $14.05_{\pm0.57}$ | $5.46_{\pm0.46}$ | $26.04_{\pm4.35}$ | $6.86_{\pm1.13}$ | $2.82_{\pm0.17}$ | 10.73 |
| Log-Rank | $8.98_{\pm1.11}$ | $13.35_{\pm0.58}$ | $4.97_{\pm0.50}$ | $29.04_{\pm3.44}$ | $6.78_{\pm1.66}$ | $2.32_{\pm0.05}$ | 10.91 |
| Entropy | $1.36_{\pm0.26}$ | $2.40_{\pm0.55}$ | $1.65_{\pm0.34}$ | $1.10_{\pm0.31}$ | $1.70_{\pm0.43}$ | $1.44_{\pm0.18}$ | 1.61 |
| NPR | $6.13_{\pm0.66}$ | $7.13_{\pm0.54}$ | $4.04_{\pm0.74}$ | $14.14_{\pm1.85}$ | $4.71_{\pm0.51}$ | $3.26_{\pm0.43}$ | 6.57 |
| DetectGPT | $4.11_{\pm0.57}$ | $3.57_{\pm0.37}$ | $3.48_{\pm0.33}$ | $5.77_{\pm1.47}$ | $3.70_{\pm0.75}$ | $3.44_{\pm0.17}$ | 4.01 |
| FastGPT | $12.38_{\pm1.60}$ | $10.30_{\pm0.26}$ | $4.58_{\pm0.68}$ | $28.86_{\pm2.71}$ | $9.01_{\pm0.66}$ | $2.72_{\pm0.43}$ | 11.31 |
| ChatGPT-D | $13.71_{\pm1.23}$ | $10.96_{\pm0.89}$ | $8.60_{\pm1.22}$ | $22.54_{\pm2.80}$ | $4.26_{\pm0.33}$ | $1.92_{\pm0.23}$ | 10.33 |
| **ChatGPT-E** | $14.18_{\pm1.94}$ | $11.49_{\pm1.57}$ | $8.35_{\pm1.68}$ | $24.37_{\pm3.84}$ | $4.25_{\pm0.65}$ | $2.11_{\pm0.33}$ | 10.79 |
| MPU | $20.44_{\pm1.57}$ | $16.39_{\pm1.40}$ | $12.98_{\pm0.94}$ | $34.36_{\pm3.47}$ | $5.42_{\pm0.85}$ | $2.58_{\pm0.04}$ | 15.36 |
| **MPU-E** | $24.19_{\pm2.45}$ | $20.47_{\pm2.10}$ | $16.72_{\pm1.70}$ | $38.02_{\pm5.16}$ | $7.01_{\pm1.46}$ | $3.30_{\pm0.35}$ | 18.29 |
| RADAR | $\mathbf{26.16}_{\pm1.98}$ | $31.74_{\pm2.98}$ | $30.90_{\pm2.32}$ | $\mathbf{39.90}_{\pm3.90}$ | $\mathbf{11.99}_{\pm1.63}$ | $\mathbf{10.09}_{\pm1.58}$ | **25.13** |
| **RADAR-E** | $25.68_{\pm2.13}$ | $\mathbf{32.45}_{\pm3.06}$ | $\mathbf{30.97}_{\pm1.14}$ | $39.16_{\pm3.35}$ | $11.98_{\pm1.23}$ | $9.82_{\pm1.30}$ | 25.01 |

settings, their enhancement effects were generally inferior to our proposed strategy. This is because our strategy not only focuses on correct classification but, more importantly, guides the model's predicted probabilities to approximate the underlying true labels through the supervisor's signal (see Theorem 3.4), offering richer supervisory information.

Table 16: Performance concerning AUROC (%) on Essay under sentence-level settings. The detection model is trained on text generated by ChatGPT-turbo.

| Method | GPT4All | ChatGPT | ChatGPT-turbo | ChatGLM | Dolly | Claude | Avg. |
|---|---|---|---|---|---|---|---|
| Likelihood | $81.83_{\pm0.40}$ | $82.98_{\pm0.28}$ | $75.88_{\pm0.46}$ | $89.59_{\pm0.41}$ | $75.42_{\pm0.33}$ | $62.62_{\pm0.35}$ | 78.05 |
| Log-Rank | $80.82_{\pm0.42}$ | $81.54_{\pm0.34}$ | $73.73_{\pm0.53}$ | $89.42_{\pm0.38}$ | $74.27_{\pm0.28}$ | $60.11_{\pm0.42}$ | 76.65 |
| Entropy | $59.77_{\pm0.44}$ | $64.67_{\pm0.21}$ | $65.86_{\pm0.26}$ | $60.94_{\pm0.33}$ | $58.29_{\pm0.43}$ | $59.10_{\pm0.29}$ | 61.44 |
| NPR | $72.16_{\pm0.47}$ | $71.46_{\pm0.36}$ | $68.92_{\pm0.74}$ | $77.78_{\pm0.38}$ | $67.21_{\pm0.59}$ | $64.14_{\pm0.59}$ | 70.28 |
| DetectGPT | $73.01_{\pm0.34}$ | $72.19_{\pm0.33}$ | $70.72_{\pm0.61}$ | $77.19_{\pm0.30}$ | $68.59_{\pm0.39}$ | $65.74_{\pm0.55}$ | 71.24 |
| FastGPT | $81.65_{\pm0.20}$ | $80.14_{\pm0.40}$ | $71.65_{\pm0.67}$ | $89.85_{\pm0.26}$ | $77.29_{\pm0.34}$ | $63.19_{\pm0.20}$ | 77.29 |
| ChatGPT-D | $77.63_{\pm2.85}$ | $76.49_{\pm1.88}$ | $75.88_{\pm2.96}$ | $82.34_{\pm2.64}$ | $63.53_{\pm2.30}$ | $60.98_{\pm1.92}$ | 72.81 |
| **ChatGPT-E** | $78.76_{\pm3.99}$ | $77.34_{\pm2.83}$ | $78.00_{\pm4.90}$ | $82.83_{\pm3.33}$ | $64.52_{\pm3.23}$ | $62.95_{\pm3.58}$ | 74.07 |
| MPU | $86.63_{\pm0.15}$ | $85.17_{\pm0.22}$ | $89.76_{\pm0.19}$ | $88.62_{\pm0.70}$ | $72.01_{\pm1.09}$ | $74.73_{\pm0.41}$ | 82.82 |
| **MPU-E** | $87.65_{\pm0.85}$ | $86.27_{\pm0.73}$ | $91.45_{\pm1.29}$ | $89.17_{\pm0.90}$ | $73.56_{\pm1.50}$ | $77.44_{\pm1.92}$ | 84.26 |
| RADAR | $89.39_{\pm0.19}$ | $90.58_{\pm0.22}$ | $94.72_{\pm0.28}$ | $90.85_{\pm0.55}$ | $81.67_{\pm0.77}$ | $\mathbf{85.46_{\pm0.82}}$ | 88.78 |
| **RADAR-E** | $\mathbf{89.94_{\pm0.43}}$ | $\mathbf{91.17_{\pm0.41}}$ | $\mathbf{95.33_{\pm0.24}}$ | $\mathbf{91.43_{\pm1.00}}$ | $\mathbf{81.83_{\pm1.28}}$ | $85.29_{\pm0.88}$ | **89.17** |

Table 17: Performance concerning TPR@FPR-1% (%) on Essay under sentence-level settings. The detection model is trained on text generated by ChatGPT-turbo.

| Method | GPT4All | ChatGPT | ChatGPT-turbo | ChatGLM | Dolly | Claude | Avg. |
|---|---|---|---|---|---|---|---|
| Likelihood | $9.18_{\pm1.56}$ | $14.05_{\pm0.57}$ | $5.46_{\pm0.46}$ | $26.04_{\pm4.35}$ | $6.86_{\pm1.13}$ | $2.82_{\pm0.17}$ | 10.73 |
| Log-Rank | $8.98_{\pm1.11}$ | $13.35_{\pm0.58}$ | $4.97_{\pm0.50}$ | $29.04_{\pm3.44}$ | $6.78_{\pm1.66}$ | $2.32_{\pm0.05}$ | 10.91 |
| Entropy | $1.36_{\pm0.26}$ | $2.40_{\pm0.55}$ | $1.65_{\pm0.34}$ | $1.10_{\pm0.31}$ | $1.70_{\pm0.43}$ | $1.44_{\pm0.18}$ | 1.61 |
| NPR | $6.13_{\pm0.66}$ | $7.13_{\pm0.54}$ | $4.04_{\pm0.74}$ | $14.14_{\pm1.85}$ | $4.71_{\pm0.51}$ | $3.26_{\pm0.43}$ | 6.57 |
| DetectGPT | $4.11_{\pm0.57}$ | $3.57_{\pm0.37}$ | $3.48_{\pm0.33}$ | $5.77_{\pm1.47}$ | $3.70_{\pm0.75}$ | $3.44_{\pm0.17}$ | 4.01 |
| FastGPT | $12.38_{\pm1.60}$ | $10.30_{\pm0.26}$ | $4.58_{\pm0.68}$ | $28.86_{\pm2.71}$ | $9.01_{\pm0.66}$ | $2.72_{\pm0.43}$ | 11.31 |
| ChatGPT-D | $15.12_{\pm1.81}$ | $10.91_{\pm0.88}$ | $11.07_{\pm1.50}$ | $20.20_{\pm2.84}$ | $4.49_{\pm0.76}$ | $2.56_{\pm0.71}$ | 10.73 |
| **ChatGPT-E** | $15.58_{\pm2.72}$ | $11.14_{\pm1.61}$ | $13.81_{\pm3.96}$ | $19.89_{\pm2.52}$ | $4.69_{\pm1.08}$ | $3.24_{\pm0.99}$ | 11.39 |
| MPU | $20.69_{\pm1.62}$ | $15.92_{\pm1.70}$ | $25.56_{\pm1.23}$ | $24.93_{\pm2.44}$ | $5.96_{\pm1.01}$ | $4.61_{\pm0.39}$ | 16.28 |
| **MPU-E** | $23.32_{\pm0.80}$ | $18.15_{\pm2.02}$ | $30.58_{\pm4.13}$ | $25.05_{\pm2.27}$ | $6.87_{\pm0.74}$ | $6.27_{\pm0.91}$ | 18.37 |
| RADAR | $24.82_{\pm1.13}$ | $26.76_{\pm1.60}$ | $43.08_{\pm2.09}$ | $25.44_{\pm2.67}$ | $11.14_{\pm1.48}$ | $\mathbf{17.00_{\pm1.96}}$ | 24.71 |
| **RADAR-E** | $\mathbf{26.80_{\pm2.46}}$ | $\mathbf{28.49_{\pm3.46}}$ | $\mathbf{48.15_{\pm2.91}}$ | $\mathbf{29.12_{\pm4.94}}$ | $\mathbf{11.85_{\pm1.89}}$ | $14.97_{\pm3.11}$ | **26.56** |

Table 18: Performance concerning AUROC (%) on Essay under sentence-level settings. The detection model is trained on text generated by ChatGLM.

| Method | GPT4All | ChatGPT | ChatGPT-turbo | ChatGLM | Dolly | Claude | Avg. |
|---|---|---|---|---|---|---|---|
| Likelihood | $81.83_{\pm0.40}$ | $82.98_{\pm0.28}$ | $75.88_{\pm0.46}$ | $89.59_{\pm0.41}$ | $75.42_{\pm0.33}$ | $62.62_{\pm0.35}$ | 78.05 |
| Log-Rank | $80.82_{\pm0.42}$ | $81.54_{\pm0.34}$ | $73.73_{\pm0.53}$ | $89.42_{\pm0.38}$ | $74.27_{\pm0.28}$ | $60.11_{\pm0.42}$ | 76.65 |
| Entropy | $59.77_{\pm0.44}$ | $64.67_{\pm0.21}$ | $65.86_{\pm0.26}$ | $60.94_{\pm0.33}$ | $58.29_{\pm0.43}$ | $59.10_{\pm0.29}$ | 61.44 |
| NPR | $72.16_{\pm0.47}$ | $71.46_{\pm0.36}$ | $68.92_{\pm0.74}$ | $77.78_{\pm0.38}$ | $67.21_{\pm0.59}$ | $64.14_{\pm0.59}$ | 70.28 |
| DetectGPT | $73.01_{\pm0.34}$ | $72.19_{\pm0.33}$ | $70.72_{\pm0.61}$ | $77.19_{\pm0.30}$ | $68.59_{\pm0.39}$ | $65.74_{\pm0.55}$ | 71.24 |
| FastGPT | $81.65_{\pm0.20}$ | $80.14_{\pm0.40}$ | $71.65_{\pm0.67}$ | $89.85_{\pm0.26}$ | $77.29_{\pm0.34}$ | $63.19_{\pm0.20}$ | 77.29 |
| ChatGPT-D | $79.25_{\pm2.22}$ | $77.72_{\pm1.89}$ | $71.41_{\pm1.90}$ | $86.94_{\pm2.28}$ | $65.20_{\pm1.85}$ | $57.67_{\pm1.53}$ | 73.03 |
| **ChatGPT-E** | $80.44_{\pm1.73}$ | $78.53_{\pm1.58}$ | $72.22_{\pm1.47}$ | $88.05_{\pm1.84}$ | $66.27_{\pm1.64}$ | $58.24_{\pm1.42}$ | 73.96 |
| MPU | $86.30_{\pm0.45}$ | $85.02_{\pm0.18}$ | $76.39_{\pm0.47}$ | $93.71_{\pm0.13}$ | $71.69_{\pm0.18}$ | $62.43_{\pm0.95}$ | 79.26 |
| **MPU-E** | $87.30_{\pm0.52}$ | $86.01_{\pm0.39}$ | $77.94_{\pm0.89}$ | $94.36_{\pm0.29}$ | $73.00_{\pm0.67}$ | $64.03_{\pm0.66}$ | 80.44 |
| RADAR | $90.68_{\pm0.45}$ | $91.86_{\pm0.31}$ | $88.95_{\pm0.37}$ | $95.30_{\pm0.60}$ | $82.51_{\pm0.61}$ | $\mathbf{73.63_{\pm0.70}}$ | 87.16 |
| **RADAR-E** | $\mathbf{91.23_{\pm0.32}}$ | $\mathbf{92.33_{\pm0.20}}$ | $\mathbf{89.03_{\pm0.50}}$ | $\mathbf{96.07_{\pm0.29}}$ | $\mathbf{83.04_{\pm0.20}}$ | $73.33_{\pm0.87}$ | **87.51** |

Table 19: Performance concerning TPR@FPR-1% (%) on Essay under sentence-level settings. The detection model is trained on text generated by ChatGLM.

| Method | GPT4All | ChatGPT | ChatGPT-turbo | ChatGLM | Dolly | Claude | Avg. |
|---|---|---|---|---|---|---|---|
| Likelihood | $9.18_{\pm1.56}$ | $14.05_{\pm0.57}$ | $5.46_{\pm0.46}$ | $26.04_{\pm4.35}$ | $6.86_{\pm1.13}$ | $2.82_{\pm0.17}$ | 10.73 |
| Log-Rank | $8.98_{\pm1.11}$ | $13.35_{\pm0.58}$ | $4.97_{\pm0.50}$ | $29.04_{\pm3.44}$ | $6.78_{\pm1.66}$ | $2.32_{\pm0.05}$ | 10.91 |
| Entropy | $1.36_{\pm0.26}$ | $2.40_{\pm0.55}$ | $1.65_{\pm0.34}$ | $1.10_{\pm0.31}$ | $1.70_{\pm0.43}$ | $1.44_{\pm0.18}$ | 1.61 |
| NPR | $6.13_{\pm0.66}$ | $7.13_{\pm0.54}$ | $4.04_{\pm0.74}$ | $14.14_{\pm1.85}$ | $4.71_{\pm0.51}$ | $3.26_{\pm0.43}$ | 6.57 |
| DetectGPT | $4.11_{\pm0.57}$ | $3.57_{\pm0.37}$ | $3.48_{\pm0.33}$ | $5.77_{\pm1.47}$ | $3.70_{\pm0.75}$ | $3.44_{\pm0.17}$ | 4.01 |
| FastGPT | $12.38_{\pm1.60}$ | $10.30_{\pm0.26}$ | $4.58_{\pm0.68}$ | $28.86_{\pm2.71}$ | $9.01_{\pm0.66}$ | $2.72_{\pm0.43}$ | 11.31 |
| ChatGPT-D | $14.16_{\pm2.15}$ | $11.23_{\pm1.72}$ | $7.85_{\pm1.12}$ | $24.64_{\pm4.12}$ | $4.29_{\pm0.43}$ | $1.66_{\pm0.37}$ | 10.64 |
| **ChatGPT-E** | $15.09_{\pm2.30}$ | $11.48_{\pm1.13}$ | $8.29_{\pm1.38}$ | $24.74_{\pm3.25}$ | $4.47_{\pm0.31}$ | $1.78_{\pm0.38}$ | 10.97 |
| MPU | $17.54_{\pm1.45}$ | $12.96_{\pm1.71}$ | $6.29_{\pm0.84}$ | $35.50_{\pm2.66}$ | $5.06_{\pm0.43}$ | $1.64_{\pm0.20}$ | 13.16 |
| **MPU-E** | $19.29_{\pm2.51}$ | $14.51_{\pm1.59}$ | $7.73_{\pm1.46}$ | $38.24_{\pm2.09}$ | $6.50_{\pm0.71}$ | $2.19_{\pm0.41}$ | 14.74 |
| RADAR | $24.88_{\pm2.66}$ | $29.56_{\pm4.15}$ | $19.96_{\pm2.51}$ | $44.89_{\pm3.60}$ | $11.41_{\pm1.65}$ | $\mathbf{5.98_{\pm0.57}}$ | 22.78 |
| **RADAR-E** | $\mathbf{26.94_{\pm1.88}}$ | $\mathbf{30.30_{\pm2.71}}$ | $\mathbf{20.12_{\pm2.60}}$ | $\mathbf{50.01_{\pm3.58}}$ | $\mathbf{12.28_{\pm2.16}}$ | $5.60_{\pm0.98}$ | **24.21** |

Table 20: Performance concerning AUROC (%) on Essay under sentence-level settings. The detection model is trained on text generated by Dolly.

| Method | GPT4All | ChatGPT | ChatGPT-turbo | ChatGLM | Dolly | Claude | Avg. |
|---|---|---|---|---|---|---|---|
| Likelihood | $81.83_{\pm 0.40}$ | $82.98_{\pm 0.28}$ | $75.88_{\pm 0.46}$ | $89.59_{\pm 0.41}$ | $75.42_{\pm 0.33}$ | $62.62_{\pm 0.35}$ | 78.05 |
| Log-Rank | $80.82_{\pm 0.42}$ | $81.54_{\pm 0.34}$ | $73.73_{\pm 0.53}$ | $89.42_{\pm 0.38}$ | $74.27_{\pm 0.28}$ | $60.11_{\pm 0.42}$ | 76.65 |
| Entropy | $59.77_{\pm 0.44}$ | $64.67_{\pm 0.21}$ | $65.86_{\pm 0.26}$ | $60.94_{\pm 0.33}$ | $58.29_{\pm 0.43}$ | $59.10_{\pm 0.29}$ | 61.44 |
| NPR | $72.16_{\pm 0.47}$ | $71.46_{\pm 0.36}$ | $68.92_{\pm 0.74}$ | $77.78_{\pm 0.38}$ | $67.21_{\pm 0.59}$ | $64.14_{\pm 0.59}$ | 70.28 |
| DetectGPT | $73.01_{\pm 0.34}$ | $72.19_{\pm 0.33}$ | $70.72_{\pm 0.61}$ | $77.19_{\pm 0.30}$ | $68.59_{\pm 0.39}$ | $65.74_{\pm 0.55}$ | 71.24 |
| FastGPT | $81.65_{\pm 0.20}$ | $80.14_{\pm 0.40}$ | $71.65_{\pm 0.67}$ | $89.85_{\pm 0.26}$ | $77.29_{\pm 0.34}$ | $63.19_{\pm 0.20}$ | 77.29 |
| ChatGPT-D | $75.66_{\pm 0.52}$ | $74.66_{\pm 0.49}$ | $69.48_{\pm 0.79}$ | $82.52_{\pm 0.22}$ | $62.39_{\pm 0.46}$ | $56.44_{\pm 0.55}$ | 70.19 |
| **ChatGPT-E** | $75.59_{\pm 0.97}$ | $74.58_{\pm 0.46}$ | $68.87_{\pm 0.68}$ | $82.66_{\pm 1.31}$ | $62.44_{\pm 1.14}$ | $56.13_{\pm 0.24}$ | 70.04 |
| MPU | $82.92_{\pm 0.64}$ | $81.04_{\pm 0.35}$ | $75.68_{\pm 0.87}$ | $88.08_{\pm 0.27}$ | $70.79_{\pm 0.29}$ | $63.25_{\pm 1.32}$ | 76.96 |
| **MPU-E** | $84.53_{\pm 0.99}$ | $82.36_{\pm 0.77}$ | $77.78_{\pm 1.20}$ | $89.25_{\pm 0.91}$ | $72.94_{\pm 1.17}$ | $65.96_{\pm 2.20}$ | 78.80 |
| RADAR | $87.88_{\pm 0.89}$ | $88.76_{\pm 0.95}$ | $88.00_{\pm 0.82}$ | $91.28_{\pm 0.65}$ | $84.46_{\pm 0.24}$ | $82.59_{\pm 0.93}$ | 87.16 |
| **RADAR-E** | $\mathbf{88.66}_{\pm 0.58}$ | $\mathbf{89.71}_{\pm 0.57}$ | $\mathbf{89.20}_{\pm 0.40}$ | $\mathbf{92.19}_{\pm 0.44}$ | $\mathbf{85.57}_{\pm 0.26}$ | $\mathbf{84.12}_{\pm 0.81}$ | **88.24** |

Table 21: Performance concerning TPR@FPR-1% (%) on Essay under sentence-level settings. The detection model is trained on text generated by Dolly.

| Method | GPT4All | ChatGPT | ChatGPT-turbo | ChatGLM | Dolly | Claude | Avg. |
|---|---|---|---|---|---|---|---|
| Likelihood | $9.18_{\pm 1.56}$ | $14.05_{\pm 0.57}$ | $5.46_{\pm 0.46}$ | $26.04_{\pm 4.35}$ | $6.86_{\pm 1.13}$ | $2.82_{\pm 0.17}$ | 10.73 |
| Log-Rank | $8.98_{\pm 1.11}$ | $13.35_{\pm 0.58}$ | $4.97_{\pm 0.50}$ | $29.04_{\pm 3.44}$ | $6.78_{\pm 1.66}$ | $2.32_{\pm 0.05}$ | 10.91 |
| Entropy | $1.36_{\pm 0.26}$ | $2.40_{\pm 0.55}$ | $1.65_{\pm 0.34}$ | $1.10_{\pm 0.31}$ | $1.70_{\pm 0.43}$ | $1.44_{\pm 0.18}$ | 1.61 |
| NPR | $6.13_{\pm 0.66}$ | $7.13_{\pm 0.54}$ | $4.04_{\pm 0.74}$ | $14.14_{\pm 1.85}$ | $4.71_{\pm 0.51}$ | $3.26_{\pm 0.43}$ | 6.57 |
| DetectGPT | $4.11_{\pm 0.57}$ | $3.57_{\pm 0.37}$ | $3.48_{\pm 0.33}$ | $5.77_{\pm 1.47}$ | $3.70_{\pm 0.75}$ | $3.44_{\pm 0.17}$ | 4.01 |
| FastGPT | $12.38_{\pm 1.60}$ | $10.30_{\pm 0.26}$ | $4.58_{\pm 0.68}$ | $28.86_{\pm 2.71}$ | $9.01_{\pm 0.66}$ | $2.72_{\pm 0.43}$ | 11.31 |
| ChatGPT-D | $12.57_{\pm 0.93}$ | $9.75_{\pm 0.25}$ | $7.11_{\pm 1.04}$ | $19.86_{\pm 1.39}$ | $4.01_{\pm 0.33}$ | $1.70_{\pm 0.36}$ | 9.17 |
| **ChatGPT-E** | $12.39_{\pm 0.53}$ | $9.79_{\pm 0.81}$ | $6.72_{\pm 0.81}$ | $20.05_{\pm 2.04}$ | $3.96_{\pm 0.36}$ | $1.64_{\pm 0.27}$ | 9.09 |
| MPU | $14.01_{\pm 0.73}$ | $10.96_{\pm 0.76}$ | $6.52_{\pm 0.59}$ | $26.85_{\pm 1.55}$ | $4.22_{\pm 0.32}$ | $1.69_{\pm 0.18}$ | 10.71 |
| **MPU-E** | $16.41_{\pm 0.82}$ | $11.93_{\pm 0.62}$ | $7.93_{\pm 0.66}$ | $28.83_{\pm 1.15}$ | $5.00_{\pm 0.58}$ | $1.94_{\pm 0.18}$ | 12.01 |
| RADAR | $18.59_{\pm 3.01}$ | $21.01_{\pm 2.09}$ | $20.25_{\pm 2.77}$ | $26.77_{\pm 3.38}$ | $11.59_{\pm 1.25}$ | $9.84_{\pm 1.08}$ | 18.01 |
| **RADAR-E** | $\mathbf{21.47}_{\pm 1.17}$ | $\mathbf{23.22}_{\pm 1.71}$ | $\mathbf{22.34}_{\pm 1.80}$ | $\mathbf{31.89}_{\pm 1.67}$ | $\mathbf{14.24}_{\pm 1.15}$ | $\mathbf{10.17}_{\pm 1.57}$ | **20.56** |

Table 22: Performance concerning AUROC (%) on Essay under sentence-level settings. The detection model is trained on text generated by Claude.

| Method | GPT4All | ChatGPT | ChatGPT-turbo | ChatGLM | Dolly | Claude | Avg. |
|---|---|---|---|---|---|---|---|
| Likelihood | $81.83_{\pm 0.40}$ | $82.98_{\pm 0.28}$ | $75.88_{\pm 0.46}$ | $89.59_{\pm 0.41}$ | $75.42_{\pm 0.33}$ | $62.62_{\pm 0.35}$ | 78.05 |
| Log-Rank | $80.82_{\pm 0.42}$ | $81.54_{\pm 0.34}$ | $73.73_{\pm 0.53}$ | $89.42_{\pm 0.38}$ | $74.27_{\pm 0.28}$ | $60.11_{\pm 0.42}$ | 76.65 |
| Entropy | $59.77_{\pm 0.44}$ | $64.67_{\pm 0.21}$ | $65.86_{\pm 0.26}$ | $60.94_{\pm 0.33}$ | $58.29_{\pm 0.43}$ | $59.10_{\pm 0.29}$ | 61.44 |
| NPR | $72.16_{\pm 0.47}$ | $71.46_{\pm 0.36}$ | $68.92_{\pm 0.74}$ | $77.78_{\pm 0.38}$ | $67.21_{\pm 0.59}$ | $64.14_{\pm 0.59}$ | 70.28 |
| DetectGPT | $73.01_{\pm 0.34}$ | $72.19_{\pm 0.33}$ | $70.72_{\pm 0.61}$ | $77.19_{\pm 0.30}$ | $68.59_{\pm 0.39}$ | $65.74_{\pm 0.55}$ | 71.24 |
| FastGPT | $81.65_{\pm 0.20}$ | $80.14_{\pm 0.40}$ | $71.65_{\pm 0.67}$ | $89.85_{\pm 0.26}$ | $77.29_{\pm 0.34}$ | $63.19_{\pm 0.20}$ | 77.29 |
| ChatGPT-D | $73.73_{\pm 2.04}$ | $73.73_{\pm 1.65}$ | $71.41_{\pm 2.02}$ | $79.08_{\pm 1.73}$ | $61.09_{\pm 1.55}$ | $60.18_{\pm 2.22}$ | 69.87 |
| **ChatGPT-E** | $73.20_{\pm 1.12}$ | $73.35_{\pm 1.20}$ | $69.91_{\pm 0.82}$ | $79.13_{\pm 1.57}$ | $60.61_{\pm 1.43}$ | $58.82_{\pm 1.77}$ | 69.17 |
| MPU | $82.63_{\pm 0.63}$ | $81.39_{\pm 0.49}$ | $81.87_{\pm 0.42}$ | $85.69_{\pm 0.57}$ | $71.18_{\pm 0.86}$ | $78.29_{\pm 1.33}$ | 80.17 |
| **MPU-E** | $\mathbf{83.91}_{\pm 0.58}$ | $82.17_{\pm 0.49}$ | $83.81_{\pm 0.74}$ | $\mathbf{86.15}_{\pm 0.49}$ | $73.54_{\pm 0.94}$ | $82.53_{\pm 1.93}$ | 82.02 |
| RADAR | $82.33_{\pm 0.65}$ | $82.91_{\pm 0.71}$ | $88.76_{\pm 0.55}$ | $81.58_{\pm 0.58}$ | $79.47_{\pm 0.69}$ | $91.48_{\pm 0.35}$ | 84.42 |
| **RADAR-E** | $82.81_{\pm 0.58}$ | $\mathbf{83.36}_{\pm 0.82}$ | $\mathbf{89.27}_{\pm 0.50}$ | $82.05_{\pm 0.90}$ | $\mathbf{79.90}_{\pm 0.90}$ | $\mathbf{91.94}_{\pm 0.50}$ | **84.89** |

Table 23: Performance concerning TPR@FPR-1% (%) on Essay under sentence-level settings. The detection model is trained on text generated by Claude.

| Method | GPT4All | ChatGPT | ChatGPT-turbo | ChatGLM | Dolly | Claude | Avg. |
|---|---|---|---|---|---|---|---|
| Likelihood | $9.18_{\pm 1.56}$ | $14.05_{\pm 0.57}$ | $5.46_{\pm 0.46}$ | $26.04_{\pm 4.35}$ | $6.86_{\pm 1.13}$ | $2.82_{\pm 0.17}$ | 10.73 |
| Log-Rank | $8.98_{\pm 1.11}$ | $13.35_{\pm 0.58}$ | $4.97_{\pm 0.50}$ | $29.04_{\pm 3.44}$ | $6.78_{\pm 1.66}$ | $2.32_{\pm 0.05}$ | 10.91 |
| Entropy | $1.36_{\pm 0.26}$ | $2.40_{\pm 0.55}$ | $1.65_{\pm 0.34}$ | $1.10_{\pm 0.31}$ | $1.70_{\pm 0.43}$ | $1.44_{\pm 0.18}$ | 1.61 |
| NPR | $6.13_{\pm 0.66}$ | $7.13_{\pm 0.54}$ | $4.04_{\pm 0.74}$ | $14.14_{\pm 1.85}$ | $4.71_{\pm 0.51}$ | $3.26_{\pm 0.43}$ | 6.57 |
| DetectGPT | $4.11_{\pm 0.57}$ | $3.57_{\pm 0.37}$ | $3.48_{\pm 0.33}$ | $5.77_{\pm 1.47}$ | $3.70_{\pm 0.75}$ | $3.44_{\pm 0.17}$ | 4.01 |
| FastGPT | $12.38_{\pm 1.60}$ | $10.30_{\pm 0.26}$ | $4.58_{\pm 0.68}$ | $28.86_{\pm 2.71}$ | $9.01_{\pm 0.66}$ | $2.72_{\pm 0.43}$ | 11.31 |
| ChatGPT-D | $13.22_{\pm 1.45}$ | $9.76_{\pm 0.70}$ | $8.60_{\pm 1.55}$ | $18.60_{\pm 1.45}$ | $4.05_{\pm 0.42}$ | $2.24_{\pm 0.40}$ | 9.41 |
| **ChatGPT-E** | $12.31_{\pm 1.19}$ | $10.15_{\pm 1.02}$ | $7.51_{\pm 0.90}$ | $19.42_{\pm 2.11}$ | $3.93_{\pm 0.47}$ | $2.04_{\pm 0.31}$ | 9.23 |
| MPU | $14.80_{\pm 0.76}$ | $11.66_{\pm 0.49}$ | $12.58_{\pm 1.49}$ | $22.34_{\pm 1.54}$ | $4.33_{\pm 0.36}$ | $6.44_{\pm 1.01}$ | 12.02 |
| **MPU-E** | $\mathbf{16.14}_{\pm 1.28}$ | $\mathbf{12.52}_{\pm 1.40}$ | $16.27_{\pm 2.01}$ | $21.28_{\pm 2.34}$ | $5.09_{\pm 0.71}$ | $10.86_{\pm 2.81}$ | 13.70 |
| RADAR | $9.80_{\pm 1.96}$ | $10.87_{\pm 2.30}$ | $19.40_{\pm 3.50}$ | $6.90_{\pm 1.42}$ | $\mathbf{7.40}_{\pm 1.13}$ | $28.88_{\pm 2.72}$ | 13.87 |
| **RADAR-E** | $10.12_{\pm 1.64}$ | $11.30_{\pm 1.80}$ | $\mathbf{20.49}_{\pm 2.85}$ | $7.71_{\pm 1.61}$ | $7.28_{\pm 1.22}$ | $\mathbf{28.90}_{\pm 4.15}$ | **14.30** |

Table 24: Performance concerning AUROC (%) on Essay under paragraph-level settings. The detection model is trained on text generated by GPT4All.

| Method | GPT4All | ChatGPT | ChatGPT-turbo | ChatGLM | Dolly | Claude | Avg. |
|---|---|---|---|---|---|---|---|
| Likelihood | $96.16_{\pm0.30}$ | $98.79_{\pm0.19}$ | $99.13_{\pm0.19}$ | $99.29_{\pm0.25}$ | $90.90_{\pm1.33}$ | $92.76_{\pm0.23}$ | 96.17 |
| Log-Rank | $96.65_{\pm0.31}$ | $98.94_{\pm0.16}$ | $99.22_{\pm0.17}$ | $99.48_{\pm0.21}$ | $90.68_{\pm1.28}$ | $92.04_{\pm0.19}$ | 96.17 |
| Entropy | $75.52_{\pm1.51}$ | $90.33_{\pm0.21}$ | $94.52_{\pm0.46}$ | $85.25_{\pm0.75}$ | $74.76_{\pm1.45}$ | $86.21_{\pm0.67}$ | 84.43 |
| NPR | $97.77_{\pm0.21}$ | $99.16_{\pm0.12}$ | $47.82_{\pm1.01}$ | $99.55_{\pm0.08}$ | $95.23_{\pm0.54}$ | $49.50_{\pm0.67}$ | 81.50 |
| DetectGPT | $95.01_{\pm0.28}$ | $97.08_{\pm0.29}$ | $46.03_{\pm1.37}$ | $96.19_{\pm0.59}$ | $92.64_{\pm0.61}$ | $46.94_{\pm1.16}$ | 78.98 |
| FastGPT | $67.33_{\pm1.02}$ | $72.35_{\pm1.44}$ | $96.81_{\pm0.35}$ | $76.43_{\pm0.88}$ | $45.40_{\pm1.98}$ | $82.70_{\pm0.74}$ | 73.50 |
| ChatGPT-D | $94.74_{\pm1.24}$ | $95.29_{\pm0.93}$ | $89.62_{\pm1.61}$ | $99.05_{\pm0.17}$ | $71.41_{\pm3.35}$ | $52.35_{\pm6.00}$ | 83.74 |
| **ChatGPT-E** | $96.07_{\pm0.48}$ | $96.05_{\pm0.34}$ | $90.48_{\pm0.66}$ | $99.14_{\pm0.20}$ | $74.92_{\pm2.30}$ | $59.04_{\pm2.76}$ | 85.95 |
| MPU | $97.86_{\pm0.15}$ | $97.83_{\pm0.29}$ | $95.46_{\pm0.31}$ | $99.63_{\pm0.11}$ | $85.28_{\pm1.16}$ | $77.65_{\pm1.47}$ | 92.29 |
| **MPU-E** | $98.20_{\pm0.14}$ | $98.15_{\pm0.33}$ | $96.00_{\pm0.41}$ | $99.73_{\pm0.10}$ | $86.74_{\pm1.21}$ | $80.30_{\pm1.85}$ | 93.19 |
| RADAR | $99.60_{\pm0.07}$ | $99.36_{\pm0.10}$ | $98.32_{\pm0.27}$ | $99.89_{\pm0.05}$ | $93.58_{\pm0.58}$ | $95.75_{\pm0.58}$ | 97.75 |
| **RADAR-E** | $\mathbf{99.70_{\pm0.06}}$ | $\mathbf{99.51_{\pm0.12}}$ | $\mathbf{98.49_{\pm0.24}}$ | $\mathbf{99.93_{\pm0.03}}$ | $\mathbf{94.45_{\pm0.71}}$ | $\mathbf{97.12_{\pm0.38}}$ | **98.20** |

Table 25: Performance concerning AUROC (%) on Essay under paragraph-level settings. The detection model is trained on text generated by ChatGPT.

| Method | GPT4All | ChatGPT | ChatGPT-turbo | ChatGLM | Dolly | Claude | Avg. |
|---|---|---|---|---|---|---|---|
| Likelihood | $96.16_{\pm0.30}$ | $98.79_{\pm0.19}$ | $99.13_{\pm0.19}$ | $99.29_{\pm0.25}$ | $90.90_{\pm1.33}$ | $92.76_{\pm0.23}$ | 96.17 |
| Log-Rank | $96.65_{\pm0.31}$ | $98.94_{\pm0.16}$ | $99.22_{\pm0.17}$ | $99.48_{\pm0.21}$ | $90.68_{\pm1.28}$ | $92.04_{\pm0.19}$ | 96.17 |
| Entropy | $75.52_{\pm1.51}$ | $90.33_{\pm0.21}$ | $94.52_{\pm0.46}$ | $85.25_{\pm0.75}$ | $74.76_{\pm1.45}$ | $86.21_{\pm0.67}$ | 84.43 |
| NPR | $97.77_{\pm0.21}$ | $99.16_{\pm0.12}$ | $47.82_{\pm1.01}$ | $99.55_{\pm0.08}$ | $95.23_{\pm0.54}$ | $49.50_{\pm0.67}$ | 81.50 |
| DetectGPT | $95.01_{\pm0.28}$ | $97.08_{\pm0.29}$ | $46.03_{\pm1.37}$ | $96.19_{\pm0.59}$ | $92.64_{\pm0.61}$ | $46.94_{\pm1.16}$ | 78.98 |
| FastGPT | $67.33_{\pm1.02}$ | $72.35_{\pm1.44}$ | $96.81_{\pm0.35}$ | $76.43_{\pm0.88}$ | $45.40_{\pm1.98}$ | $82.70_{\pm0.74}$ | 73.5 |
| ChatGPT-D | $90.46_{\pm2.18}$ | $93.52_{\pm1.71}$ | $87.73_{\pm1.74}$ | $98.79_{\pm0.36}$ | $61.99_{\pm4.86}$ | $35.55_{\pm6.30}$ | 78.00 |
| **ChatGPT-E** | $91.14_{\pm2.17}$ | $94.17_{\pm1.43}$ | $89.20_{\pm1.66}$ | $98.68_{\pm0.67}$ | $63.31_{\pm4.82}$ | $37.87_{\pm6.00}$ | 79.06 |
| MPU | $96.90_{\pm0.51}$ | $97.78_{\pm0.33}$ | $95.10_{\pm0.82}$ | $99.56_{\pm0.16}$ | $82.09_{\pm2.53}$ | $70.68_{\pm3.87}$ | 90.35 |
| **MPU-E** | $97.35_{\pm0.37}$ | $98.11_{\pm0.28}$ | $95.76_{\pm0.68}$ | $99.61_{\pm0.13}$ | $83.89_{\pm1.63}$ | $74.72_{\pm3.03}$ | 91.57 |
| RADAR | $99.64_{\pm0.10}$ | $99.70_{\pm0.06}$ | $99.00_{\pm0.18}$ | $99.92_{\pm0.06}$ | $93.07_{\pm0.75}$ | $95.49_{\pm0.78}$ | 97.80 |
| **RADAR-E** | $\mathbf{99.68_{\pm0.09}}$ | $\mathbf{99.81_{\pm0.04}}$ | $\mathbf{99.21_{\pm0.12}}$ | $\mathbf{99.95_{\pm0.03}}$ | $\mathbf{93.67_{\pm0.87}}$ | $\mathbf{96.46_{\pm0.64}}$ | **98.13** |

Table 26: Performance concerning TPR@FPR-1% (%) on Essay under paragraph-level settings. The detection model is trained on text generated by ChatGPT.

| Method | GPT4All | ChatGPT | ChatGPT-turbo | ChatGLM | Dolly | Claude | Avg. |
|---|---|---|---|---|---|---|---|
| Likelihood | $46.33_{\pm16.49}$ | $68.62_{\pm13.32}$ | $73.60_{\pm14.63}$ | $92.86_{\pm4.84}$ | $20.67_{\pm10.79}$ | $12.36_{\pm6.32}$ | 52.41 |
| Log-Rank | $63.74_{\pm12.98}$ | $79.47_{\pm7.88}$ | $81.29_{\pm11.25}$ | $96.61_{\pm2.49}$ | $25.92_{\pm9.00}$ | $19.47_{\pm8.24}$ | 61.08 |
| Entropy | $3.78_{\pm0.91}$ | $11.07_{\pm2.00}$ | $16.31_{\pm1.79}$ | $8.35_{\pm3.44}$ | $3.91_{\pm1.40}$ | $6.22_{\pm1.85}$ | 8.27 |
| NPR | $78.50_{\pm3.99}$ | $85.91_{\pm1.56}$ | $9.02_{\pm0.57}$ | $95.13_{\pm1.73}$ | $58.52_{\pm7.27}$ | $8.40_{\pm1.17}$ | 55.91 |
| DetectGPT | $31.53_{\pm6.96}$ | $39.47_{\pm8.30}$ | $7.24_{\pm1.24}$ | $30.31_{\pm7.82}$ | $20.48_{\pm3.80}$ | $5.64_{\pm0.74}$ | 22.45 |
| FastGPT | $0.18_{\pm0.17}$ | $0.40_{\pm0.26}$ | $68.27_{\pm4.92}$ | $1.70_{\pm0.73}$ | $0.00_{\pm0.00}$ | $7.69_{\pm1.45}$ | 13.04 |
| ChatGPT-D | $52.35_{\pm6.34}$ | $44.00_{\pm4.87}$ | $31.20_{\pm2.66}$ | $84.02_{\pm4.65}$ | $12.84_{\pm4.11}$ | $1.20_{\pm0.64}$ | 37.60 |
| **ChatGPT-E** | $47.02_{\pm18.77}$ | $40.62_{\pm10.56}$ | $30.36_{\pm7.71}$ | $70.00_{\pm29.17}$ | $13.27_{\pm5.71}$ | $1.24_{\pm0.75}$ | 33.75 |
| MPU | $69.16_{\pm7.02}$ | $66.93_{\pm4.17}$ | $44.71_{\pm5.36}$ | $95.09_{\pm1.86}$ | $26.59_{\pm5.21}$ | $5.47_{\pm1.37}$ | 51.32 |
| **MPU-E** | $69.20_{\pm9.92}$ | $71.47_{\pm6.81}$ | $50.27_{\pm8.68}$ | $93.26_{\pm4.86}$ | $27.54_{\pm5.97}$ | $7.16_{\pm2.16}$ | 53.15 |
| RADAR | $92.94_{\pm2.53}$ | $92.44_{\pm1.69}$ | $72.31_{\pm8.51}$ | $98.48_{\pm0.96}$ | $50.36_{\pm9.08}$ | $47.38_{\pm5.11}$ | 75.65 |
| **RADAR-E** | $\mathbf{93.71_{\pm1.73}}$ | $\mathbf{95.87_{\pm1.19}}$ | $\mathbf{79.51_{\pm6.85}}$ | $\mathbf{99.20_{\pm0.70}}$ | $\mathbf{54.80_{\pm8.92}}$ | $\mathbf{55.56_{\pm5.05}}$ | **79.77** |

Table 27: Performance concerning AUROC (%) on Essay under paragraph-level settings. The detection model is trained on text generated by ChatGPT-turbo.

| Method | GPT4All | ChatGPT | ChatGPT-turbo | ChatGLM | Dolly | Claude | Avg. |
|---|---|---|---|---|---|---|---|
| Likelihood | $96.16_{\pm0.30}$ | $98.79_{\pm0.19}$ | $99.13_{\pm0.19}$ | $99.29_{\pm0.25}$ | $90.90_{\pm1.33}$ | $92.76_{\pm0.23}$ | 96.17 |
| Log-Rank | $96.65_{\pm0.31}$ | $98.94_{\pm0.16}$ | $99.22_{\pm0.17}$ | $99.48_{\pm0.21}$ | $90.68_{\pm1.28}$ | $92.04_{\pm0.19}$ | 96.17 |
| Entropy | $75.52_{\pm1.51}$ | $90.33_{\pm0.21}$ | $94.52_{\pm0.46}$ | $85.25_{\pm0.75}$ | $74.76_{\pm1.45}$ | $86.21_{\pm0.67}$ | 84.43 |
| NPR | $78.56_{\pm38.29}$ | $79.44_{\pm39.37}$ | $47.98_{\pm1.30}$ | $79.68_{\pm39.68}$ | $77.36_{\pm36.02}$ | $49.45_{\pm0.63}$ | 68.74 |
| DetectGPT | $76.83_{\pm36.14}$ | $78.13_{\pm37.75}$ | $46.78_{\pm2.70}$ | $77.54_{\pm37.09}$ | $75.83_{\pm33.94}$ | $47.33_{\pm1.90}$ | 67.07 |
| FastGPT | $67.33_{\pm1.02}$ | $72.35_{\pm1.44}$ | $96.81_{\pm0.35}$ | $76.43_{\pm0.88}$ | $45.40_{\pm1.98}$ | $82.70_{\pm0.74}$ | 73.50 |
| ChatGPT-D | $87.78_{\pm2.06}$ | $92.25_{\pm1.48}$ | $88.03_{\pm3.35}$ | $98.85_{\pm0.30}$ | $59.03_{\pm3.49}$ | $29.69_{\pm4.81}$ | 75.94 |
| **ChatGPT-E** | $88.73_{\pm2.11}$ | $93.76_{\pm0.69}$ | $90.58_{\pm1.47}$ | $98.85_{\pm0.17}$ | $59.50_{\pm3.30}$ | $30.99_{\pm4.41}$ | 77.07 |
| MPU | $96.39_{\pm0.45}$ | $97.44_{\pm0.33}$ | $97.02_{\pm0.61}$ | $99.46_{\pm0.15}$ | $80.87_{\pm2.00}$ | $68.71_{\pm5.39}$ | 89.98 |
| **MPU-E** | $97.18_{\pm0.42}$ | $97.96_{\pm0.24}$ | $97.74_{\pm0.46}$ | $99.58_{\pm0.14}$ | $83.69_{\pm1.75}$ | $74.41_{\pm4.57}$ | 91.76 |
| RADAR | $99.37_{\pm0.06}$ | $99.43_{\pm0.14}$ | $99.51_{\pm0.09}$ | $99.80_{\pm0.08}$ | $92.65_{\pm0.53}$ | $93.23_{\pm0.54}$ | 97.33 |
| **RADAR-E** | $\mathbf{99.43_{\pm0.07}}$ | $\mathbf{99.56_{\pm0.14}}$ | $\mathbf{99.66_{\pm0.08}}$ | $\mathbf{99.87_{\pm0.04}}$ | $\mathbf{93.30_{\pm0.48}}$ | $\mathbf{94.44_{\pm0.82}}$ | **97.71** |

Table 28: Performance concerning TPR@FPR-1% (%) on Essay under paragraph-level settings. The detection model is trained on text generated by ChatGPT-turbo.

| Method | GPT4All | ChatGPT | ChatGPT-turbo | ChatGLM | Dolly | Claude | Avg. |
|---|---|---|---|---|---|---|---|
| Likelihood | $46.33_{\pm16.49}$ | $68.62_{\pm13.32}$ | $73.60_{\pm14.63}$ | $92.86_{\pm4.84}$ | $20.67_{\pm10.79}$ | $12.36_{\pm6.32}$ | 52.41 |
| Log-Rank | $63.74_{\pm12.98}$ | $79.47_{\pm7.88}$ | $81.29_{\pm11.25}$ | $96.61_{\pm2.49}$ | $25.92_{\pm9.00}$ | $19.47_{\pm8.24}$ | 61.08 |
| Entropy | $3.78_{\pm0.91}$ | $11.07_{\pm2.00}$ | $16.31_{\pm1.79}$ | $8.35_{\pm3.44}$ | $3.91_{\pm1.40}$ | $6.22_{\pm1.85}$ | 8.27 |
| NPR | $63.28_{\pm31.87}$ | $68.80_{\pm34.43}$ | $7.29_{\pm3.68}$ | $75.76_{\pm37.91}$ | $46.54_{\pm24.25}$ | $6.31_{\pm3.21}$ | 44.66 |
| DetectGPT | $23.23_{\pm12.60}$ | $32.80_{\pm18.12}$ | $5.73_{\pm3.12}$ | $24.64_{\pm14.56}$ | $16.66_{\pm9.13}$ | $4.31_{\pm2.22}$ | 17.90 |
| FastGPT | $0.18_{\pm0.17}$ | $0.40_{\pm0.26}$ | $68.27_{\pm4.92}$ | $1.70_{\pm0.73}$ | $0.00_{\pm0.00}$ | $7.69_{\pm1.45}$ | 13.04 |
| ChatGPT-D | $53.17_{\pm7.00}$ | $44.89_{\pm1.41}$ | $35.24_{\pm4.20}$ | $81.38_{\pm2.54}$ | $16.56_{\pm6.48}$ | $1.11_{\pm0.63}$ | 38.73 |
| **ChatGPT-E** | $54.03_{\pm4.84}$ | $45.64_{\pm3.52}$ | $39.82_{\pm4.48}$ | $81.12_{\pm2.22}$ | $15.04_{\pm3.37}$ | $1.29_{\pm0.57}$ | 39.49 |
| MPU | $68.20_{\pm6.85}$ | $69.82_{\pm7.34}$ | $61.82_{\pm7.1}$ | $95.00_{\pm0.56}$ | $25.16_{\pm4.17}$ | $4.71_{\pm1.37}$ | 54.12 |
| **MPU-E** | $70.02_{\pm9.61}$ | $71.56_{\pm7.78}$ | $65.02_{\pm8.97}$ | $95.58_{\pm0.52}$ | $27.83_{\pm6.39}$ | $6.22_{\pm1.87}$ | 56.04 |
| RADAR | $86.97_{\pm3.00}$ | $86.36_{\pm3.32}$ | $86.04_{\pm3.02}$ | $95.18_{\pm1.58}$ | $45.39_{\pm9.18}$ | $35.07_{\pm5.81}$ | 72.50 |
| **RADAR-E** | $\mathbf{87.33}_{\pm2.62}$ | $\mathbf{89.87}_{\pm3.23}$ | $\mathbf{90.31}_{\pm2.64}$ | $\mathbf{96.96}_{\pm0.64}$ | $\mathbf{50.21}_{\pm5.61}$ | $\mathbf{39.56}_{\pm6.83}$ | **75.71** |

Table 29: Performance concerning AUROC (%) on Essay under paragraph-level settings. The detection model is trained on text generated by ChatGLM.

| Method | GPT4All | ChatGPT | ChatGPT-turbo | ChatGLM | Dolly | Claude | Avg. |
|---|---|---|---|---|---|---|---|
| Likelihood | $96.16_{\pm0.30}$ | $98.79_{\pm0.19}$ | $99.13_{\pm0.19}$ | $99.29_{\pm0.25}$ | $90.90_{\pm1.33}$ | $92.76_{\pm0.23}$ | 96.17 |
| Log-Rank | $96.65_{\pm0.31}$ | $98.94_{\pm0.16}$ | $\mathbf{99.22}_{\pm0.17}$ | $99.48_{\pm0.21}$ | $90.68_{\pm1.28}$ | $92.04_{\pm0.19}$ | 96.17 |
| Entropy | $75.52_{\pm1.51}$ | $90.33_{\pm0.21}$ | $94.52_{\pm0.46}$ | $85.25_{\pm0.75}$ | $74.76_{\pm1.45}$ | $86.21_{\pm0.67}$ | 84.43 |
| NPR | $97.77_{\pm0.21}$ | $99.16_{\pm0.12}$ | $47.82_{\pm1.01}$ | $99.55_{\pm0.08}$ | $95.23_{\pm0.54}$ | $49.50_{\pm0.67}$ | 81.50 |
| DetectGPT | $95.01_{\pm0.28}$ | $97.08_{\pm0.29}$ | $46.03_{\pm1.37}$ | $96.19_{\pm0.59}$ | $92.64_{\pm0.61}$ | $46.94_{\pm1.16}$ | 78.98 |
| FastGPT | $67.33_{\pm1.02}$ | $72.35_{\pm1.44}$ | $96.81_{\pm0.35}$ | $76.43_{\pm0.88}$ | $45.40_{\pm1.98}$ | $82.70_{\pm0.74}$ | 73.50 |
| ChatGPT-D | $87.52_{\pm2.41}$ | $92.28_{\pm1.54}$ | $86.68_{\pm2.15}$ | $99.03_{\pm0.35}$ | $59.47_{\pm3.06}$ | $28.59_{\pm3.53}$ | 75.60 |
| **ChatGPT-E** | $87.30_{\pm2.02}$ | $92.22_{\pm1.03}$ | $87.14_{\pm1.47}$ | $99.16_{\pm0.29}$ | $59.72_{\pm2.99}$ | $29.45_{\pm3.11}$ | 75.83 |
| MPU | $96.51_{\pm0.27}$ | $97.20_{\pm0.20}$ | $93.34_{\pm0.83}$ | $99.75_{\pm0.09}$ | $80.68_{\pm1.12}$ | $64.40_{\pm1.68}$ | 88.65 |
| **MPU-E** | $96.94_{\pm0.24}$ | $97.51_{\pm0.27}$ | $93.88_{\pm0.71}$ | $99.81_{\pm0.05}$ | $82.06_{\pm1.26}$ | $66.71_{\pm1.32}$ | 89.48 |
| RADAR | $99.53_{\pm0.06}$ | $99.45_{\pm0.16}$ | $98.44_{\pm0.47}$ | $99.93_{\pm0.03}$ | $92.65_{\pm0.60}$ | $94.47_{\pm0.56}$ | 97.41 |
| **RADAR-E** | $\mathbf{99.60}_{\pm0.07}$ | $\mathbf{99.58}_{\pm0.12}$ | $98.57_{\pm0.49}$ | $\mathbf{99.95}_{\pm0.02}$ | $\mathbf{93.30}_{\pm0.67}$ | $\mathbf{95.47}_{\pm0.58}$ | **97.74** |

Table 30: Performance concerning TPR@FPR-1% (%) on Essay under paragraph-level settings. The detection model is trained on text generated by ChatGLM.

| Method | GPT4All | ChatGPT | ChatGPT-turbo | ChatGLM | Dolly | Claude | Avg. |
|---|---|---|---|---|---|---|---|
| Likelihood | $46.33_{\pm16.49}$ | $68.62_{\pm13.32}$ | $73.60_{\pm14.63}$ | $92.86_{\pm4.84}$ | $20.67_{\pm10.79}$ | $12.36_{\pm6.32}$ | 52.41 |
| Log-Rank | $63.74_{\pm12.98}$ | $79.47_{\pm7.88}$ | $\mathbf{81.29}_{\pm11.25}$ | $96.61_{\pm2.49}$ | $25.92_{\pm9.00}$ | $19.47_{\pm8.24}$ | 61.08 |
| Entropy | $3.78_{\pm0.91}$ | $11.07_{\pm2.00}$ | $16.31_{\pm1.79}$ | $8.35_{\pm3.44}$ | $3.91_{\pm1.40}$ | $6.22_{\pm1.85}$ | 8.27 |
| NPR | $78.50_{\pm3.99}$ | $85.91_{\pm1.56}$ | $9.02_{\pm0.57}$ | $95.13_{\pm1.73}$ | $58.52_{\pm7.27}$ | $8.40_{\pm1.17}$ | 55.91 |
| DetectGPT | $31.53_{\pm6.96}$ | $39.47_{\pm8.30}$ | $7.24_{\pm1.24}$ | $30.31_{\pm7.82}$ | $20.48_{\pm3.80}$ | $5.64_{\pm0.74}$ | 22.45 |
| FastGPT | $0.18_{\pm0.17}$ | $0.40_{\pm0.26}$ | $68.27_{\pm4.92}$ | $1.70_{\pm0.73}$ | $0.00_{\pm0.00}$ | $7.69_{\pm1.45}$ | 13.04 |
| ChatGPT-D | $53.99_{\pm7.42}$ | $45.42_{\pm10.79}$ | $30.04_{\pm4.88}$ | $87.37_{\pm4.14}$ | $15.75_{\pm3.83}$ | $1.38_{\pm0.71}$ | 38.99 |
| **ChatGPT-E** | $53.99_{\pm4.66}$ | $47.69_{\pm10.47}$ | $30.67_{\pm4.68}$ | $89.02_{\pm3.70}$ | $15.47_{\pm3.50}$ | $1.56_{\pm0.85}$ | 39.73 |
| MPU | $69.07_{\pm5.41}$ | $65.96_{\pm5.11}$ | $43.07_{\pm6.73}$ | $96.07_{\pm0.79}$ | $25.73_{\pm4.87}$ | $3.56_{\pm1.33}$ | 50.57 |
| **MPU-E** | $72.71_{\pm4.55}$ | $70.67_{\pm3.19}$ | $47.24_{\pm4.22}$ | $97.01_{\pm0.52}$ | $27.59_{\pm4.22}$ | $4.22_{\pm1.11}$ | 53.24 |
| RADAR | $90.52_{\pm1.19}$ | $88.71_{\pm4.31}$ | $63.16_{\pm13.88}$ | $98.26_{\pm0.95}$ | $46.01_{\pm7.91}$ | $42.67_{\pm8.84}$ | 71.56 |
| **RADAR-E** | $\mathbf{92.44}_{\pm1.17}$ | $\mathbf{90.71}_{\pm3.85}$ | $65.38_{\pm13.65}$ | $\mathbf{98.53}_{\pm1.14}$ | $48.45_{\pm8.19}$ | $47.51_{\pm9.11}$ | **73.84** |

Table 31: Performance concerning AUROC (%) on Essay under paragraph-level settings. The detection model is trained on text generated by Dolly.

| Method | GPT4All | ChatGPT | ChatGPT-turbo | ChatGLM | Dolly | Claude | Avg. |
|---|---|---|---|---|---|---|---|
| Likelihood | $96.16_{\pm0.30}$ | $98.79_{\pm0.19}$ | $99.13_{\pm0.19}$ | $99.29_{\pm0.25}$ | $90.90_{\pm1.33}$ | $92.76_{\pm0.23}$ | 96.17 |
| Log-Rank | $96.65_{\pm0.31}$ | $98.94_{\pm0.16}$ | $\mathbf{99.22}_{\pm0.17}$ | $99.48_{\pm0.21}$ | $90.68_{\pm1.28}$ | $92.04_{\pm0.19}$ | 96.17 |
| Entropy | $75.52_{\pm1.51}$ | $90.33_{\pm0.21}$ | $94.52_{\pm0.46}$ | $85.25_{\pm0.75}$ | $74.76_{\pm1.45}$ | $86.21_{\pm0.67}$ | 84.43 |
| NPR | $97.77_{\pm0.21}$ | $99.16_{\pm0.12}$ | $47.82_{\pm1.01}$ | $99.55_{\pm0.08}$ | $95.23_{\pm0.54}$ | $49.50_{\pm0.67}$ | 81.50 |
| DetectGPT | $95.01_{\pm0.28}$ | $97.08_{\pm0.29}$ | $46.03_{\pm1.37}$ | $96.19_{\pm0.59}$ | $92.64_{\pm0.61}$ | $46.94_{\pm1.16}$ | 78.98 |
| FastGPT | $67.33_{\pm1.02}$ | $72.35_{\pm1.44}$ | $96.81_{\pm0.35}$ | $76.43_{\pm0.88}$ | $45.40_{\pm1.98}$ | $82.70_{\pm0.74}$ | 73.50 |
| ChatGPT-D | $93.43_{\pm0.85}$ | $92.97_{\pm1.31}$ | $86.10_{\pm2.08}$ | $98.81_{\pm0.10}$ | $71.61_{\pm2.43}$ | $49.83_{\pm2.51}$ | 82.13 |
| **ChatGPT-E** | $93.94_{\pm1.20}$ | $94.18_{\pm0.69}$ | $88.13_{\pm1.39}$ | $98.57_{\pm0.20}$ | $72.10_{\pm4.49}$ | $53.01_{\pm5.66}$ | 83.32 |
| MPU | $97.07_{\pm0.40}$ | $97.62_{\pm0.29}$ | $94.71_{\pm0.79}$ | $99.63_{\pm0.21}$ | $87.51_{\pm0.78}$ | $78.44_{\pm2.09}$ | 92.50 |
| **MPU-E** | $97.35_{\pm0.54}$ | $97.82_{\pm0.50}$ | $95.25_{\pm1.04}$ | $99.58_{\pm0.23}$ | $88.71_{\pm1.25}$ | $81.48_{\pm3.28}$ | 93.37 |
| RADAR | $99.54_{\pm0.13}$ | $99.10_{\pm0.31}$ | $98.36_{\pm0.50}$ | $99.79_{\pm0.11}$ | $96.08_{\pm0.39}$ | $97.81_{\pm0.49}$ | 98.45 |
| **RADAR-E** | $\mathbf{99.64}_{\pm0.10}$ | $\mathbf{99.18}_{\pm0.24}$ | $98.35_{\pm0.43}$ | $\mathbf{99.83}_{\pm0.08}$ | $\mathbf{96.66}_{\pm0.34}$ | $\mathbf{98.38}_{\pm0.37}$ | **98.67** |

Table 32: Performance concerning TPR@FPR-1% (%) on Essay under paragraph-level settings. The detection model is trained on text generated by Dolly.

| Method | GPT4All | ChatGPT | ChatGPT-turbo | ChatGLM | Dolly | Claude | Avg. |
|---|---|---|---|---|---|---|---|
| Likelihood | $46.33_{\pm 16.49}$ | $68.62_{\pm 13.32}$ | $73.60_{\pm 14.63}$ | $92.86_{\pm 4.84}$ | $20.67_{\pm 10.79}$ | $12.36_{\pm 6.32}$ | 52.41 |
| Log-Rank | $63.74_{\pm 12.98}$ | $79.47_{\pm 7.88}$ | $\mathbf{81.29}_{\pm 11.25}$ | $\mathbf{96.61}_{\pm 2.49}$ | $25.92_{\pm 9.00}$ | $19.47_{\pm 8.24}$ | 61.08 |
| Entropy | $3.78_{\pm 0.91}$ | $11.07_{\pm 2.00}$ | $16.31_{\pm 1.79}$ | $8.35_{\pm 3.44}$ | $3.91_{\pm 1.40}$ | $6.22_{\pm 1.85}$ | 8.27 |
| NPR | $78.50_{\pm 3.99}$ | $\mathbf{85.91}_{\pm 1.56}$ | $9.02_{\pm 0.57}$ | $95.13_{\pm 1.73}$ | $58.52_{\pm 7.27}$ | $8.40_{\pm 1.17}$ | 55.91 |
| DetectGPT | $31.53_{\pm 6.96}$ | $39.47_{\pm 8.30}$ | $7.24_{\pm 1.24}$ | $30.31_{\pm 7.82}$ | $20.48_{\pm 3.80}$ | $5.64_{\pm 0.74}$ | 22.45 |
| FastGPT | $0.18_{\pm 0.17}$ | $0.40_{\pm 0.26}$ | $68.27_{\pm 4.92}$ | $1.70_{\pm 0.73}$ | $0.00_{\pm 0.00}$ | $7.69_{\pm 1.45}$ | 13.04 |
| ChatGPT-D | $51.57_{\pm 2.43}$ | $42.18_{\pm 4.93}$ | $30.49_{\pm 3.07}$ | $82.37_{\pm 1.16}$ | $14.80_{\pm 2.73}$ | $1.56_{\pm 1.17}$ | 37.16 |
| **ChatGPT-E** | $50.71_{\pm 2.40}$ | $39.42_{\pm 6.67}$ | $30.09_{\pm 4.12}$ | $79.60_{\pm 2.87}$ | $12.94_{\pm 2.14}$ | $1.91_{\pm 0.87}$ | 35.78 |
| MPU | $70.75_{\pm 7.12}$ | $69.87_{\pm 4.10}$ | $48.58_{\pm 5.84}$ | $95.49_{\pm 2.38}$ | $32.55_{\pm 4.65}$ | $7.91_{\pm 3.11}$ | 54.19 |
| **MPU-E** | $65.24_{\pm 16.68}$ | $63.69_{\pm 10.27}$ | $43.11_{\pm 11.50}$ | $94.64_{\pm 3.23}$ | $31.07_{\pm 9.56}$ | $8.53_{\pm 4.58}$ | 51.05 |
| RADAR | $91.48_{\pm 3.80}$ | $79.11_{\pm 7.76}$ | $55.82_{\pm 9.60}$ | $96.21_{\pm 2.13}$ | $62.58_{\pm 6.52}$ | $59.42_{\pm 9.76}$ | 74.10 |
| **RADAR-E** | $\mathbf{92.12}_{\pm 4.11}$ | $81.07_{\pm 4.33}$ | $55.33_{\pm 6.25}$ | $96.21_{\pm 1.39}$ | $\mathbf{63.48}_{\pm 6.94}$ | $\mathbf{66.13}_{\pm 9.38}$ | **75.72** |

Table 33: Performance concerning AUROC (%) on Essay under paragraph-level settings. The detection model is trained on text generated by Claude.

| Method | GPT4All | ChatGPT | ChatGPT-turbo | ChatGLM | Dolly | Claude | Avg. |
|---|---|---|---|---|---|---|---|
| Likelihood | $96.16_{\pm 0.30}$ | $98.79_{\pm 0.19}$ | $99.13_{\pm 0.19}$ | $99.29_{\pm 0.25}$ | $90.90_{\pm 1.33}$ | $92.76_{\pm 0.23}$ | 96.17 |
| Log-Rank | $96.65_{\pm 0.31}$ | $98.94_{\pm 0.16}$ | $\mathbf{99.22}_{\pm 0.17}$ | $99.48_{\pm 0.21}$ | $90.68_{\pm 1.28}$ | $92.04_{\pm 0.19}$ | 96.17 |
| Entropy | $75.52_{\pm 1.51}$ | $90.33_{\pm 0.21}$ | $94.52_{\pm 0.46}$ | $85.25_{\pm 0.75}$ | $74.76_{\pm 1.45}$ | $86.21_{\pm 0.67}$ | 84.43 |
| NPR | $78.56_{\pm 38.29}$ | $79.44_{\pm 39.37}$ | $47.98_{\pm 1.30}$ | $79.68_{\pm 39.68}$ | $77.36_{\pm 36.02}$ | $49.45_{\pm 0.63}$ | 68.74 |
| DetectGPT | $58.99_{\pm 44.10}$ | $59.30_{\pm 46.15}$ | $48.24_{\pm 3.82}$ | $59.08_{\pm 45.29}$ | $58.54_{\pm 41.78}$ | $48.86_{\pm 3.07}$ | 55.50 |
| FastGPT | $67.33_{\pm 1.02}$ | $72.35_{\pm 1.44}$ | $96.81_{\pm 0.35}$ | $76.43_{\pm 0.88}$ | $45.40_{\pm 1.98}$ | $82.70_{\pm 0.74}$ | 73.50 |
| ChatGPT-D | $93.43_{\pm 1.96}$ | $93.40_{\pm 1.49}$ | $87.13_{\pm 1.81}$ | $98.72_{\pm 0.31}$ | $70.72_{\pm 4.37}$ | $52.88_{\pm 8.49}$ | 82.71 |
| **ChatGPT-E** | $93.80_{\pm 1.48}$ | $94.66_{\pm 1.01}$ | $88.86_{\pm 1.59}$ | $98.55_{\pm 0.35}$ | $70.61_{\pm 2.85}$ | $56.46_{\pm 9.11}$ | 83.82 |
| MPU | $97.21_{\pm 0.24}$ | $97.86_{\pm 0.25}$ | $96.36_{\pm 0.47}$ | $99.61_{\pm 0.20}$ | $86.90_{\pm 0.85}$ | $86.70_{\pm 1.30}$ | 94.11 |
| **MPU-E** | $97.51_{\pm 0.21}$ | $98.11_{\pm 0.29}$ | $96.98_{\pm 0.42}$ | $99.59_{\pm 0.20}$ | $88.27_{\pm 1.46}$ | $89.24_{\pm 1.49}$ | 94.95 |
| RADAR | $99.70_{\pm 0.09}$ | $99.49_{\pm 0.22}$ | $98.59_{\pm 0.49}$ | $99.92_{\pm 0.04}$ | $96.14_{\pm 0.21}$ | $99.38_{\pm 0.14}$ | 98.87 |
| **RADAR-E** | $\mathbf{99.74}_{\pm 0.08}$ | $\mathbf{99.56}_{\pm 0.20}$ | $98.55_{\pm 0.55}$ | $\mathbf{99.94}_{\pm 0.03}$ | $\mathbf{96.55}_{\pm 0.25}$ | $\mathbf{99.59}_{\pm 0.10}$ | **98.99** |

Table 34: Performance concerning TPR@FPR-1% (%) on Essay under paragraph-level settings. The detection model is trained on text generated by Claude.

| Method | GPT4All | ChatGPT | ChatGPT-turbo | ChatGLM | Dolly | Claude | Avg. |
|---|---|---|---|---|---|---|---|
| Likelihood | $46.33_{\pm 16.49}$ | $68.62_{\pm 13.32}$ | $73.60_{\pm 14.63}$ | $92.86_{\pm 4.84}$ | $20.67_{\pm 10.79}$ | $12.36_{\pm 6.32}$ | 52.41 |
| Log-Rank | $63.74_{\pm 12.98}$ | $79.47_{\pm 7.88}$ | $\mathbf{81.29}_{\pm 11.25}$ | $96.61_{\pm 2.49}$ | $25.92_{\pm 9.00}$ | $19.47_{\pm 8.24}$ | 61.08 |
| Entropy | $3.78_{\pm 0.91}$ | $11.07_{\pm 2.00}$ | $16.31_{\pm 1.79}$ | $8.35_{\pm 3.44}$ | $3.91_{\pm 1.40}$ | $6.22_{\pm 1.85}$ | 8.27 |
| NPR | $63.28_{\pm 31.87}$ | $68.80_{\pm 34.43}$ | $7.29_{\pm 3.68}$ | $75.76_{\pm 37.91}$ | $46.54_{\pm 24.25}$ | $6.31_{\pm 3.21}$ | 44.66 |
| DetectGPT | $19.23_{\pm 15.77}$ | $25.87_{\pm 22.24}$ | $4.40_{\pm 3.79}$ | $19.02_{\pm 17.30}$ | $12.65_{\pm 10.97}$ | $3.11_{\pm 2.58}$ | 14.05 |
| FastGPT | $0.18_{\pm 0.17}$ | $0.40_{\pm 0.26}$ | $68.27_{\pm 4.92}$ | $1.70_{\pm 0.73}$ | $0.00_{\pm 0.00}$ | $7.69_{\pm 1.45}$ | 13.04 |
| ChatGPT-D | $53.99_{\pm 3.98}$ | $45.73_{\pm 6.02}$ | $32.67_{\pm 3.60}$ | $83.12_{\pm 5.40}$ | $14.70_{\pm 2.04}$ | $2.49_{\pm 2.27}$ | 38.78 |
| **ChatGPT-E** | $50.80_{\pm 6.76}$ | $42.58_{\pm 9.09}$ | $32.40_{\pm 4.47}$ | $79.42_{\pm 7.56}$ | $13.17_{\pm 1.56}$ | $2.71_{\pm 2.14}$ | 36.85 |
| MPU | $71.48_{\pm 5.19}$ | $72.40_{\pm 5.98}$ | $52.62_{\pm 7.02}$ | $96.52_{\pm 1.79}$ | $34.37_{\pm 6.93}$ | $14.58_{\pm 4.15}$ | 56.99 |
| **MPU-E** | $75.17_{\pm 5.85}$ | $70.27_{\pm 6.48}$ | $51.82_{\pm 7.75}$ | $95.71_{\pm 2.18}$ | $33.17_{\pm 7.76}$ | $16.22_{\pm 6.14}$ | 57.06 |
| RADAR | $\mathbf{95.40}_{\pm 1.87}$ | $90.09_{\pm 5.02}$ | $64.84_{\pm 15.09}$ | $98.48_{\pm 0.74}$ | $61.77_{\pm 6.46}$ | $91.73_{\pm 3.01}$ | 83.72 |
| **RADAR-E** | $95.08_{\pm 1.79}$ | $\mathbf{91.24}_{\pm 4.64}$ | $64.31_{\pm 15.02}$ | $\mathbf{98.84}_{\pm 0.59}$ | $\mathbf{62.86}_{\pm 6.09}$ | $\mathbf{94.27}_{\pm 2.02}$ | **84.43** |

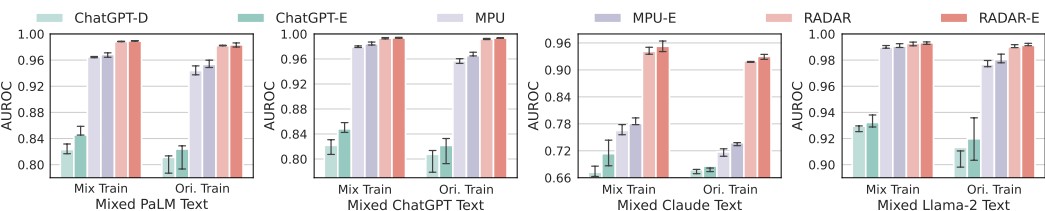

Figure 12: Test performance (AUROC) under various LLM mixed texts. Detectors are trained on text generated by PaLM. For each sub-figure, the left group: detectors are trained on mixed text, and the right group: detectors are trained on original text.

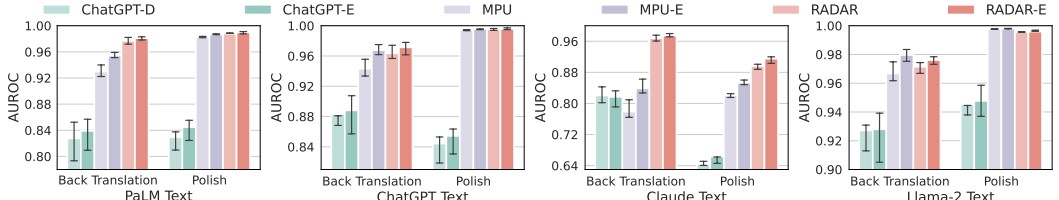

Figure 13: Robustness (AUROC) against paraphrasing attacks (Back Translation and Polish). Detectors are trained on the PaLM texts and tested on the paraphrasing texts of various LLMs.

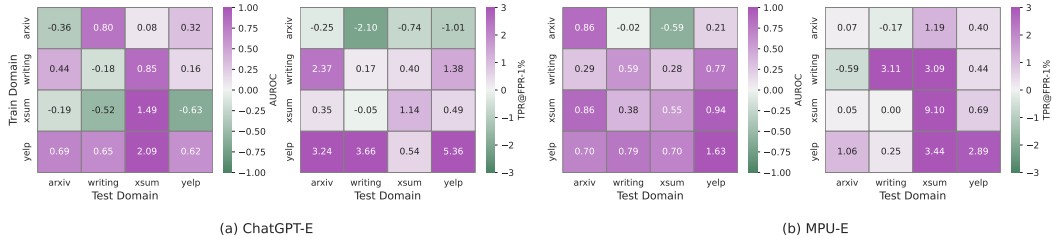

Figure 14: The performance improvement of the enhanced ChatGPT-E and MPU-E compared to ChatGPT-D and MPU. The figure shows their differences for AUROC and TPR@FPR-1. Positive values indicate performance improvement.

Table 35: Performance on newer LLMs. The detection model is trained on text generated by GPT4All.

| Method | TPR@FPR-1% | | | | | AUROC | | | | |
|---|---|---|---|---|---|---|---|---|---|---|
| | GPT-4o | GPT-4.1 | DeepSeek-R1 | Llama4 Maverick | Avg. | GPT-4o | GPT-4.1 | DeepSeek-R1 | Llama4 Maverick | Avg. |
| ChatGPT-D | 2.98 | 1.20 | 0.00 | 45.87 | 12.51 | 54.94 | 36.48 | 9.37 | 95.48 | 49.07 |
| **ChatGPT-E** | **3.47** | **1.33** | 0.00 | **47.91** | **13.18** | **57.68** | **38.14** | **12.00** | **95.90** | **50.93** |
| MPU | 6.76 | 3.07 | **0.13** | 59.60 | 17.39 | 76.37 | 63.57 | 40.98 | 97.20 | 69.53 |
| **MPU-E** | **8.53** | **3.64** | 0.09 | **62.18** | **18.61** | **79.02** | **66.81** | **44.84** | **97.54** | **72.05** |
| RADAR | 66.67 | 50.53 | 55.78 | 86.09 | 64.77 | 98.79 | 97.27 | 98.55 | 99.50 | 98.53 |
| **RADAR-E** | **77.47** | **63.16** | **68.53** | **91.56** | **75.18** | **99.18** | **98.09** | **99.01** | **99.67** | **98.99** |

Table 36: Running time under sentence-level setting.

| Sentence-level | Training Time | | Test Time | |
|---|---|---|---|---|
| | Essay | DetectRL | Essay | DetectRL |
| Likelihood | 42.8 | 47.5 | 192.3 | 213.8 |
| Log-Rank | 43.9 | 47.8 | 197.6 | 215.1 |
| Entropy | 42.6 | 47.0 | 190.5 | 213.3 |
| NPR | 1485.9 | 2396.7 | 6680.3 | 10774.8 |
| DetectGPT | 1769.6 | 2306.0 | 7955.6 | 10367.1 |
| FastGPT | 106.7 | 116.2 | 480.0 | 522.5 |
| ChatGPT-D | 28.7 | 29.5 | 5.9 | 6.3 |
| **ChatGPT-E** | 30.9 | 30.9 | 5.9 | 6.3 |
| MPU | 27.3 | 29.6 | 5.8 | 6.3 |
| **MPU-E** | 29.8 | 30.1 | 5.8 | 6.3 |
| RADAR | 76.7 | 78.2 | 16.9 | 18.4 |
| **RADAR-E** | 78.4 | 81.6 | 17.1 | 18.3 |

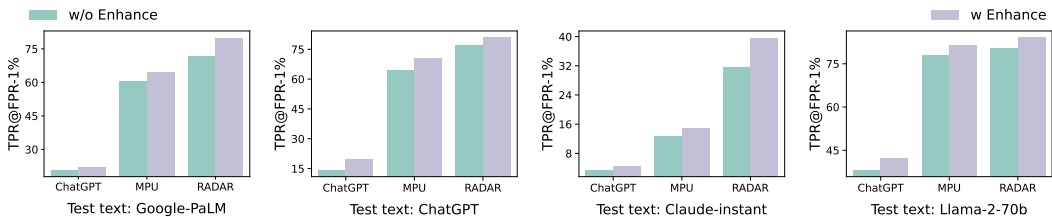

Figure 15: The Impact of noise label regarding TPR-FPR-1% on DetectRL dataset.

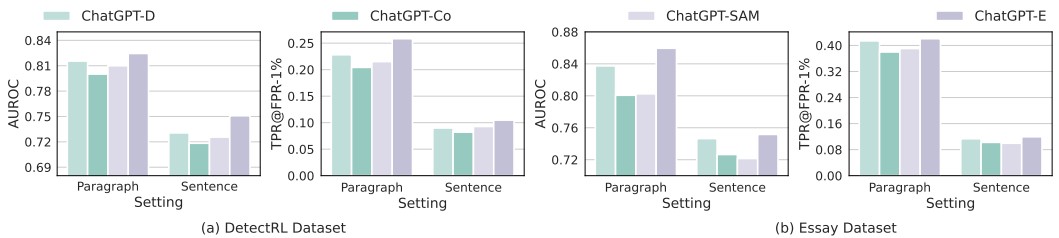

Figure 16: Performance comparison (AUROC and TPR@FPR-1%) with NLL methods. The version applying Co-teaching and SAM to ChatGPT-D is denoted as ChatGPT-Co and ChatGPT-SAM. The supervisor is trained on PaLM texts on DetectRL and GPT4All texts on Essay.

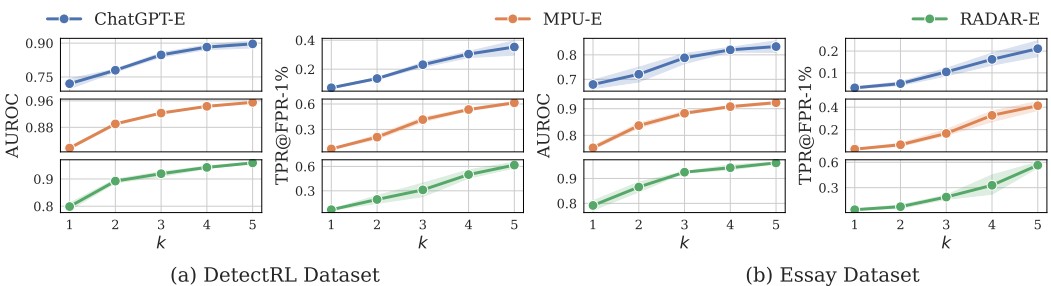

Figure 17: Performance (AUROC and TPR@FPR-1%) of the supervisor under different text number $k$ for longer text. The supervisor is trained on PaLM texts on DetectRL and GPT4All texts on Essay.

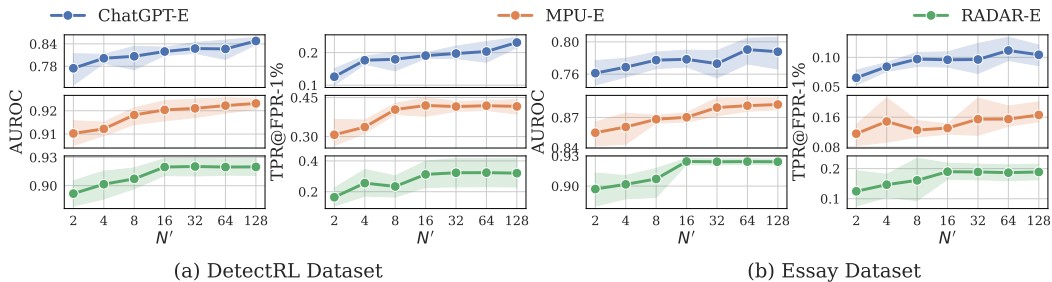

Figure 18: Performance (AUROC and TPR@FPR-1%) of the supervisor under different longer text numbers per batch. The supervisor is trained on PaLM texts on DetectRL and GPT4All texts on Essay.

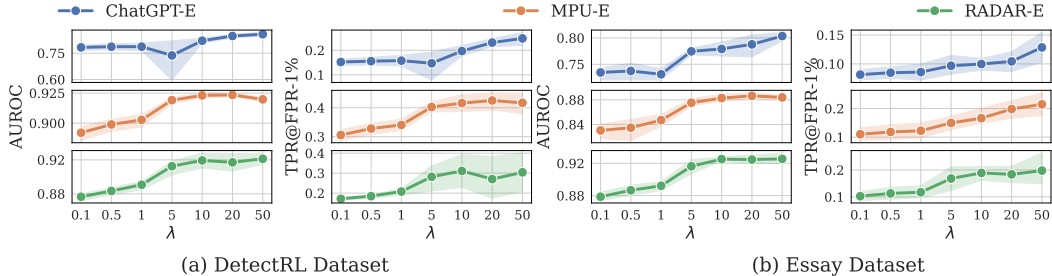

Figure 19: Performance (AUROC and TPR@FPR-1%) of the supervisor under different supervision loss coefficient $\lambda$. The supervisor is trained on PaLM texts on DetectRL and GPT4All texts on Essay.

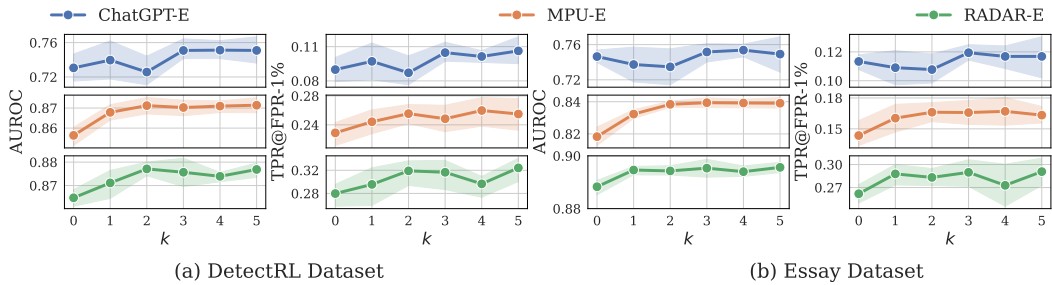

Figure 20: Performance (AUROC and TPR@FPR-1%) of the enhanced detectors under different text number $k$ for longer text. The detector is trained on PaLM texts on DetectRL and GPT4All texts on Essay.

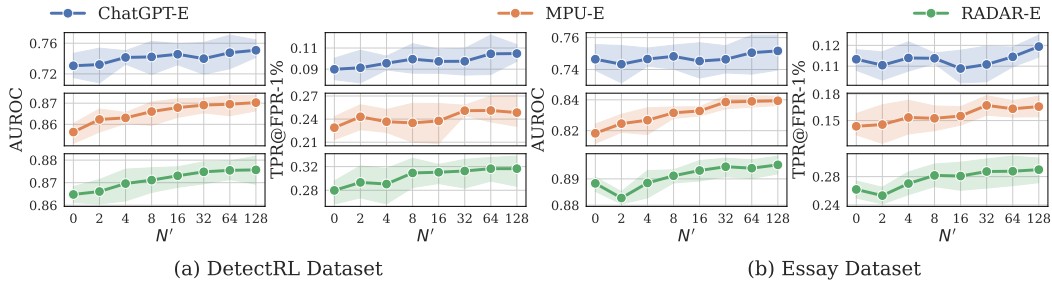

Figure 21: Performance (AUROC and TPR@FPR-1%) of the enhanced detectors under different longer text numbers per batch. $N' = 0$ represents the original model without enhancement. The detector is trained on PaLM texts on DetectRL and GPT4All texts on Essay.

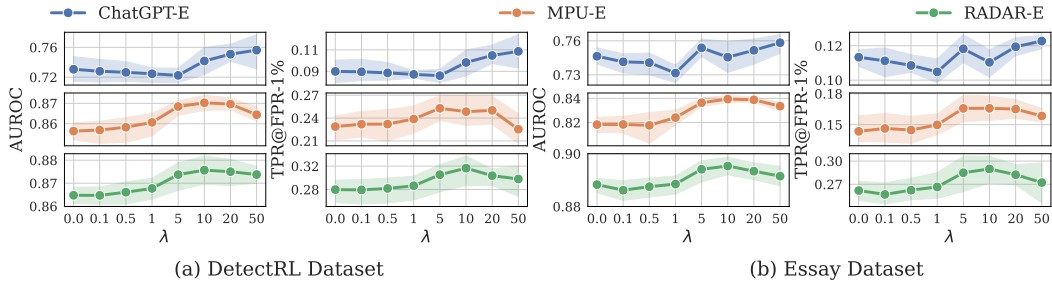

Figure 22: Performance (AUROC and TPR@FPR-1%) of the enhanced detectors under different supervision loss coefficient $\lambda$. The detector is trained on PaLM texts on DetectRL and GPT4All texts on Essay.

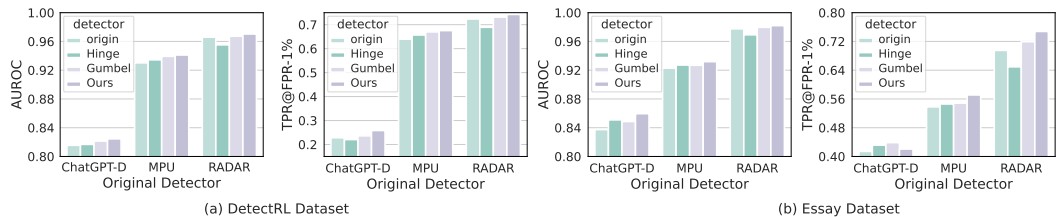

(a) DetectRL Dataset

(b) Essay Dataset

Figure 23: Performance comparison with category-based supervision signals (Hinge and Gumbel). The detector is trained on PaLM texts on DetectRL and GPT4All texts on Essay.

