# OpenReview forum: "Advancing Machine-Generated Text Detection from an Easy to Hard Supervision Perspective"
_NeurIPS.cc/2025/Conference — NeurIPS 2025 poster_

### Official Review · Reviewer_Vuke · 2025-06-29

**Clarity:** 2
**Significance:** 2
**Originality:** 4
**Rating:** 5
**Confidence:** 4

**Summary:**

This paper presents a new algorithm for classification-based detection of AI-generated text.

The key intution in the paper's method is that longer chunks of text provide a more reliable supervision for detecting AI-written text, a fact that's been well established in prior literature.

This paper's converts this intuition into a training-time algorithm. During training, gumbel-softmax sampled predictions over continguous text chunks are aggregated based on the detector's current weights, and then sent to an external "supervisor" network. The supervisor and detectors are then jointly optimized, with a much higher weight (10:1) on the supervisor network.

Instead of optimizing on potentially noisy individual text labels, the detector is incentivized to optimize for detecting a longer chunk more holistically --- potentially learning more generalizable features.

**Questions:**

See review above

**Ethical Concerns:**

["NO or VERY MINOR ethics concerns only"]

**Final Justification:**

Nice new experiments on new LLMs which should be incorporated in the final version. Agreement with other reviewers.

**Limitations:**

Yes

**Quality:**

3

**Strengths And Weaknesses:**

*Strengths*

1. This is a pretty interesting and novel formulation for AI-generated text detection modeling. It's a nice conversion of the "long text is easier to detect" to an operational learning algorithm. I couldn't think of any cleaner alternative formulations during the review.

2. The proposed formulation seems to beat ablated baselines across all tested settings, proving its effectiveness.

3. The paper has great experimental breadth, including experiments on multiple datasets, base LLMs, detector baselines. The paper also includes experiments on paraphrased outputs and mixed-text detection which strengthens the contribution.

4. The proposed method has no extra latency at inference time, and only a slight increase in training time latency.

*Weaknesses*

1. The base LLMs (whose text is being detected) in this paper are quite old --> ChatGPT / GPT-turbo / Claude / PaLM are models from 2022-2023. Do these methods generalize to newer LLMs from 2024/2025, which likely have a very different distribution?

2. The paper was lacking in qualitative analysis and the kinds of features being learnt by this detector (vs the baseline). I would imagine this detector has learnt more robust / generalizable features pertaining to AI-generated text rather than overfit to noisy datapoints. Is there any qualitative evidence / win-analysis over individual test datapoints to justify this?

3. The paper's presentation could really be improved, it was really hard to understand the paper's contributions from the abstract/introduction. I felt the authors were underselling their work by talking mostly about previous efforts. The mathematical notation and naming is also confusing. For instance, the notation in L179, the use of `'` is too subtle to distinguish differences between detector and supervisor labels. Also in L244 the naming isn't great, looking at the table it's hard to understand `-E` refers to the method proposed in this work.

4. As a thought experiment, would this method work at scale? While I agree that supervising long-form detection indirectly is a good inductive bias in this setup, tradtionally inductive biases have usually lost out to scaled training (http://www.incompleteideas.net/IncIdeas/BitterLesson.html).

---

> ### Author Rebuttal · Authors · 2025-07-31
>
> Many thanks for your positive comments and constructive feedback. Please see the below responses to your comments.
>
> > **[W1]**  The base LLMs (whose text is being detected) in this paper are quite old --> ChatGPT / GPT-turbo / Claude / PaLM are models from 2022-2023. Do these methods generalize to newer LLMs from 2024/2025, which likely have a very different distribution?
> >
>
> Thanks for your valuable suggestion. Based on the Essay prompts, we employed the latest LLMs—GPT-4o, GPT-4.1, DeepSeek-R1, and Llama4 Maverick—to generate machine text. The detection results are shown in Tables 7 and 8 below, where detectors are trained on GPT4All texts from Essay dataset. As can be seen, our proposed enhancement strategy remains significantly effective on these newer LLMs. Moreover, compared to the LLMs used in Table 2 of our paper, the detection performance on these more advanced LLMs has declined, underscoring the ongoing arms race between detection and large-model development: detectors design stronger strategies based on current generation models, and then newly emerging LLMs produce higher-quality text that is harder to distinguish, thus driving further progress on both sides.
>
> Table 7. Performance concerning TPR@FPR-1% (%). The detection model is trained on text generated by GPT4All.
>
> | Method | GPT-4o | GPT-4.1 | DeepSeek-R1 | Llama4 Maverick | Avg. |
> | --- | --- | --- | --- | --- | --- |
> | ChatGPT-D | 2.98 | 1.20 | 0 | 45.87 | 12.51 |
> | **ChatGPT-E** | **3.47** | **1.33** | 0 | **47.91** | **13.18** |
> | MPU | 6.76 | 3.07 | **0.13** | 59.60 | 17.39 |
> | **MPU-E** | **8.53** | **3.64** | 0.09 | **62.18** | **18.61** |
> | RADAR | 66.67 | 50.53 | 55.78 | 86.09 | 64.77 |
> | **RADAR-E** | **77.47** | **63.16** | **68.53** | **91.56** | **75.18** |
>
> Table 8. Performance concerning AUROC (%). The detection model is trained on text generated by GPT4All.
>
> | Method | GPT-4o | GPT-4.1 | DeepSeek-R1 | Llama4 Maverick | Avg. |
> | --- | --- | --- | --- | --- | --- |
> | ChatGPT-D | 54.94 | 36.48 | 9.37 | 95.48 | 49.07 |
> | **ChatGPT-E** | **57.68** | **38.14** | **12.00** | **95.90** | **50.93** |
> | MPU | 76.37 | 63.57 | 40.98 | 97.20 | 69.53 |
> | **MPU-E** | **79.02** | **66.81** | **44.84** | **97.54** | **72.05** |
> | RADAR | 98.79 | 97.27 | 98.55 | 99.50 | 98.53 |
> | **RADAR-E** | **99.18** | **98.09** | **99.01** | **99.67** | **98.99** |
>
> > **[W2]**  The paper was lacking in qualitative analysis and the kinds of features being learnt by this detector (vs the baseline). I would imagine this detector has learnt more robust / generalizable features pertaining to AI-generated text rather than overfit to noisy datapoints. Is there any qualitative evidence / win-analysis over individual test datapoints to justify this?
> >
>
> Traditional case studies are more suitable for metric-based methods, where detection are based on specific interpretable features. However, it is challenging for model-based detectors that our work focuses on. This is because they are highly complex black-box models that lack interpretable features.
>
> Fortunately, we introduce a knowledge distillation-based analysis method to verify how the proposed method improves detection. Specifically, the core of our framework is that the supervisor guides the detector to learn more exact “golden” labels (Theorem 3.4). Thus, the detector’s improvement should stem from learning better soft scores from the supervisor. Accordingly, we can use the detectors before and after enhancement as teacher models and distill a student model respectively.  By comparing the performance of these student models, we can assess whether our method strengthens detection through this mechanism. If the student distilled from the enhanced detector outperforms the student distilled from the original detector, it indicates that our approach has indeed enabled the detector to learn better detection scores from the supervisor, thereby improving detection performance. Tables 9 and Table 10 below present the distillation performance on the Essay and DetectRL datasets, where Origin-KD and Ours-KD denote the original RADAR detector and our enhanced RADAR (i.e., RADAR-E) guided MPU detector, respectively. Comparing  Origin-KD and Ours-KD, we observe that the latter achieves superior performance, proving that the proposed method learns better detection scores from the supervisor, echoing Theorem 3.4.
>
> Table 9. The Impact of Teacher Model (RADAR) Enhancement on Knowledge Distillation on DetectRL dataset.
>
> | Method | PaLM | ChatGPT | Claude | Llama-2 | Avg |
> | --- | --- | --- | --- | --- | --- |
> | Origin-KD | 66.01  | 73.28  | 15.77  | 83.66  | 59.68  |
> | Ours-KD | 67.49  | 76.00  | 16.81  | 84.77  | 61.27  |
>
> Table 10. The Impact of Teacher Model (RADAR) Enhancement on Knowledge Distillation on Essay dataset.
>
> | Method | GPT4All | ChatGPT | ChatGPT-turbo | ChatGLM | Dolly | Claude | Avg. |
> | --- | --- | --- | --- | --- | --- | --- | --- |
> | Origin-KD | 69.34  | 41.29  | 93.97  | 24.92  | 64.00  | 5.47  | 49.83  |
> | Ours-KD | 70.11  | 42.53  | 93.97  | 25.68  | 65.47  | 6.00  | 50.63  |
>
> > **[W3]**  The paper's presentation could really be improved, it was really hard to understand the paper's contributions from the abstract/introduction. I felt the authors were underselling their work by talking mostly about previous efforts. The mathematical notation and naming is also confusing. For instance, the notation in L179, the use of is too subtle to distinguish differences between detector and supervisor labels. Also in L244 the naming isn't great, looking at the table it's hard to understand refers to the method proposed in this work.
> >
>
> Thank you for your thoughtful suggestion, which provides a clear roadmap for improving the paper. We will address each of the points:
>
> - **Clarification of Contributions**. We will revise the abstract and introduction to focus on our contributions. In the abstract, we remove the discussion of knowledge distillation to more directly state our contributions regarding the research problem, technical framework, theoretical insights, and extensive evaluations. In the introduction, we remove the discussion on noisy label learning and streamline the second paragraph to provide the necessary background instead of underselling it. Furthermore, we emphasize our key contributions more: the third paragraph will highlight the significance of identifying the 'inexact supervision' problem, and the final paragraph will underscore the novelty of our proposed 'easy-to-hard supervisor' framework from the technical and theoretical aspects. These revisions are intended to give readers a clearer understanding of our work’s value and novelty.
> - **Clarification of Notation**. The supervisor’s input will be updated from $(x'',y')$ to the more meaningful $(x_{long},y_{long})$ to distinguish it from the detector input $(x,y)$ to avoid confusion. In addition, we change the enhanced version from the suffix “-E” to “-E2H” (i.e., easy-to-hard), and emphasize it in bold in L244.
>
> In addition, we will comprehensively correct grammatical errors and typos in the revised paper. Thank you again for your careful review, and we are grateful for the opportunity to improve our paper with these valuable suggestions.
>
> > **[W4]**  As a thought experiment, would this method work at scale? While I agree that supervising long-form detection indirectly is a good inductive bias in this setup, tradtionally inductive biases have usually lost out to scaled training (http://www.incompleteideas.net/IncIdeas/BitterLesson.html).
> >
>
> We thank the reviewers for raising questions regarding scalability in the context of “The Bitter Lesson.” We examine the scalability of our approach to large-scale scenarios from the perspectives of computational compatibility and framework generality.
>
> - **Computational compatibility.** A key aspect of our framework is its computational efficiency. Incorporating an auxiliary supervisor throughout the training process adds negligible overhead, as shown in Table 3 in the paper. This supervisor can be designed as a lightweight component that avoids bottlenecks, ensuring seamless integration into standard large-scale training pipelines with large models and massive datasets. In essence, the bias is in the training objective (Equation 5), not in a restrictive architectural change that would hinder scaling.
> - **Framework generality.** The proposed framework applies to model-based approaches that rely on data and computing power rather than metric-based methods centered on specific features. This reflects the philosophy of “The Bitter Lesson”—instead of forcing explanations of the detection mechanism through specific features, we learn to construct the most effective representations for detection using data and computation. Moreover, the enhancement framework is not built on narrowly designed knowledge; rather, it represents a general learning architecture under inexact supervision by using an easier supervisor to refine labels for the target model. Its applicability may extend beyond text detection to any inexact supervision problem characterized by inherently ambiguous labels.
>
> Essentially, our framework employs an efficient computational method (training the supervisor on longer texts) to provide better learning signals for another method (the detector), rather than imposing a rigid, artificially designed bias that limits learning and makes it scalable. Thank you again for this thought-provoking question, which provides a clearer positioning for our future exploration on larger-scale data.

---

> > ### Comment · Reviewer_Vuke · 2025-08-05
> > **Thank you, raising score to 5**
> >
> > Thank you for the very thorough rebuttal and extra experiments! I have raised my score to 5. Please include experiments on the newer LLMs with at least as much rigor as presented in this rebuttal.
> >
> > Also regarding qualitative analysis I just mean a win/loss table showcasing generalization, ideally some OOD test data points which the detector got wrong previously but is able to figure out correctly now.

---

> ### Author Response · Authors · 2025-08-05
> **Supplementary Quantitative Analysis and Thank You for Your Support**
>
> We sincerely thank you for your kind support and for raising your score. We will carefully incorporate these discussions into the final version of the paper.
>
> Besides, following your suggestion, we prepared some examples as follows for quantitative analysis (More examples will be added to the revised version). Specifically, the table below shows sequentially selected three MGTs from the DetectRL dataset. The text in the left column is accurately identified as "MGT" by both the baseline version of RADAR and our improved version RADAR-E. The text in the right column represents "win" cases for our approach—they are misclassified by RADAR but correctly detected by RADAR-E. We can find that the easy-to-detect texts (left column) typically exhibit more complex idioms, formal vocabulary, and convoluted expressions, which are typical of machine-generated text. Instead, the difficult-to-detect texts (right column), which our method successfully detects, are characterized by short, plain, straightforward, and somewhat colloquial sentences, which are much closer to a typical human-like distribution. The characteristics of difficult-to-detect text and the effectiveness of the enhancement strategy intuitively demonstrate the advantages of the proposed strategy in processing ambiguous machine-generated text.
>
> Thanks again for helping us improve the paper, and wish you the best in your research!
>
> Table 1. Case study on DetectRL.
>
> |  | MGTs that can be detected by both RADAR and RADAR-E | MGTs that can only be detected by RADAR-E |
> | --- | --- | --- |
> | Case 1 | When I thought of the comment, I wrote it as Milio's, and I thought'meh. This is just another pizza shop. But after eating there, I realized that Milio was not just another Pizza shop. The food is delicious, the service is very good, and the atmosphere is warm and seductive. …… | Tհe atmosphere is excellent with lots of big screen TV's to watch the game. The service was decеnt as well. The waitress was very friendly and watchful, but the kitchen seemed backed up because our food took over an hour to comе out. … |
> | Case 2 | The two-dimensional kagome lattice has been identified as a promising platform to realize quantum spin liquid states due to its unique geometry. In this paper, we report the synthesis and characterization of a series of two-dimensional kagome antiferromagnets, ZnxCu4-x(OD)6Cl2, with x = 0, 0.5, 1, 1.5, and 2. … | Sussex further improved their hopes of reaching the One-Day Taza knockout stages by beating Surrey for a third consecutive regrouped win. … |
> | Case 3 | Scientists, intrigued by my resilisence, devised a daring plan to send me into the eeigmatic depths of a black hole. Embracing my inner badass, I agreed, embarking on a journey that would test the ilmits of my extraordinary abilities. … | This place should have scas of glowing reviews! My wife and I had a great tife. It’s hard to believe that such a grea restaurant could be in a strip mall. … |

---

### Official Review · Reviewer_7nNE · 2025-07-02

**Clarity:** 3
**Significance:** 3
**Originality:** 3
**Rating:** 4
**Confidence:** 3

**Summary:**

In this work, the authors propose an easy-to-hard enhancement framework for training a detector on machine-generated text. It employs an easy supervisor targeting longer-text detection tasks to enhance the more challenging target detector. Extensive experiments validate the effectiveness of the proposed approach.

**Questions:**

See weakness

**Ethical Concerns:**

["NO or VERY MINOR ethics concerns only"]

**Final Justification:**

I think my main concerns have been addressed and tend to raise my score.

**Quality:**

3

**Strengths And Weaknesses:**

# Strengths

1. The paper is well-written and easy to follow.

2. The proposed easy-to-hard claim seems to be reasonable and practical.

# Weakness

1. The authors claim that the label of machine-generated text is hard to obtain but I cannot figure out the reason. I can simply use the text generated by LLM as the machine generated text. Do your indicate that it might be hard to find some human generated text currently?

2. It is unclear how the supervisor helps improve the detector using the same dataset as depicted in Algorithm 1.

3. Why does a single weak model (supervisor) can lead to a powerful model (detection).

4. It might be unfair for comparison by using the baselines trained on different datasets.

---

> ### Author Rebuttal · Authors · 2025-07-31
>
> Thank you for the valuable feedback! Please find the point-to-point responses below. Any further comments and discussions are welcome!
>
> > **[W1]**  The authors claim that the label of machine-generated text is hard to obtain but I cannot figure out the reason. I can simply use the text generated by LLM as the machine generated text. Do your indicate that it might be hard to find some human generated text currently?
> >
>
> We fully agree that **obtaining a text’s categorical label (hard label) is not difficult. Instead, the paper emphasize the difficulty of obtaining the underlying “golden” label (soft label)**, which needs to be distinguished from the categorical label you mentioned: given the increasing generative capabilities of LLMs and the extensive human–machine collaboration that blurs the boundary between them (as shown in Figure 1 in the paper), the text’s inherent “golden” label may not simply be “yes” or “no”, but rather a nuanced score indicating its “machine similarity” (the effectiveness of label smoothing in Figure 2 verifies this). In other words, while we can easily acquire a categorical label by determining whether the text is generated by an LLM, its intrinsic golden label (e.g., might be 0.85) is difficult to obtain because assigning such a score lies beyond human cognition. To avoid confusion, we will revise the manuscript to explicitly differentiate between the easily obtained “categorical label” and the hard-to-obtain “golden label”.
>
> > **[W2]**  It is unclear how the supervisor helps improve the detector using the same dataset as depicted in Algorithm 1.
> **[W3]**  Why does a single weak model (supervisor) can lead to a powerful model (detection).
> >
>
> Thank you for this comment! We will clarify how the supervisor enhances detection from both the methodological and evaluation perspectives:
>
> - **Methodology**. Although the supervisor’s data comes from the same dataset as the detector, the supervisor is trained on longer reconstructed texts gated by the detector, thereby guiding the detector to learn more precise “golden” labels for enhancement. Specifically, Theorem 3.2 shows that the supervisor’s performance is the lower bound of $1-1/2\cdot(1-\delta)^{2nk(1-\alpha)}$. This allows us to couple the detector’s performance to these factors (e.g., $k$), then optimizing the supervisor will optimize the relevant factors, thus improving the detector. To this end, we employ a Gumbel-based gating mechanism (Equation 3) linking $k$ to detector performance, such that a larger $k$ corresponds to higher detector accuracy. Accordingly, the enhancement roadmap is: **maximize supervisor performance → maximize $k$ → maximize detector performance**. More importantly, Theorem 3.4 indicates that **maximizing supervisor performance drives the mathematical expectation of the Gumbel distribution (i.e., the detector’s prediction) to converge to the underlying “golden” label.** In contrast, traditional supervised training forces the detector to converge to a hard label (i.e., 0 or 1). Given the inherent ambiguity of generated texts (Figure 1), this convergence to the underlying “golden” label is central to achieving enhancement.
> - **Verification**. To further verify whether the supervisor guides the detector to learn more accurate "golden" labels for enhancement, we introduce a knowledge distillation-based analysis method for verification. Specifically, we use the detectors before and after enhancement as teacher models, distill a student model respectively. By comparing the performance of these student models, we can verify whether the supervisor provides more accurate supervision signals. If the student distilled from the enhanced detector outperforms the student distilled from the original detector, it indicates that our approach has indeed enabled the detector to learn better detection scores from the supervisor, thereby improving detection performance. Tables 5 and 6 below present the distillation performance on the Essay and DetectRL datasets, where Origin-KD and Ours-KD denote the original RADAR detector and our enhanced RADAR (i.e., RADAR-E) guided MPU detector, respectively. By comparison, we observe that Ours-KD achieves superior performance, proving that the proposed method learns better detection scores from the supervisor to enhance detection.
>
> In summary, we methodologically and empirically demonstrate that weak supervisors guide target detectors to learn more accurate “golden” labels for augmentation.
>
> Table 5. The Impact of Teacher Model (RADAR) Enhancement on Knowledge Distillation on DetectRL dataset.
>
> | Method | PaLM | ChatGPT | Claude | Llama-2 | Avg |
> | --- | --- | --- | --- | --- | --- |
> | Origin-KD | 66.01  | 73.28  | 15.77  | 83.66  | 59.68  |
> | Ours-KD | 67.49  | 76.00  | 16.81  | 84.77  | 61.27  |
>
> Table 6. The Impact of Teacher Model (RADAR) Enhancement on Knowledge Distillation on Essay dataset.
>
> | Method | GPT4All | ChatGPT | ChatGPT-turbo | ChatGLM | Dolly | Claude | Avg. |
> | --- | --- | --- | --- | --- | --- | --- | --- |
> | Origin-KD | 69.34  | 41.29  | 93.97  | 24.92  | 64.00  | 5.47  | 49.83  |
> | Ours-KD | 70.11  | 42.53  | 93.97  | 25.68  | 65.47  | 6.00  | 50.63  |
>
> > **[W4]**  It might be unfair for comparison by using the baselines trained on different datasets.
> >
>
> Apologize for the unclear experimental setup. In fact, the compared ChatGPT-D, MPU, and RADAR are all fine-tuned on the used datasets (i.e., Essay and DetectRL) in our experiments, as described in Line 955.

---

> ### Comment · Reviewer_7nNE · 2025-08-05
> **Thanks for your rebuttal**
>
> Thanks for the authors' response. I think my main concerns have been addressed and tend to raise my score.

---

> > ### Author Response · Authors · 2025-08-05
> >
> > We sincerely thank you for confirming that your main concerns have been addressed and for your decision to raise the score. We will carefully incorporate these discussions into the final version of the paper. Thanks again for your time and wish you the best in your research!

---

> > ### Author Response · Authors · 2025-08-07
> >
> > Dear Reviewer 7nNE,
> >
> > We sincerely thank you for confirming that your main concerns have been addressed and for your intention to raise the score. As the deadline for the author-reviewer discussion phase (August 8th) is fast approaching, but we haven't yet seen a change in your score (initial: 3) on the system. We would be very grateful if you could clarify the score when you have a moment. Of course, if there is anything we need further clarification on, we would be happy to continue the discussion.
> >
> > Best regards,
> >
> > Authors of # 9278

---

### Official Review · Reviewer_a1gn · 2025-07-02

**Clarity:** 3
**Significance:** 2
**Originality:** 3
**Rating:** 5
**Confidence:** 4

**Summary:**

This paper tackles the challenge of detecting machine-generated text (MGT) under inexact or ambiguous labeling. The authors propose an easy-to-hard supervision framework where a simpler supervisor, trained on longer and more distinguishable texts, guides the training of a more complex target detector operating on shorter. The framework is theoretically justified and empirically validated across diverse datasets, tasks, and attack settings, showing consistent improvements over baseline detectors.

**Questions:**

1.	The paper builds on the premise that label ambiguity between human- and machine-generated text necessitates an easy-to-hard supervision framework. However, given the increasing fluency of modern language models, aren't there some case where human and AI texts are inherently indistinguishable? How does the proposed method address such ambiguity?
2.	How do the authors justify that longer texts improve separability between human and machine-generated content? Given that prior work [1] [2] links their difference to long-tail distributions, what is the connection between longer-text construction and capturing those distinctive patterns?
3.	While the paper provides extensive quantitative results, are there any concrete case studies ？These case studies may conduct between the proposed method and baselines. Such comparisons would help illustrate how the method improves detection in practice.
4.	To what extent does the effectiveness of the proposed method depend on the quality of the original labels? Given that even humans often fail to distinguish between real and machine-generated texts, how does the framework handle ambiguous?

**Ethical Concerns:**

["NO or VERY MINOR ethics concerns only"]

**Final Justification:**

I think they have well addressed my concerns, and I think it is a good paper.

**Limitations:**

1.	While text length correlates with detectability supported theoretically, real-world MGT (e.g., headlines or tweets) often appear in short form, where this assumption may break down.
2.	Although the paper constructs longer texts from labeled data, if the original labels are flawed or biased, the framework may reinforce these issues.
3.	Some theoretical results assume idealized conditions (e.g., total variation distances, independence), which may not strictly hold in practice. These need some evidences empirically.

**Quality:**

3

**Strengths And Weaknesses:**

Weaknesses:
1.	Figure 1 is the estanblised information from model collapse papers[1] [2];
2.	Theorem 3.1 Distribution Difference for Longer Text lacks of emprical supports.
3.	The Easy-to-Hard Supervision Framework seems a bit complex. The paper assumes the existence of underlying "golden" labels approximated by machine-likeness, but this definition may lack grounding in real-world annotation practices. I’m just wondering—would using only supervised fine-tuning (SFT) be sufficient to achieve good performance on your collected data?


Strengths:
1.	This paper provides solid formal theorems to justify why longer texts are more discriminative and how the supervisor’s performance bounds the detector.
2.	Authors conduct comprehensive experiments across tasks, detectors, and attack scenarios, with consistent gains.
[1] Shumailov I, Shumaylov Z, Zhao Y, et al. AI models collapse when trained on recursively generated data[J]. Nature, 2024, 631(8022): 755-759.
[2] How to Synthesize Text Data without Model Collapse?, ICML, 2025

---

> ### Author Rebuttal · Authors · 2025-07-31
>
> Many thanks for your positive comments and constructive feedback.
>
> > **[W1]**  Figure 1 is the estanblised information from model collapse papers;
> >
>
> We thank the reviewer for raising the interesting issue of model collapse. Although Figure 1 may resembel model collapse visually, the underlying mechanisms differ substantially:
>
> - **Model and data**. Model collapse is a degenerative process affecting generations of learned generative models, in which the data they generate end up polluting the training set of the next generation. However, MGT detector is a binary classifier (predicting the probability of AI-generated), not a generative model. Its training data is fixed and static, with no iterative generation-feedback loop.
> - **Phenomenon**. Model collapse leads to an accumulation of low-perplexity texts and a loss of the long-tail characteristic of human text. Instead, in the inexact supervision detection, the opposite occurs: predictions for human text concentrate at the head, while AI-generated text shows a pronounced long-tail distribution (see Figure 1 in our paper). This is because AI-generated texts possess human-like characteristics, resulting in a wide range of prediction probabilities.
>
> > [**W2**]  Theorem 3.1 Distribution Difference for Longer Text lacks of emprical supports.
> [**Q2-1**] How to justify that longer texts improve separability between human and machine-generated content?
> >
>
> Thanks for your suggestion. Due to the challenges of directly computing distributional differences, we follow prior work [1] and use detection performance (AUROC) as a proxy, leveraging its theoretical connection to the total variation (TV) distance between distributions (see Proposition 1 in [1]). We evaluate detector performance (ChatGPT-D and MPU) across varying maximum text lengths, as shown in Tables 3 below. Results demonstrate that longer texts enhance detection, indicating it tends to have better separability.
>
> Table 3. Performance (AUROC) under different maximum text lengths (32-192) on DetectRL.
>
> |  | 32 | 64 | 96 | 128 | 160 | 192 |
> | --- | --- | --- | --- | --- | --- | --- |
> | ChatGPT-D | 73.06 | 79.90 | 80.03 | 80.84 | 81.42 | 83.24 |
> | MPU | 85.64 | 90.77 | 92.21 | 92.03 | 92.32 | 92.42 |
>
> > **[W3]**  The Easy-to-Hard Supervision Framework seems a bit complex. … I’m just wondering—would using only supervised fine-tuning (SFT) be sufficient to achieve good performance on your collected data?
> >
>
> Thank you for your valuable comment!
>
> - **Regarding the framework's complexity.** Our framework does not change the original detector, but incorporating an auxiliary supervisor throughout the training process adds negligible overhead, as shown in Table 3 in the paper. Therefore, this supervisor can be designed as a lightweight component that avoids bottlenecks, ensuring seamless integration into standard large-scale training pipelines. In essence, our framework introduces a training objective (Equation 5), instead of making restrictive architectural changes that would hinder scalability. This ensures the simplicity and flexibility of our framework.
> - **About using only supervised fine-tuning techniques.** We clarify that the baseline methods to be enhanced all use fine-tuning techniques. As specified in Line 955, ChatGPT-D, MPU, and RADAR are all fine-tuned on the collected dataset. Extensive comparisons demonstrate that our easy-to-hard supervision strategy outperforms theirs.
>
> > **[Q1]**  … aren't there some case where human and AI texts are inherently indistinguishable? How does the proposed method address such ambiguity?
> **[Q4]**  To what extent does the effectiveness of the proposed method depend on the quality of the original labels? Given that even humans often fail to distinguish between real and machine-generated texts, how does the framework handle ambiguous?
> >
>
> We agree that some texts are inherently indistinguishable (e.g., “Please sit down”) and thus, effective detection—relying on quantifiable differences in data—cannot address such cases, which are beyond the current scope of MGT detection. Notably, the proposed framework does not aim to magically detect such inherently indistinguishable texts. Instead, our strength lies in providing reliable likelihood scores for texts that are challenging yet still differentiable, rather than a simple binary “yes” or “no” judgment. This is achieved by coupling the detector and supervisor via a gating mechanism (Equation 3), where optimizing the supervisor aligns the expected Gumbel output with the true soft label (Theorem 3.4), thus enhancing detection. Experimental results (see our response to your Q3) further confirm that the proposed method can learn more generalized detection scores.
>
> Finally, we emphasize that MGT detection and LLM development is a continuous “arms race”: as detectors strengthen, LLMs evolve to generate higher-quality, less distinguishable text, driving further advances on both sides. Ideally, LLMs produce inherently indistinguishable text and detectors will ultimately fail. We look forward to seeing such advances in generated text, as they will benefit the development of LLMs.
>
> > **[Q3]**  While the paper provides extensive quantitative results, are there any concrete case studies？…
> **[Q2-2]**  What is the connection between longer-text construction and capturing those distinctive patterns?
> >
>
> Thank you for your comment! Traditional case studies are suitable for metric-based methods with specific interpretable features, but are challenging for complex, black-box model-based detectors which our work focuses on. To this end, we introduce a knowledge distillation-based analysis method, which can also intuitively demonstrate the effectiveness of our method. Specifically, since the supervisor guides the detector to learn more precise “golden” labels (Theorem 3.4), the detector’s improvement should stem from learning better soft scores from the supervisor. Thus, we compare student models distilled respectively from detectors before and after enhancement: if the student of the enhanced detector outperforms the original, it confirms that our approach has enabled the detector to learn better detection scores from the supervisor, thereby improving detection performance.
>
> Due to space limitations, the results refer to Table 1 and Table 2 in the response to Reviewer 9f7L. By comparison, we observe that Ours-KD achieves superior performance, proving that the proposed method learns better detection scores from the supervisor.
>
> > **[L1]**  While text length correlates with detectability supported theoretically, real-world MGT (e.g., headlines or tweets) often appear in short form, where this assumption may break down.
> >
>
> Thank you for this important question. In MGT detection, short texts often fall into the “inherently indistinguishable” category you mentioned above and thus commonly excluded, as seen in MMD [2] (filtering texts <5 words) and M4GT-Bench [3] (filtering texts <50 words).
>
> Nonetheless, the proposed Easy-to-Hard framework offers hope for short-text detection: our framework does not impose length restrictions on the target detector or the supervisor, suggesting potential for short-text detection. Although this is beyond our work, we consider this a valuable direction for future research.
>
> > **[L2]**  Although the paper constructs longer texts from labeled data, if the original labels are flawed or biased, the framework may reinforce these issues.
> >
>
> We thank the reviewer for this critical question. The proposed framework may help mitigate, rather than reinforce. Assuming b% of the machine text is mislabeled (similar analysis for mislabeled human text), i.e., b% of  human text mixed in, this increases the mixed ratio $\alpha$ in Theorem 3.1. Similar to the proof of Theorem 3.4, optimizing the supervisor encourages the detector to predict those noisy texts as 0 (i.e., correctly classified as human text); otherwise, the supervisor can be further optimized when $\alpha’>\alpha$.
>
> To further verify, we randomly flipped 10% of the labels and then evaluated detectors trained on these noisy data, as shown in Table 4 below. First, we found that the proposed enhanced framework remained effective. Second, compared to the noiseless results in Tables 1 and 2 in the paper, our method is generally less affected by noise, verifying our mitigation efforts.
>
> Table 4. The Impact of noise label regarding TPR-FPR-1% on DetectRL dataset.
>
> | Method | Google-PaLM | ChatGPT | Claude-instant | Llama-2-70b | Avg. |
> | --- | --- | --- | --- | --- | --- |
> | ChatGPT-D | 20.77 | 14.26 | 3.44 | 38.15 | 19.15 |
> | **ChatGPT-E** | **21.98** | **20** | **4.45** | **42.03** | **22.11** |
> | MPU | 60.40 | 64.55 | 12.61 | 77.87 | 53.86 |
> | **MPU-E** | **64.52** | **70.80** | **14.81** | **81.51** | **57.91** |
> | RADAR | 71.92 | 77.13 | 31.55 | 80.42 | 65.25 |
> | **RADAR-E** | **80.05** | **80.87** | **39.73** | **84.08** | **71.18** |
>
> > **[L3]**  Some theoretical results assume idealized conditions (e.g., total variation distances, independence), which may not strictly hold in practice.
> >
>
> Thanks for your question. First, we assume a nonzero total variation distance between generated and human text, which is practical: if $TV(m,h)=0$, any text are inherently indistinguishable and detection becomes meaningless. Second, our assumption of text comprising $n$ independent sequences is general, as it holds even for dependent text when $n=1$. We will clarify these points in the revised paper.
>
> Due to space limitations, we only provide DetectRL results in our reply. Similar findings are found in Essay, and we are happy to continue to provide full results during the discussion phase if needed.
>
> [1] Position: On the Possibilities of AI-Generated Text Detection. ICML’24.
>
> [2] Detecting machine-generated texts by multi-population aware optimization for maximum mean discrepancy. ICLR’24.
>
> [3] M4GT-Bench: Evaluation Benchmark for Black-Box Machine-Generated Text Detection. ACL’24.

---

### Official Review · Reviewer_9f7L · 2025-07-02

**Clarity:** 2
**Significance:** 3
**Originality:** 3
**Rating:** 5
**Confidence:** 4

**Summary:**

The paper proposes an easy-to-hard enhancement framework for machine-text detectors. It enhances the performance a detector by relying on supervision signals coming from another that is trained to distinguish longer texts. Relying on previous theoretical insights regarding the increase in distributional distance as the size of the text increases, the authors propose a framework where optimizing the supervisor indirectly optimizes the detector.

**Questions:**

•	Clarification – In Equation 3, there is $x^j$ multiplied by the Gumbel sample, however, if I understand it correctly the $x^j$ is in fact a string. Or, was it mean that the supervisor takes the set of Gumbel-softmax samples from the detector, and then predicts human or machine?

•	Generally, I think a stronger Knowledge Distillation baseline is needed (W3), more ablations are necessary (W5), and the writing has to be polished (W1). Tackling those, in that order, would ease some of my concerns.

**Ethical Concerns:**

["NO or VERY MINOR ethics concerns only"]

**Final Justification:**

The paper makes a significant contribution, extending the insight of Chakraborty's theorems into a recipe for training better machine-text detectors. The authors have run a good amount of experiments and addressed my concerns in their rebuttal.

**Limitations:**

Yes

**Paper Formatting Concerns:**

No concerns.

**Quality:**

3

**Strengths And Weaknesses:**

Strengths:

•	The method is generally interesting and novel, and is a nice extension of Chakraborty’s work (https://arxiv.org/pdf/2304.04736) for enhancing the robustness of detectors.

•	The author’s deal with the inherent uncertainty in the labels of human-written and machine-generated text, better focusing in on the more relevant case where the label is in fact smooth (partial machine and human).

Weaknesses:

•	W1- I found the language of this paper very hard to follow, and it needs a lot of polishing before it’s published.

•	W2- Line 47 states that Figure 2 is clear in showing that label smoothing exhibits better performance. It’s not clear from the figure that the label smoothing variants do exhibit better performance. In fact, the variance of the results is very high, and in fact, many of the medians are worse than those that use hard labels (left hand side in particular). In general, a lot of the results on mixed texts tend to be quite marginal (Figure 4), with strong models (RADAR) remaining close to the performance of their enhanced counterparts.

•	W3- When comparing against knowledge distillation, it would’ve been good to also report distilling a version of RADAR that has been trained on long texts against a version of RADAR that hasn’t to more closely mimic your approach.

•	W4- I found that many of the theorems are quite similar to Chakraborty’s (https://arxiv.org/pdf/2304.04736) and distract from the most important one being Theorem 3.4 (the effectiveness of the approach). For example, increasing the length of the text  amplifies the distributional distance is something that was already shown by Chakraborty. Similar comments hold for Theorem 3.2, in that the larger the k (the larger the text), the better the AUROC is also something that is known from Chakraborty.

•	W5 – Ablations regarding the number of texts concatenated are missing. Also, although randomly splicing sentences does preserve the label as machine-generated or human-written, it does shift the distribution of natural text generated by humans or machines.

---

> ### Author Rebuttal · Authors · 2025-07-31
>
> Many thanks for your positive comments and constructive feedback. Please see the responses to your comments below.
>
> > **[W1]**  I found the language of this paper very hard to follow, and it needs a lot of polishing before it’s published.
> >
>
> Thank you for your suggestion!  We will comprehensively revised the paper to make it clearer, including (1) streamlining the discussion of related work to only provide the essential background; (2) rewriting the introduction to consistently emphasize our contributions in each paragraph, e.g., highlighting the identification of the ‘inexact supervision’ in Paragraph 3 to provide clarity about the contribution of research scenario; (3) refining the notation for clarity, for example by updating $(x', y')$ to the more meaningful$(x_{long}, y_{long})$ and changing our model's suffix from '-E' to a more meaningful '-E2H' (Easy-to-Hard); and (4) correcting grammatical errors and typos, e.g., "simple to hard" was changed to "easy to hard". We are confident that these revisions will substantially improve the paper's readability.
>
> > **[W2]**  Line 47 states that Figure 2 is clear in showing that label smoothing exhibits better performance. It’s not clear from the figure that the label smoothing variants do exhibit better performance. In fact, the variance of the results is very high, and in fact, many of the medians are worse than those that use hard labels (left hand side in particular). In general, a lot of the results on mixed texts tend to be quite marginal (Figure 4), with strong models (RADAR) remaining close to the performance of their enhanced counterparts.
> >
>
> We acknowledge that with such a coarse label smoothing strategy, even a small smoothing factor cannot prevent some labels from deviating further from the underlying ground-truth labels. This may introduce noise and lead to greater training instability, i.e., the high variance you pointed out. Nonetheless, it is undeniable that **the higher upper bound of label smoothing indicates its potential to enhance detection performance**. At the same time, **its instability highlights that addressing inexact supervision is a challenging task, which further highlights the necessity of our work.** To avoid confusion, we will revise the statement to: “a higher upper bound for label smoothing means that it has the potential for enhancement.” For the results shown in Figure 4, we believe that achieving consistent positive gains for state-of-the-art detectors, with negligible training overhead, constitutes a valuable contribution.
>
> Thank you again for your feedback, which will further improve the clarity of our paper and better highlight our contributions.
>
> > **[W3]**  When comparing against knowledge distillation, it would’ve been good to also report distilling a version of RADAR that has been trained on long texts against a version of RADAR that hasn’t to more closely mimic your approach.
> >
>
> Thanks for your insightful suggestion. A comparison between the distilled versions of RADAR and RADAR-E (our method) will provide better insight into whether our approach learns higher-quality underlying labels. Tables 1 and 2 below present the distillation detection performance on the Essay and DetectRL datasets, where Origin-KD and Ours-KD denote the original RADAR detector and our enhanced version (i.e., RADAR-E) guided MPU detector, respectively. Comparing Origin-KD and Ours-KD, we observe that the latter achieves superior performance, reflecting that our method yields higher-quality labels that are closer to the true underlying labels. Furthermore, when comparing distilled and original versions (Origin-KD vs. Origin, Ours-KD vs. Ours), the distilled versions perform worse. This indicates the limitations of knowledge distillation under inexact supervision and highlights the superiority of the proposed enhanced framework.
>
> Table 1. Performance (TPR@FPR-1%) of distilled MPU on DetectRL dataset.
>
> | Method | PaLM | ChatGPT | Claude | Llama-2 | Avg |
> | --- | --- | --- | --- | --- | --- |
> | Origin | 70.43  | 79.16  | 18.22  | 87.89  | 63.92  |
> | Origin-KD | 66.01  | 73.28  | 15.77  | 83.66  | 59.68  |
> | Ours-KD | 67.49  | 76.00  | 16.81  | 84.77  | 61.27  |
> | Ours | 75.75  | 84.23  | 19.78  | 90.31  | 67.52  |
>
> Table 2. Performance (TPR@FPR-1%) of distilled MPU on Essay dataset.
>
> | Method | GPT4All | ChatGPT | ChatGPT-turbo | ChatGLM | Dolly | Claude | Avg. |
> | --- | --- | --- | --- | --- | --- | --- | --- |
> | Origin | 71.07  | 48.98  | 94.78  | 28.69  | 71.64  | 7.64  | 53.80  |
> | Origin-KD | 69.34  | 41.29  | 93.97  | 24.92  | 64.00  | 5.47  | 49.83  |
> | Ours-KD | 70.11  | 42.53  | 93.97  | 25.68  | 65.47  | 6.00  | 50.63  |
> | Ours | 78.31  | 52.09  | 96.88  | 32.08  | 74.09  | 9.20  | 57.11  |
>
> > **[W4]**  I found that many of the theorems are quite similar to Chakraborty’s (https://arxiv.org/pdf/2304.04736) and distract from the most important one being Theorem 3.4 (the effectiveness of the approach). For example, increasing the length of the text amplifies the distributional distance is something that was already shown by Chakraborty. Similar comments hold for Theorem 3.2, in that the larger the k (the larger the text), the better the AUROC is also something that is known from Chakraborty.
> >
>
> We appreciate the reviewer’s careful observation regarding the connection with the foundational work [1]. Our theoretical results build upon Chakraborty’s foundational theory but differ substantially in several aspects:
>
> 1. **Lower bounds vs. upper bounds.** While [1] establishes critical theoretical lower bounds for detection, we extend to the theoretical upper bounds (Theorem 3.2 and the right-hand side of Theorem 3.1) to further explore the potential of detection. Although they are both related to text length, they measure different bounds.
> 2. **Guidance for the proposed framework.** In addition to exploring the detection potential, it is more important to guide the design of the proposed framework: by coupling the influencing factors of the supervisor’s upper bound with the detector, maximizing the supervisor's performance indirectly optimizes the detector, i.e., maximizing the supervisor's performance -> optimizing the influencing factors of the upper bound -> optimizing the detector. Instead, such a guided optimization mechanism cannot be achieved with the theoretical lower bounds.
>
> Therefore, although the findings revealed by these theorems appear similar, their bounds and contributions are fundamentally different. We will revise the manuscript to more clearly state that our work extends from lower bounds in prior theory to a guideable upper bound, thereby avoiding confusion with established theory in [1].
>
> > **[W5]**  Ablations regarding the number of texts concatenated are missing. Also, although randomly splicing sentences does preserve the label as machine-generated or human-written, it does shift the distribution of natural text generated by humans or machines.
> >
>
> Thank you for your suggestion! We wish to clarify that the sensitivity analyses for the number of concatenated texts ($k$) have provided in the Appendix D.8: the influence of $k$ on the supervisor's performance is detailed in **Figure 16**, while its impact on the detector's efficacy is presented in **Figure 19,** where $k=1$ corresponds to the ablation condition without text concatenation. The results from both figures consistently indicate that an increase in $k$ leads to a corresponding performance improvement for both components.
>
> Besides, we fully agree that the distribution of concatenated sentences differs from that of natural text. Therefore, our framework takes this into account: **the target detector is trained only on the original, natural text,** which allows it to learn the true data distribution. Instead, **the concatenated text is used only to train the supervisor**. **This intentional separation ensures the distribution shift does not compromise the target detector's performance.**
>
> > **[Q1]**  Clarification – In Equation 3, there is $x_j$ multiplied by the Gumbel sample, however, if I understand it correctly the  $x_j$ is in fact a string. Or, was it mean that the supervisor takes the set of Gumbel-softmax samples from the detector, and then predicts human or machine?
> >
>
> In our implementation, the product operator $\odot$ is performed at the input embedding level, as shown in Line 187 of our provided code (https://anonymous.4open.science/r/Easy2Hard/methods/ChatGPT_D.py). Therefore, in Equation 3, $x^{(j)}$ should be revised with $e_{x^{(j)}}$, where $e_{x^{(j)}}$ denotes the input embedding corresponding to $x^{(j)}$. For example, if a longer text of length 3 is represented as $[x^{(1)},x^{(2)},x^{(3)}]$ and the detector gives the gumbel softmax value of them as $[1,0,1]$, then the input to the supervisor is $[e_{x^{(1)}},0,e_{x^{(3)}}]$. We will revise the notation accordingly in the updated version to avoid confusion.
>
> > **[Q2]**  Generally, I think a stronger Knowledge Distillation baseline is needed (W3), more ablations are necessary (W5), and the writing has to be polished (W1). Tackling those, in that order, would ease some of my concerns.
> >
>
> We sincerely thank the reviewer for providing this clear summary of the main points for improvement. We have made relevant responses above, and we hope that our responses have addressed your concerns. If you have any further questions or suggestions, we welcome the rebuttal opportunity to communicate further.
>
> [1] Position: On the possibilities of AI-generated text detection. ICML 2024.

---

> > ### Comment · Reviewer_9f7L · 2025-08-04
> >
> > Thank you for the thoughtful rebuttal. I appreciate the inclusion of the knew knowledge distillation results, the clarifications on the differences between the theorems in the paper and how they differ from Chakraborty's work, and the clarification on how the gumbel-softmax outputs were integrated. The reviewer thinks this is interesting work and novel work and is raising his score accordingly.

---

> > > ### Author Response · Authors · 2025-08-05
> > >
> > > We sincerely thank you for your support of our work and for raising your score. We will carefully incorporate these valuable suggestions into the final version of the paper. Thanks again for your help, and wish you the best in your research!

---

### Note · Authors · 2025-08-13

Dear Area Chair and Reviewers,

We sincerely appreciate AC in chairing our paper and all reviewers for your deep understanding of our method, the expertise you possess, and the constructive, considerate and insightful suggestions. Besides, we are grateful that all reviewers had a positive opinion of our paper.

Below is a summary of the reviews and our responses:

- **Reviewer 9f7L** praised our work, stating that "**the method is generally interesting and novel**". Regarding concerns about the knowledge distillation experiments and differences from existing theory, we added relevant experiments and clarified the differences with Chakraborty's work. This response was recognized by Reviewer 9f7L, "**The reviewer thinks this is interesting and novel work**”.
- **Reviewer a1gn** praised us for "**This paper provides solid formal theories**" and "**Authors conduct comprehensive experiments across tasks**". During the rebuttal phase, we added new experiments to address the reviewer's concerns about theoretical correctness and label flaws, and clarified the reasonableness of our theoretical assumptions. This response was recognized by Reviewer a1gn, who said, "**I think they have well addressed my concerns, and I think it is a good paper**”.
- **Reviewer 7nNE** praised us for "**The paper is well-written and easy to follow,**" and "**The proposed easy-to-hard claim seems reasonable and practical**". During the rebuttal phase, we clarified misunderstandings about the core concept of "golden labels" and demonstrated how our weak supervisor guides the detector through theoretical explanations and knowledge distillation experiments. This response was recognized by Reviewer 7nNE, who said, "**I think my main concerns have been addressed and tend to improve my score**”.
- **Reviewer Vuke** praised us for "**This is a pretty interesting and novel formulation**" and "**The paper has great experimental breadth**". During the rebuttal phase, we addressed the reviewer's concerns about the effectiveness, generalization, and flexibility by supplementing experiments on the latest LLM texts and elaborating on our framework’s scalability. This was recognized by Reviewer Vuke, who said, "**Thank you for the very thorough rebuttal and extra experiments! I have raised my score to 5**”.

We will incorporate these enhancements, along with your recommendations, into the final manuscript. We look forward to your final assessments.

Best regards,

Authors of # 9278

---

### Decision · Program_Chairs · 2025-09-17

**Decision:**

Accept (poster)

**Comment:**

This paper proposes a novel perspective on machine-generated text detection through an easy-to-hard enhancement. Reviewers are uniformly positive, highlighting the strong technical contributions, solid empirical validation across diverse datasets, and clear presentation. The rebuttal further clarified minor questions and reinforced confidence in the work. Given the overall enthusiasm and the paper’s timely and well-executed contributions, I recommend acceptance.